# TwinWeaver: An LLM-Based Foundation Model Framework for Pan-Cancer Digital Twins

Nikita Makarov [1 2 3]  Maria Bordukova [1 2 3]  Lena Voith von Voithenberg [4]  Estrella Pivel-Villanueva [1]
Sabrina Mielke [5]  Jonathan Wickes [6]  Hanchen Wang [6 7]  Mingyu Derek Ma [5]  Keunwoo Choi [5]
Kyunghyun Cho [5 8]  Stephen Ra [5]  Raul Rodriguez-Esteban [9 +]  Fabian Schmich [1 +]  Michael Menden [2 10 +]

## Abstract

Precision oncology requires forecasting clinical events and trajectories, yet modeling sparse, multi-modal clinical time series remains a critical challenge. We introduce TwinWeaver, an open-source framework that serializes longitudinal patient histories into text, enabling unified event prediction as well as forecasting with large language models, and use it to build Genie Digital Twin (GDT) on 93,054 patients across 20 cancer types. In benchmarks, GDT significantly reduces forecasting errors, achieving a median Mean Absolute Scaled Error (MASE) of 0.87 compared to 0.97 for the strongest time-series baseline ($p < 0.001$). Furthermore, GDT improves risk stratification, achieving an average concordance index (C-index) of 0.703 across survival, progression, and therapy switching tasks, surpassing the best baseline of 0.662. GDT also generalizes to out-of-distribution clinical trials, matching trained baselines at zero-shot and surpassing them with fine-tuning, achieving a median MASE of 0.75–0.88 and outperforming the strongest baseline in event prediction with an average C-index of 0.672 versus 0.652. Finally, TwinWeaver enables an interpretable clinical reasoning extension, providing a scalable and transparent foundation for longitudinal clinical modeling.

[+]Joint supervision [1]Roche, Penzberg, Germany [2]Computational Health Center, Helmholtz Munich, Germany [3]Department of Biology, Ludwig Maximilian University, Munich, Germany [4]Roche, Schlieren, Switzerland [5]Genentech, New York, USA [6]Genentech, South San Francisco, USA [7]Department of Computer Science, Stanford University, Stanford, USA [8]Center for Data Science, New York University, New York, USA [9]Roche, Basel, Switzerland [10]Department of Biochemistry and Pharmacology, Bio21 Institute, The University of Melbourne, Melbourne, Australia. Correspondence to: Fabian Schmich <fabian.schmich@roche.com>, Michael Menden <michael.menden@unimelb.edu.au>.

*Proceedings of the 43$^{rd}$ International Conference on Machine Learning*, Seoul, South Korea. PMLR 306, 2026. Copyright 2026 by the author(s).

## 1. Introduction

Precision oncology aims to personalize cancer care through computational modeling of patient trajectories (Zhuang et al., 2025). Patient Digital Twins (PDTs) serve as digital replicas of patients with a bidirectional information flow, learning patient-specific representations from longitudinal, multi-modal molecular and clinical data to support forecasting and simulation. This necessitates scalable methods that can handle heterogeneous and sparse real-world time series (Kamel Boulos & Zhang, 2021; Bordukova et al., 2024). Recent advances in foundation models have further increased the feasibility of large-scale clinical trajectory modeling (Wornow et al., 2023b; Makarov et al., 2025).

Here, we introduce TwinWeaver, a longitudinal framework for large language model (LLM) generated PDTs, and Genie Digital Twin (GDT), a pan-cancer model instantiated using TwinWeaver to simulate patient trajectories (Figure 1). TwinWeaver serializes patient journeys into text, which GDT uses to forecast clinical events and trajectories across 20 cancer indications. Leveraging a large-scale dataset of 93,054 real-world oncology patients from the Flatiron Health-Foundation Medicine Clinico-Genomic Database (FH-FMI CGDB; Appendix A), we present four key contributions: 1) the open-source TwinWeaver framework[1] for serializing clinical histories; 2) the GDT model, built using TwinWeaver, which demonstrates competitive performance relative to advanced time-series and electronic health record (EHR) models; 3) demonstration of zero-shot generalization to clinical trials; and 4) highlighting a reinforcement learning-based interpretable clinical reasoning extension to the base model.

## 2. Related Work

**Clinical Event Prediction**  Clinical event prediction estimates time-to-event outcomes from longitudinal data using classical methods such as Cox Proportional Hazards models (Cox, 1972), Random Survival Forests (RSF) (Ishwaran et al., 2008), and deep survival models (Katzman et al.,

---

[1]http://github.com/MendenLab/TwinWeaver

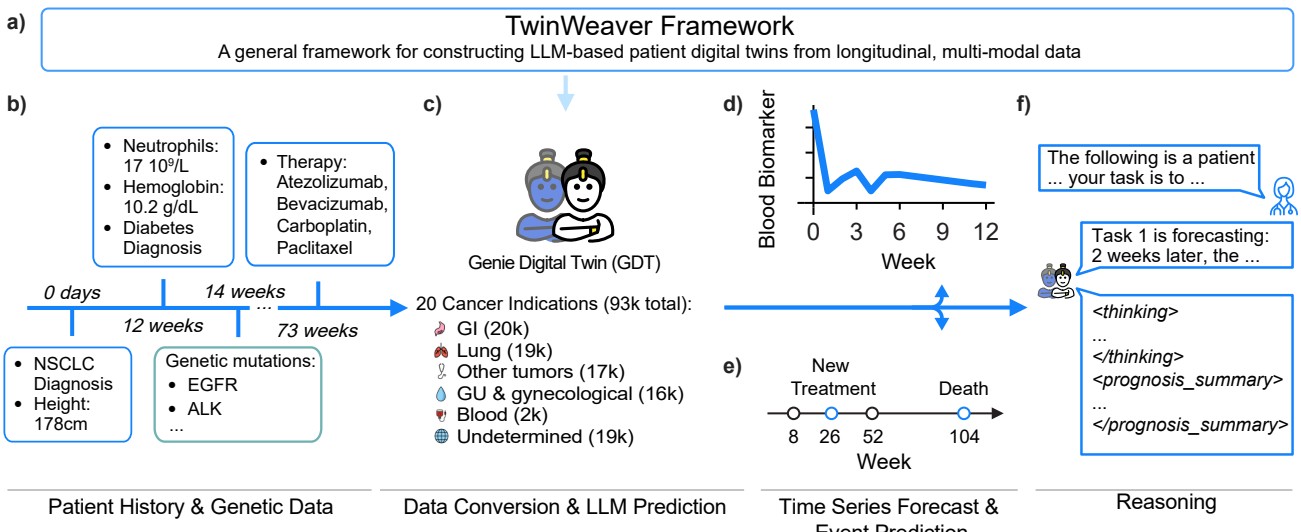

Figure 1. **The TwinWeaver framework serializes longitudinal patient histories into text to train the Genie Digital Twin (GDT) pan-cancer foundation model. a)** The TwinWeaver framework **b)** serializes multi-modal EHR data (history, genetics) into text. **c)** We use this to develop GDT on 93,054 real-world patients across 20 cancer indications (patient counts shown in parentheses). GDT jointly **d)** forecasts continuous biomarkers and **e)** predicts landmark clinical events. **f)** The model can be further trained for interpretable predictions using reasoning-based outputs. NSCLC is non-small cell lung cancer, GI is gastrointestinal, GU is genitourinary.

2018). These approaches often require manual data pre-processing to handle real-world data (RWD) and typically decouple event prediction from trajectory forecasting, limiting information gain. These limitations call for flexible, unified models that jointly capture longitudinal dynamics and censored clinical events.

**Clinical & EHR Foundation Models** Recent foundation models have shifted clinical modeling from static risk prediction toward dynamic trajectory forecasting (Guo et al., 2023). LLM-based approaches such as DT-GPT (Makarov et al., 2025) and EHR2Path (Pellegrini et al., 2025) exemplify this progress, yet they remain constrained by the number of forecastable variables or their temporal scope. EHR foundation models such as CLMBR-T (Wornow et al., 2023a) and MOTOR (Steinberg et al., 2024a) improve event prediction, while SCOPE (Hussain et al., 2024) further enables joint longitudinal and event tasks. Further EHR models have been proposed, including ETHOS (Renc et al., 2024) and Redekop et al. (2025) exploring zero-shot performance, whilst Curiosity (Waxler et al., 2025) and TransformEHR (Yang et al., 2023) demonstrate that such models scale to large datasets. Additionally, HALO generated synthetic EHR (Theodorou et al., 2023), ARES (Renc et al., 2025) extend ETHOS to adaptive risk estimation, with McDermott (2025) developing a generalized EHR code framework.

However, these models rely on fixed, code-based vocabularies that constrains out-of-distribution adaptability (Bedi et al., 2026) and inclusion of further complex modalities,

such as genetic data. Foresight (Kraljevic et al., 2024) adds a natural language processing code extraction module for flexibility, but has limited ability for numerical forecasting. GenHPF (Hur et al., 2023) converts EHRs into text for cross-institutional adaptation, yet remains focused on classifying predefined outcomes rather than generative patient forecasting. These constraints underscore the need for unified forecasting and prediction models for heterogeneous, long-horizon clinical trajectories, capable of generalizing beyond their training distribution.

**Time-Series Forecasting Foundation Models** Deep learning architectures such as TiDE (Das et al., 2023) and foundation models such as Chronos (Ansari et al., 2024; Hoo et al., 2024) have advanced time-series forecasting. Whereas LLMs can be used for forecasting by treating numerical sequences as text (Gruver et al., 2023), their EHR applications remain limited to small variable sets. Moreover, general models such as Chronos capture temporal dependencies but lack medical context and struggle with integrating sparse genetic data. The limitations highlight the need for richer multi-modal temporal representations.

**Reasoning in Clinical AI** Recent work, such as OncoReason (Hemadri et al., 2025), demonstrates the utility of LLMs for complex clinical reasoning and outcome prediction. These advances highlight the opportunity for models that couple numerical forecasting with interpretable textual rationales to improve transparency, trust, and clinical usability.

# 3. TwinWeaver Framework

We introduce TwinWeaver, a framework that enables large language models (LLMs) to perform clinical prediction by jointly forecasting numeric time-series and censored clinical events. TwinWeaver fine-tunes LLMs on procedurally generated text representations of longitudinal patient journeys, providing a flexible representation for multi-modal clinical data. This serialization leverages the tokenization and generative capabilities of LLMs to model irregular clinical timelines and learn shared latent patient states that support both continuous biomarker forecasting and discrete clinical event prediction.

We define our core notation for the model. Let $\mathcal{P}$ be the set of patients, where each $p = (\mathbf{s}_p, \mathcal{H}_p) \in \mathcal{P}$ has static attributes $\mathbf{s}_p$ and an observed history $\mathcal{H}_p = [h_{p1}, ..., h_{pn}]$ where $h_{pi} = (\text{timestamp}, \text{event}, \text{value})$, for a total of $n$ observations. The next-token probability of the LLM is defined as $q_\theta(x_i \mid x_{1:i-1})$ for a given token sequence $x$, with $x_i$ being the $i$-th token, and parameterized by $\theta$.

## 3.1. Training

The training pipeline preprocesses longitudinal data, generates input-target pairs by splitting patient trajectories at critical time points, such as therapy start dates, converts the pairs into text prompts, and fine-tunes a pretrained LLM with them.

For the trajectory splitting for patient $p$, we define the input as the combination of static data and the history up to a split time $t$, $\mathcal{X}_p(t) = (\mathbf{s}_p, \mathcal{H}_p(t))$, with $\mathcal{H}_p(t) = [h_{pi} \mid h_{pi} \in \mathcal{H}_p, h_{pi,\text{timestamp}} \leq t]$. The target $\mathcal{Y}_p(t)$ is defined as a set of future predictions of various tasks.

### 3.1.1. FORECASTING TASK

The first task type is time-series forecasting, used to predict frequently measured values such as blood biomarkers or vital signs.

Managing LLM context size during training and inference requires limiting the number of variables forecast per instance; accordingly, a subset of variables is sampled for each training example to ensure computational feasibility. We denote the set of output variables as $V$, with $V' \subseteq V$ being a sampled subset. This sampling is weighted proportionally to each variable's observation count and volatility (Appendix B.1.1).

The forecasting target $\mathcal{Y}_p^{forecast}(t^{forecast}, V')$ comprises future values for the sampled variables $V'$ up to the forecasting horizon $t^{forecast} = t + \Delta t^{forecast}$ with $\Delta t^{forecast} \in \mathbb{N}^+$. We treat unmeasured values within this window and any values of new competing events (e.g., a new line of therapy) as missing targets (Appendix B.1.2). This subsampling

strategy maintains flexibility across time horizons while ensuring computational feasibility.

### 3.1.2. LANDMARK EVENTS TASK

The second task, predicting a patient's event status at a future time point is a critical clinical task, which we formulate using the landmarking framework (Van Houwelingen, 2007) with a focus on patient risk ranking.

Our data sampling methodology enables the model to predict event status for an arbitrarily sampled future horizon (Appendix B.1.3; Algorithm 1). We define $E \in \mathcal{E}$ as the specific clinical event from all possible events $\mathcal{E}$, e.g. disease progression, and $t^{event} = t + \Delta t^{event}$ as the horizon, based on $\Delta t^{event} \in \mathbb{N}^+$. The set of event labels is $Y = \{\text{occurred}, \text{not occurred}, \text{censored}\}$. The model is thus trained to predict the patient's status of event $E$ at the landmark time $t^{event}$ as one of three classes $y \in Y$.

This formulation explicitly models the observation process and competing events by treating censoring as a distinct target class. Consistent with established real-world data methodologies, we handle treatment switching as a censoring event (Collet et al., 2025), as well as patients who reach the end of their recorded trajectory without the occurrence of death, or who exceed the dataset's global cutoff date.

### 3.1.3. COMBINING, CONVERTING TO TEXT & TRAINING

The TwinWeaver framework transforms the structured patient history $\mathcal{X}_p(t)$ and multi-task targets $\mathcal{Y}_p(t)$ into a single, human-readable text prompt. As outlined in Appendix B.1.4 (Algorithm 2), this serialized prompt includes static data, the chronological visit history, such as blood measurements and biomarkers, a summary of recent critical information, and a numbered list of prediction tasks. To manage context constraints, we truncate intermediate visits while strictly preserving the initial visit to retain baseline diagnostic information. Additionally, to mitigate the 'lost-in-the-middle' phenomenon, we explicitly repeat the most recent clinical observations and genetic events immediately prior to the task definitions. The model is trained using a standard causal language modeling loss, computed only on the target completion tokens.

## 3.2. Inference

For evaluation, inference for forecasting and landmark event prediction is handled separately.

### 3.2.1. FORECASTING

For forecasting inference, we generate $\{\hat{\mathcal{Y}}_p^{\ m}\}_{m=1}^M, M \in \mathbb{N}^+$ independent completions from the model's predictive distribution $q_\theta$ for a given prompt for patient $p$ (Makarov et al., 2025). These $M$ text-based completions are decoded

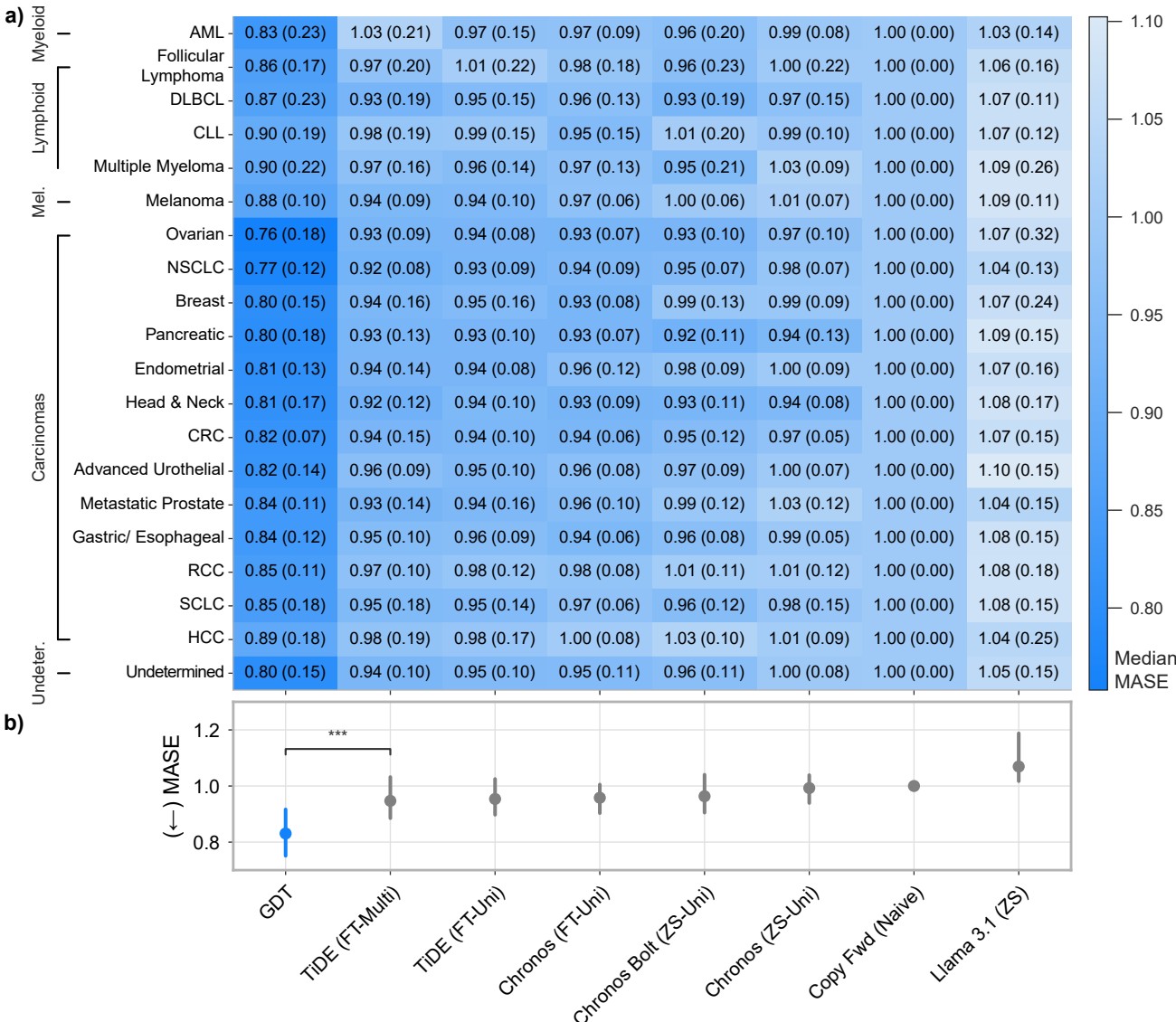

*Figure 2.* **GDT achieves reduced forecasting error in highly dynamic blood biomarker forecasting across the majority of cancer indications. a)** Heatmap of the median Mean Absolute Scaled Error (MASE - lower is better). GDT shows lower error (darker blue) compared to baselines for the top 30 most time-changing variables per indication. **b)** Aggregated performance of the top 30 most changing variables shows GDT achieves a median MASE of 0.830, significantly outperforming the second-best baseline, TiDE multivariate ($p < 0.001$, Wilcoxon signed-rank test). Error bars and numbers in parentheses denote the Interquartile Range (IQR) across variables. ZS denotes zero-shot, FT is fine tuned, Uni is univariate input, Multi is multivariate input, all cancer indication abbreviations are in Appendix C.1.

back into numerical trajectories using deterministic rules. We then average these trajectories for each variable and time point separately to produce a stable mean prediction:
$$\hat{\mathcal{Y}}_p^{\text{forecast}} = \frac{1}{M} \sum_{m=1}^{M} \hat{\mathcal{Y}}_p^m.$$

### 3.2.2. LANDMARK EVENTS

Event probabilities are derived from the model's three-class outputs (occurred, not occurred, censored) using log-likelihoods, which provide continuous scores required for survival analysis evaluation rather than discrete class predic-

tions.

Specifically, we first compute the length-normalized log-likelihood score $L_{y,p}$ for each of the three possible target strings $y \in Y$:

$$L_{y,p} = \frac{1}{|x| - k + 1} \sum_{i=k}^{|x|} \log(q_\theta(x_i \mid x_{1:i-1}))$$

where $k$ is the index of the first token of the target and $x = [\text{prompt}; y]$ is the prompt concatenated with the target string

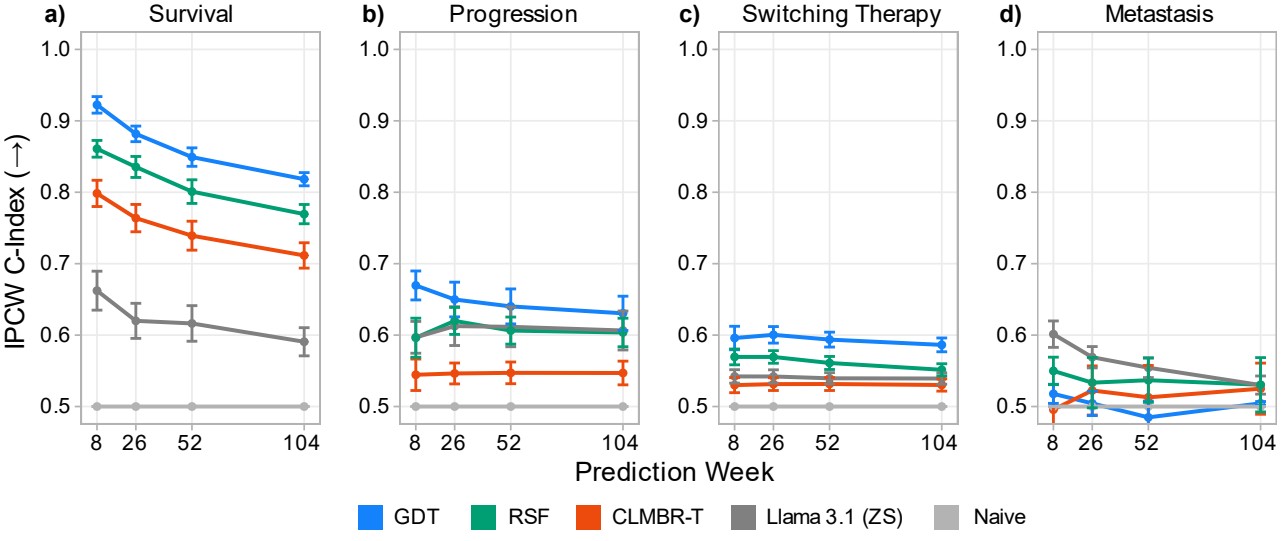

*Figure 3.* **GDT demonstrates improved risk stratification in predicting survival, progression, and switching therapy, while metastasis prediction remains constrained by limited data availability.** We report the mean IPCW C-Index (higher is better) and standard error across 20 indications for **a)** survival, **b)** disease progression, **c)** switching therapy, and **d)** metastasis. GDT demonstrates robust ranking capabilities, outperforming the strongest baselines in 12 of 16 evaluated time points. Lower performance is seen in predicting metastasis outcomes, likely due to low data availability ($N = 3$ indications).

$y$. We then normalize these scores using a softmax function to obtain a probability estimate over the three outcomes:

$$\hat{P}(y \mid \mathcal{X}_p(t), E, \Delta t^{event}) = \frac{\exp(L_{y,p})}{\sum_{y' \in Y} \exp(L_{y',p})}$$

The final ranking score of the event $E$ is then taken from the 'occurred' class probability, and is used for all downstream evaluation metrics:

$$\text{Score}(p, E, \Delta t^{event}) = \\ \hat{P}(y = \text{occurred} \mid \mathcal{X}_p(t), E, \Delta t^{event})$$

Although the formulation models the joint likelihood of event occurrence and observation and lacks guarantees for monotonic cumulative incidence functions, it remains effective for *ranking*-based clinical tasks such as risk stratification and comparative treatment evaluation (Steck et al., 2007).

This approach preserves the LLM architecture and training procedure, enabling direct integration with existing software libraries for unified forecasting, event prediction, and reasoning. However, the output scores should be interpreted only for ranking of risk. We also provide an extension with conditioning without censoring, together with monotonicity guarantees, by applying isotonic regression (Appendix B.2.1).

## 4. Pan-Cancer LLM

As an instance of the TwinWeaver framework, we train Genie Digital Twin (GDT), a pan-cancer LLM, on a large-scale real-world dataset from the FH-FMI CGDB (Appendix A). The cohort comprises 93,054 patients with recorded drug visits across 20 cancer types and is split at the patient level into 82,753 training, 4,991 validation, and 4,999 test patients to ensure robust evaluation.

### 4.1. Data Preprocessing

The pipeline transforms raw multi-modal inputs, including diagnoses, laboratory measurements, and genetic mutation panels, into a structured textual representation of longitudinal patient trajectories at weekly resolution (Appendix C.1.1).

### 4.2. Task Configuration

Training focuses on clinically relevant periods by sampling split times $t$ uniformly from visits within 90 days of a new line of therapy. This strategy allows us to generate multiple training samples from a single patient without overfitting on specific patterns, specifically 10 for GDT, leveraging their full history across different treatments.

GDT is configured to perform two primary prediction tasks simultaneously. First, we set the blood biomarker time-series horizon $\Delta t^{forecast} = 13$ weeks, a horizon highly relevant for monitoring treatment response and toxicity. Second, we predict four key clinical events (survival, metastasis,

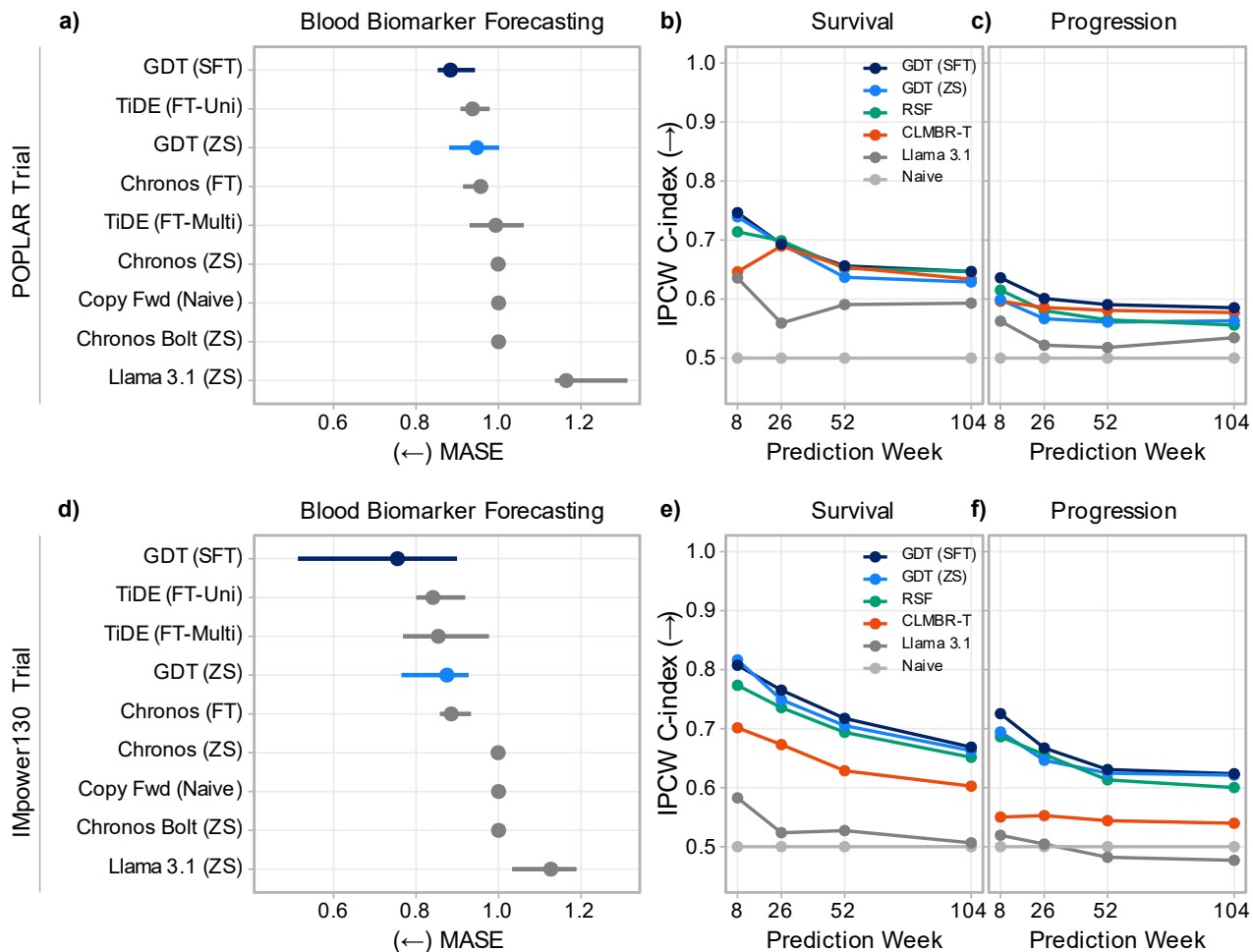

*Figure 4.* **GDT with supervised fine-tuning (SFT) outperforms baselines on out-of-distribution clinical trial tasks.** The model is evaluated on the unseen trials **a-c)** POPLAR and **d-f)** IMpower130. GDT (SFT) achieves the lowest MASE (Mean Absolute Scaled Error - lower is better; IQR error bars across 17 variables) in blood biomarker forecasting (**a, d**) and highest C-Indexes (higher is better) for survival (**b, e**) and progression (**c, f**) events, demonstrating strong generalization in cold start scenarios where historical data is sparse. ZS denotes zero-shot predictions, FT means the model is fine-tuned, whilst Uni is univariate input, Multi is multivariate input, RSF is Random Survival Forest.

disease progression, and therapy switching) at longer horizons sampled uniformly $\Delta t^{event} \in \{1, \dots, 104\}$, enabling flexible predictions across various clinically relevant horizons. Whilst the TwinWeaver framework is model-agnostic, we choose to instantiate GDT using an established open-source model, Llama 3.1 8B Instruct. We train for 1 epoch using the AdamW optimizer, with a learning rate of $10^{-5}$ and a context length of 8000 tokens (Appendix C.2.1).

### 4.3. Reasoning Extension

Using fine-tuned LLMs as a prediction platform enables downstream tasks such as reasoning. We focus on neutrophil forecasting on a subsample of the NSCLC cohort ($N = 2,385$). To address the scarcity of gold-standard clinical rationales, we employ a knowledge distillation approach, prompting a teacher model, Qwen3 Next 80B-A3B (Yang et al., 2025), to synthesize structured chains-of-thought conditioned on patient history and ground-truth outcomes. The student GDT model first undergoes supervised fine-tuning on this synthetic corpus, followed by alignment via Group Relative Policy Optimization (GRPO) (Shao et al., 2024b). Crucially, to ground the generated reasoning in empirical accuracy, we utilize a deterministic reward function defined by the negative Mean Absolute Error (MAE) of the numerical forecast. This optimization objective ensures that the generated rationale serves as a functional intermediate representation aligned with accurate trajectory modeling, rather than merely plausible-sounding text (Appendix C.2.2).

### 4.4. Baselines

We compare GDT against forecasting and event-prediction baselines (Appendix C.3). Forecasting baselines include

copy-forward, TiDE, and the foundation models Chronos and Chronos Bolt (Ansari et al., 2024), which were pretrained on over 700k time series. On the other hand, event prediction baselines include Random Survival Forest (RSF) and the EHR foundation model CLMBR-T, pretrained on 2.5 million patients and adapted with a Cox proportional hazards head. In real-world data experiments, baselines are trained on a representative 2k-patient subset; Appendix D.1 shows that GDT retains its advantage under the same constraint. All clinical trial evaluations use identical datasets.

# 5. Results

We benchmark GDT across 20 cancer indications on real-world clinical data as well as on out-of-distribution clinical trial tasks, evaluating on realistic, challenging settings.

## 5.1. RWD Forecasting Results

GDT consistently outperforms forecasting baselines across all indications. Aggregated across all variables, GDT achieves a median MASE of 0.867 (IQR 0.186), significantly outperforming the next-best model, multivariate TiDE (0.966; $p < 0.001$, Wilcoxon signed-rank test). Other foundation models and baselines, including Chronos and Llama 3.1, exhibit higher error rates (Appendix D.2).

This performance advantage is even more pronounced on clinically critical, highly volatility variables. Evaluated on the top 30 most time-varying biomarkers per indication (Figure 2), GDT achieves a median MASE of 0.830 (IQR 0.165), significantly outperforming multivariate TiDE (0.947; $p < 0.001$). These results demonstrate GDT's effectiveness in modeling dynamic patient trajectories relative to time-series models and general-purpose LLMs. Notably, GDT retains this performance advantage even when trained on the same 2,000-patient subset as the baselines (Appendix D.1).

## 5.2. RWD Event Results

GDT shows statistically significant improvements across the majority of modeled clinical outcomes, specifically those with sufficient indication coverage (Figure 3). For survival, progression, and therapy switching (each with $N > 10$ indications), GDT outperforms the strongest baseline, RSF, across all evaluated time points on Inverse Probability of Censoring Weighting (IPCW) C-index (0.703 vs 0.662), and is statistically significant in 11 of 12 time points ($p_{adj} < 0.05$). This advantage does not extend to time-to-metastasis (IPCW C-index 0.502 vs 0.563; $N = 3$), where limited training diversity likely constrains performance, highlighting a low-data boundary for the proposed architecture.

Interestingly, we observe lower performance on this task

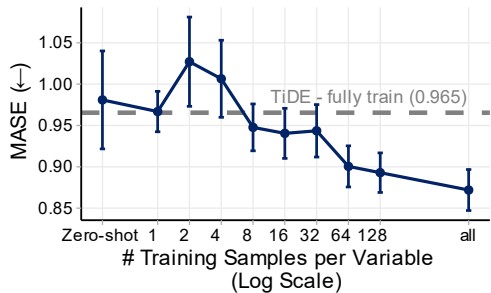

*Figure 5.* **Forecasting error as a function of training sample size.** Evaluated on the test set of the POPLAR trial, using Mean Absolute Scaled Error (MASE), relative to training samples from the OAK trial (log X-axis; error bars denote IQR across 17 variables). Zero-shot performance is competitive, while low-data fine-tuning shows an initial error increase before stabilizing and improving at higher counts ($all \approx 880$/variable).

for the EHR foundation model CLMBR-T despite its larger pretraining cohort and task-specific fine-tuning. This may be attributable to its fixed vocabulary, which covers only 11 of the 283 therapies present in the dataset.

## 5.3. Validation on Clinical Trials

We further evaluate the method on out-of-distribution clinical trial tasks, using two trials for testing (POPLAR, IMpower130) and two additional trials for optional fine-tuning (OAK, IMpower131; Appendix D.3). Clinical trials can be considered out-of-distribution due to differences in data measurement and format, as well as distinct patient populations, in comparison to the RWD training set (Orcutt et al., 2025). The task focuses on cold-start prediction from baseline measurements only, reflecting a challenging and clinically relevant trial setting. Note, the data is identical across all models in the same setup with respect to fine-tuning and zero-shot settings. GDT is evaluated in both zero-shot and fine-tuned settings, with identical data used across all models within each setup (Figure 4).

### 5.3.1. CLINICAL TRIAL BLOOD BIOMARKER FORECASTING

In the cold-start setting, GDT shows strong generalization performance. GDT, fine-tuned on OAK and IMpower131, then evaluated on POPLAR and IMpower130 attains the lowest error rates (median MASE 0.883 and 0.754), significantly outperforming the strongest baseline, univariate TiDE ($p < 0.05$), while zero-shot GDT matches task-specific fine-tuned models (Appendix D.3).

### 5.3.2. CLINICAL TRIAL EVENT PREDICTION

For clinical trial event prediction, fine-tuned GDT shows strong performance, achieving the highest C-index in all

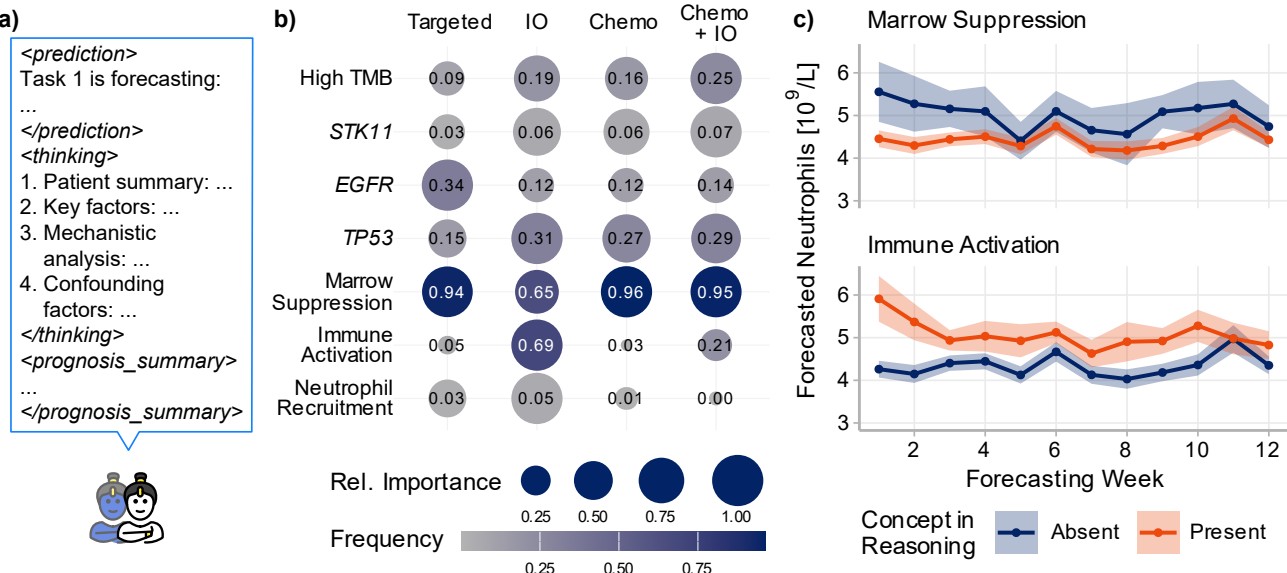

*Figure 6.* **GDT can be extended to provide clinically aligned reasoning. a)** The model generates a structured reasoning chain, including patient summary, key factors, and mechanistic analysis, alongside its numerical forecast. **b)** Keyword analysis shows alignment with clinical expectations, linking marrow suppression to chemotherapy, immune activation to immunotherapy (IO), and *EGFR* mutations to targeted therapy **c)** Stratifying predicted neutrophil trajectories by presence of reasoning concepts confirms validated effects: 'Marrow Suppression' is linked to lower counts, while 'Immune Activation' is linked to higher counts.

evaluation time points, with the zero-shot variant performing comparably high (IPCW C-indexes 0.672 and 0.656; Figure 4). RSF provides a competitive baseline (IPCW C-index 0.652) without using longitudinal trajectories, while CLMBR-T shows variable performance across POPLAR and IMpower130, potentially reflecting vocabulary constraints. Statistical testing was not conducted due to limited sample sizes.

### 5.3.3. SAMPLE EFFICIENCY ANALYSIS

We evaluate the sample efficiency of GDT in an out-of-distribution setting by training on the OAK trial and testing on POPLAR, which share identical treatment regimens (Figure 5). Performance is assessed by progressively subsampling the training data across 17 target variables.

While zero-shot performance is comparable to fully trained TiDE, fine-tuning GDT leads to consistently better performance at 64 samples per variable. Interestingly, the error increases initially, potentially hinting that training on few samples might overfit the model and be worse than the zero-shot setting.

### 5.4. Reasoning Extension

We demonstrate the flexibility of the TwinWeaver framework by extending GDT to generate interpretable clinical reasoning alongside forecasts for neutrophil trajectories in NSCLC (Figure 6; Appendix D.4). Keyword analysis confirms that GDT generated reasoning chains correspond to

clinically relevant features and genetic contexts, correctly associating high tumor mutational burden (TMB) more with immunotherapy-based medications (Shao et al., 2024a), while identifying *EGFR* mutations as key drivers in targeted therapies and delineating distinct neutrophil trajectories based on patients' immune state (Orr et al., 2007; Dunbar et al., 2021). The model additionally identifies *STK11* as a key genetic mutation, which has been found to be associated with effects on neutrophil levels, as well as a prognostic marker (Koyama et al., 2016; Zheng et al., 2024). Although the reasoning extension incurs a modest increase in forecasting error (MASE 0.862 vs 0.828), it offers a calculated trade-off for clinical utility by providing transparent, biologically grounded context for hypothesis generation.

### 5.5. Training Synergy Across Cancer Indications

We find that the pan-cancer training improves the performance of individual cancer indications (Appendix D.5), thus demonstrating the value of training across a set of diseases, particularly for indications that present less data.

### 5.6. Patient History Analysis

We further show that incorporating richer patient histories can enhance clinical trial survival predictions beyond current data collection practices (Appendix D.6). We select patients from the NSCLC test set with at least 52 weeks of history, and systematically cut patient history whilst comparing one-year survival IPCW C-index performance. We observe 9.9%

improvement of the C-index when using the full patient history, in comparison to using baseline information only, which is often done in the context of clinical trials.

## 6. Conclusion

Overall, our findings suggest that the TwinWeaver framework transforms pretrained LLMs using specific fine-tuning into clinical prediction engines, exemplified with GDT. Our benchmarks highlight that, despite standard methods remaining competitive baselines, fine-tuned LLMs overcome critical limitations regarding fixed vocabularies and multimodal integration, thus improving prediction performance over existing methods. Specifically, the ability of GDT to generalize to new clinical trials as well as its pan-cancer capabilities offers a promising solution for data-scarce scenarios.

Future work should focus on systematic hyperparameter optimization, including sampling strategies, effect of weekly aggregation, impact of three sigma filtering and variations in prompt design. Further refinement of time-to-event modeling is needed to produce well-calibrated survival curves while maintaining generative flexibility, as well as to overcome any potential informative censoring bias. Given the high volume of tokens generated by both the text serialization and subsequent simulations of all necessary variables across multiple samples, a critical direction for future research is to evaluate how maximum context length and various truncation approaches affect model performance. Future iterations should also integrate modalities such as histopathology images, single-cell and spatial omics, especially to improve prediction performance in complex biological events such as metastasis. Integrating reasoning directly into model training may enable reasoning-first clinical foundation models, while a further exploration of hybrid numerical tokens could improve forecasting performance. Finally, from a translational perspective, rigorous clinical validation is required to assess real-world utility.

In conclusion, results with GDT demonstrate the effectiveness of the TwinWeaver framework, indicating that LLMs can be adapted successfully as a platform for patient digital twins.

## Acknowledgements

We thank the patients, families, caregivers, and site personnel who participated in the data from which were included in this analysis.

We also thank Pietro Belligoli, Clara Wong-Fannjiang, Maksim Eremeev, Jesse Swanson, Claudia Iriondo and Guy Amster for their feedback and valuable discussions. We would also like to thank Anna Bauer-Mehren, Monika Bug and Luise Wolf for supporting the collaborations, as well as Emilia Leogrande and Andrew Whitehead for enabling the publication.

This study was funded by F. Hoffmann-La Roche and the European Union's Horizon 2020 Research and Innovation Programme (Grant agreement No. 950293–COMBAT-RES). N.M., M.B. and M.P.M are supported by the Helmholtz Association under the joint research school "Munich School for Data Science - MUDS".

## Conflict of Interest Disclosure

N.M. conducted this work as an industrial PhD student at F. Hoffmann-La Roche and has since joined Valinor Industries, Inc. M.B., R.R.E., L.V.V. and F.S. are all employees of F. Hoffmann-La Roche. S.M., J.W., D.M. are employees of Genentech. M.P.M. collaborates and is financially supported by GSK, F. Hoffmann-La Roche, and AstraZeneca. M.P.M. consults Merck Sharp & Dohme and McKinsey & Company.

## Impact Statement

This work advances precision oncology by enabling interpretable Patient Digital Twins via Large Language Models. By providing transparent reasoning alongside forecasts, TwinWeaver fosters clinical trust and offers a scalable solution for data-scarce scenarios in drug development. However, applying generative models to healthcare entails risks regarding hallucination and the propagation of biases inherent in Real-World Data. We emphasize that this framework is currently a research tool, rigorous prospective validation is an essential prerequisite before clinical deployment.

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

# A. Data Sources

## A.1. Flatiron Health-Foundation Medicine Clinico-Genomic Database

This study used the US-based deidentified Flatiron Health-Foundation Medicine Breast Cancer, Colorectal Cancer, Pancreatic Cancer, Ovarian Cancer, Metastatic Prostate Cancer, Gastric and Esophageal Cancer, Endometrial Cancer, Melanoma, Advanced Urothelial Carcinoma, Renal Cell Carcinoma, Head and Neck Cancer, Small Cell Lung Cancer, Multiple Myeloma, Hepatocellular Carcinoma, Acute Myeloid Leukemia, Chronic Lymphocytic Leukemia, Diffuse Large B-cell Lymphoma, and Follicular Lymphoma Clinico-Genomic Database (FH-FMI CGDB).

Clinical data from the Flatiron Health Research Database (Flatiron Health, 2025) are linked to genomic data, derived from FMI's comprehensive genomic profiling (CGP) tests (FoundationOne®CDx, FoundationOne®, FoundationOne®Liquid CDx, FoundationOne®Liquid and/or FoundationOne®Heme), in the FH-FMI CGDB by deterministic matching, providing a deidentified dataset (Singal et al., 2019).

The FH-FMI CGDB data that support the findings of this study were originated by and are the property of Flatiron Health, Inc. and Foundation Medicine, Inc. Requests for data sharing by license or by permission for the specific purpose of replicating results in this manuscript can be submitted to PublicationsDataAccess@flatiron.com and cgdb-fmi@flatiron.com.

## A.2. Clinical Trial Data

Qualified researchers may request access to individual clinical trial patient-level data for this analysis through a data request platform. At the time of writing, this request platform is https://vivli.org/ourmember/roche/. For up to date details on Roche's Global Policy on the Sharing of Clinical Information and how to request access to related clinical study documents, see here: https://go.roche.com/data_sharing. Anonymized records for individual patients across more than one data source external to Roche cannot, and should not, be linked due to a potential increase in risk of patient re-identification. For the clinical trial cohort, 4 clinical studies sponsored by Roche/Genentech were identified with the appropriate informed consent for secondary data use analysis. The analysis was conducted according to Good Clinical Practice and the Declaration of Helsinki.

CONSENT FOR PUBLICATION

This manuscript is sufficiently anonymized and does not contain any personal and/or medical information about an identifiable patient.

# B. Methodological Details

This section details the TwinWeaver framework, the specific GDT implementation details, and the inference post-processing methods. We note that TwinWeaver supports arbitrary split times and this choice is task-specific.

## B.1. TwinWeaver Framework Details

### B.1.1. FORECASTING VARIABLE SAMPLING

The probability $p(v)$ of sampling a variable $v \in V$ for the forecasting task is set proportionally to its count and its volatility. We define volatility using the root mean squared error (RMSE) of a simple "copy-forward" baseline, which predicts the next value will be the same as the last observed value. We filter variables to include only those which have at least a certain minimum number of observations, in the case of GDT, is set to 50.

Let $\mathcal{S}(v) = \bigcup_{p \in \mathcal{P}} \{(\mathbf{m}_p(v, \tau_i), \mathbf{m}_p(v, \tau_{i+1}))\}$ be the set of all consecutive value pairs $(y, y')$ for $v$ (where $\tau_{i+1}$ is the next observation time after $\tau_i$ for patient $p$) across all patients. Let $N_v = |\mathcal{S}(v)|$. The copy-forward RMSE is:

$$RMSE_v = \sqrt{\frac{1}{N_v} \sum_{(y,y') \in \mathcal{S}(v)} (y' - y)^2}$$

Let $\sigma_v$ be the standard deviation of all observed values of $v$. We define the normalized RMSE (NRMSE) as:

$$NRMSE_v = \frac{RMSE_v}{\sigma_v}$$

Let $C_v$ be the total number of observations for variable $v$ in the dataset. The unnormalized sampling score $s_v$ is defined as:

$$s_v = \log_2(C_v \times NRMSE_v)$$

The final sampling probability $p(v)$ is obtained by normalizing these scores over all candidate variables in $V$. This strategy biases sampling towards variables that are frequently measured ($C_v$) and exhibit high, non-trivial variance over time ($NRMSE_v$), which are often the most clinically relevant and challenging to predict.

### B.1.2. FORECASTING DETAILS

The forecasting target $\mathcal{Y}^{\text{forecast}}$ is defined as the set of lists of future values for the sampled variables $V'$. We only include target values up to the forecast horizon and censor the prediction window at the start of any new competing event, such as a new line of therapy, which could fundamentally alter patient trajectory. Formally, the target is:

$$\mathcal{Y}_p^{forecast}(t, V') = \{[\mathbf{m}_p(v, t + k) \mid k \in \{1, \ldots, \Delta t^{forecast}\},$$
$$t + k < \tau_{censor}(p, t)] \mid v \in V'\}$$

where $\mathbf{m}_p(v, \tau)$ is the value of lab $v$ at week $\tau$, $\tau_{censor}(p, t)$ is the time of the next competing event after $t$, and unmeasured values are treated as missing targets.

**Additional Forecasting Task** To ensure the model remains relevant across a multitude of tasks, we also include a forecasting task for binned values. In this setup, the task is to predict which of 5 equally numbered quintiles the variable's value lands in across the future time steps. However, due to its relevance and scope, we do not further evaluate this task in this paper, and it should be explored in future work.

### B.1.3. LANDMARK EVENTS DETAILS

The uniform sampling of the horizon $\Delta t$ acts as a form of data augmentation, training the model to generalize to any on-demand prognostic query at inference time. Although computationally feasible, this sampling approach prevents overfitting to canonical time points (e.g., 1-year survival) and encourages a more robust understanding of time-aware risk.

---

**Algorithm 1** TwinWeaver: Landmark Event Sampling Strategy

---

**Require:** Patient $p$, history $\mathcal{X}_p(t)$ up to split time $t$, event set $\mathcal{E}$, horizon range $h$
**Ensure:** Training tuple $(\mathcal{X}_p(t), E, \Delta t^{event}, y)$ with $y \in Y$
 1: Randomly sample an event $E \in \mathcal{E}$ (e.g., disease progression).
 2: Sample a horizon $\Delta t^{event} \sim \text{Uniform}[1, h]$.
 3: Set $t^{event} \leftarrow t + \Delta t^{event}$.
 4: Determine whether $t^{event}$ is censored (e.g., by new therapy) and whether $E$ occurred by $t^{event}$:

  - If a censoring event occurs in $(t, t^{event}]$ before occurrence of event $E$, set $y \leftarrow$ censored.

  - Else if $E$ occurs by $t^{event}$, set $y \leftarrow$ occurred.

  - Else set $y \leftarrow$ not occurred.

 5: **return** $(\mathcal{X}_p(t), E, \Delta t^{event}, y)$.

---

We also recognize that a uniform sampling strategy could be questioned, as clinical event hazards are rarely uniform. This mismatch could introduce a bias, potentially leading the model to be poorly calibrated at extreme horizons. Given the large capacity of the LLM and the millions of training samples, we hypothesize that this bias is minimized, but we explicitly evaluate this by testing on stratified, predetermined landmark times.

B.1.4. COMBINING PROMPT

---

**Algorithm 2** TwinWeaver: Prompt Construction for Supervised Fine-Tuning

---

**Require:** Split date $t$, patient history $\mathcal{X}_p(t)$ up to split time $t$, forecasting variables $V' \subseteq V$, targets $\mathcal{Y}_p^{forecast}(t)$ containing
  forecasting targets $\mathcal{Y}_p^{forecast}(t^{forecast}, V')$, at least one event setup $\{(E_j, \Delta t_j, y_j)\}_j$.
**Ensure:** Input string $S_{\text{prompt}}$ and target completion $S_{\text{target}}$
 1: Encode a fixed starting prompt $\pi_0$ into $S_{\text{prompt}}$.
 2: Encode the patient's constant data ($\mathbf{s}_p$; e.g., demographics).
 3: Encode the history $\mathcal{H}_p(t)$ visit-by-visit in chronological order; alphabetically sort items within each visit. All missing values are skipped. If total size of prompt exceeds context length, we omit visits until the total prompt is less than the context length, while ensuring to include the first visit, which often contains important initial diagnoses.
 4: Repeat the latest critical observations (e.g. genetic observations, the latest line of therapy, and the last observed values of all $v \in V'$), to avoid the lost in the middle problem.
 5: Encode all tasks in text and assign each a numeric task ID $j = 1, \ldots, |\mathcal{T}|$.
 6: Construct $S_{\text{target}}$ by encoding the corresponding task answers in the same order and required format, skipping missing data.
 7: **return** $(S_{\text{prompt}}, S_{\text{target}})$.

---

In Algorithm 2, we see how the final full prompt is constructed. Note, that whilst the approach is limited by context length, we anticipate that this will become less of an issue as the current large, state of the art models can process over 100k tokens.

**B.2. Inference Post-Processing**

B.2.1. CONSTRUCTING MONOTONIC CUMULATIVE INCIDENCE FUNCTION (ISOTONIC REGRESSION)

Whereas the landmarking approach described in Section 3.2.2 provides robust risk estimates for specific horizons, treating each horizon $\Delta t$ as an independent query can result in temporal inconsistencies. To recover a consistent Cumulative Incidence Function (CIF) $F(t)$ for individual patient analysis, we employ a two-step post-processing strategy: *conditioning* followed by *isotonic calibration*.

**Step 1: Conditioning.** We first isolate the risk signal from the censoring distribution. Under the assumption of non-informative censoring, the probability of the event occurring by time $t$, given that the observation was not censored before $t$, is a consistent estimator of the true cumulative incidence. We compute this conditional probability $\hat{y}_i$ for each horizon $\tau_i$ by

renormalizing the model's softmax outputs over the known outcomes:

$$\hat{y}_i = \frac{P(\text{occurred} \mid \tau_i)}{P(\text{occurred} \mid \tau_i) + P(\text{not occurred} \mid \tau_i)}$$

**Step 2: Isotonic Calibration.** As the independent estimates $\hat{y}_i$ are not guaranteed to be monotonic, we solve for the calibrated sequence $y_1^*, \ldots, y_K^*$ for a set of $K$ landmark times that minimizes the squared error subject to the monotonicity constraint (Cumulative Incidence cannot decrease):

$$\min_{y^*} \sum_{i=1}^{K} (y_i^* - \hat{y}_i)^2 \quad \text{subject to} \quad y_i^* \leq y_{i+1}^* \quad \forall i$$

We implement this using the Pool Adjacent Violators Algorithm (PAVA). This effectively projects the noisy LLM predictions onto the space of valid probability distribution functions, ensuring clinical interpretability without retraining the model.

# C. Experimental Setup

## C.1. Dataset Statistics

*Table 1.* Dataset Statistics: Patient Counts, Biomarkers, and Prediction Capabilities by Indication

| Indication | Abbr. | Patients | For. Biomarkers | Event Types Predicted | | | |
|---|---|---|---|---|---|---|---|
| | | | | Surv. | Prog. | Switch | Met. |
| Undetermined/multiple | Und. | 19,339 | 90 | ✓ | | ✓ | |
| Non-Small Cell Lung Cancer | NSCLC | 17,834 | 74 | ✓ | ✓ | ✓ | ✓ |
| Breast Cancer | BC | 13,901 | 87 | ✓ | ✓ | ✓ | ✓ |
| Colorectal Cancer | CRC | 11,085 | 80 | ✓ | ✓ | ✓ | |
| Ovarian Cancer | OC | 5,423 | 82 | ✓ | ✓ | ✓ | |
| Metastatic Prostate Cancer | mPC | 5,251 | 78 | ✓ | | ✓ | |
| Pancreatic Cancer | Panc | 4,506 | 69 | ✓ | | ✓ | |
| Gastric / Esophageal Cancer | GE | 3,826 | 67 | ✓ | ✓ | ✓ | |
| Endometrial Cancer | EC | 2,335 | 78 | ✓ | | ✓ | |
| Advanced Urothelial Carcinoma | aUC | 2,071 | 62 | ✓ | ✓ | ✓ | |
| Melanoma | MEL | 1,951 | 72 | ✓ | ✓ | ✓ | ✓ |
| Renal Cell Carcinoma (Kidney) | RCC | 1,385 | 67 | ✓ | ✓ | ✓ | |
| Head and Neck Squamous Cell | HNSCC | 934 | 55 | ✓ | | ✓ | |
| Small Cell Lung Cancer | SCLC | 852 | 53 | ✓ | ✓ | ✓ | |
| Multiple Myeloma | MM | 564 | 87 | ✓ | ✓ | ✓ | |
| Acute Myeloid Leukemia | AML | 504 | 67 | ✓ | | ✓ | |
| Hepatocellular Carcinoma (Liver) | HCC | 405 | 50 | ✓ | ✓ | ✓ | |
| Diffuse Large B-Cell Lymphoma | DLBCL | 350 | 57 | ✓ | | ✓ | |
| Chronic Lymphocytic Leukemia | CLL | 337 | 55 | ✓ | | ✓ | |
| Follicular Lymphoma | FL | 201 | 50 | ✓ | | ✓ | |

*Note: For. = Forecasted, Surv. = Survival, Prog. = Progression, Switch = Switching Treatments, Met. = Metastasis.*

### C.1.1. DATA PREPROCESSING DETAILS

Our preprocessing pipeline converts raw multi-modal data such as diagnoses, genetic tests, labs, etc., into a weekly longitudinal format. Importantly, we encode the patient's genetic information as text as well, noting somatic variants identified from targeted sequencing, including details of the specific mutation.

We include tables on diagnoses, drug administrations, ECOG scores, progression, mortality, response, metastases, surgery, labs, vitals, demographics, cancer-specific biomarkers, extracted genetic signatures, basic biomarkers, and paneled genetic data. While the raw datasets are similar, each required specific adjustments for tables and columns. We aggregate all events on a weekly level to align with the typical frequency of cancer therapy events and avoid over-representing acute hospital visits. Numeric and categorical variables are aggregated using their mean and mode, respectively.

To reduce the issue of outliers in the forecasting task, we apply 3-sigma filtering to all target blood lab values during preprocessing (Pukelsheim, 1994). When encoding numerical values into strings, we limit the number of post-decimal digits to 2 to prevent single long numbers from dominating the string. Note that a small number of patients ($\approx 311$) that were present in multiple indication datasets were discarded from the test and validation sets.

## C.2. GDT Model & Training Details

### C.2.1. TRAINING DETAILS

We limit the total context length to 8000 tokens to manage GPU memory constraints, truncating the earliest visits if necessary while always preserving the first.

For training, we generate $L$ number of splits per patient per line of therapy. We experimented with $L \in \{1, 10\}$ and observed a lower validation loss with $L = 10$. This higher sampling rate per patient increases the total sample count to 2.49 million and provides the model with more diverse examples from each patient's trajectory. In future work with more computational resources, higher sampling rates can also be explored. For outlier filtering, we also apply 3-sigma filtering, based on the per-variable statistics using the training dataset, following (Makarov et al., 2025).

For all smaller fine-tuning runs, we also explore using the cosine learning rate scheduler as well as a 10% warm up ratio, determined then based on the validation loss of the respective run.

For the fine-tuning, formally, let the full text sequence be $x = (x_1, \ldots, x_N)$, where $x_{\text{prompt}} = (x_1, \ldots, x_{k-1})$ and

*Table 2.* Patient demographics and visit statistics by indication. Drug visits is the number of visits where a drug was administered for a given patient.

| Indication | Birth Year (mean) | Gender (% Female) | Visits (mean) | Drug Visits (mean) |
|---|---|---|---|---|
| AML | 1954.1 | 42.9 | 132.6 | 37.1 |
| aUC | 1949.2 | 28.8 | 67.8 | 21.2 |
| BC | 1958.5 | 98.9 | 94.7 | 32.6 |
| CLL | 1951.9 | 38.0 | 116.9 | 18.4 |
| CRC | 1957.8 | 45.7 | 92.8 | 29.1 |
| DLBCL | 1955.4 | 39.7 | 89.0 | 17.3 |
| EC | 1952.7 | 100.0 | 79.6 | 20.6 |
| FL | 1955.4 | 45.3 | 98.6 | 24.7 |
| GE | 1954.3 | 26.7 | 66.7 | 20.1 |
| HCC | 1953.4 | 24.9 | 54.2 | 9.9 |
| HNSCC | 1955.3 | 21.0 | 83.1 | 24.0 |
| MEL | 1954.7 | 34.6 | 61.2 | 17.5 |
| mPC | 1948.6 | 0.0 | 65.2 | 12.0 |
| MM | 1953.3 | 42.6 | 145.9 | 66.9 |
| NSCLC | 1950.8 | 51.1 | 62.3 | 18.1 |
| OC | 1954.3 | 100.0 | 100.3 | 29.1 |
| Panc | 1953.0 | 45.5 | 68.7 | 18.6 |
| Undetermined | 1956.7 | 50.1 | 73.3 | 19.2 |
| RCC | 1955.9 | 28.5 | 73.2 | 20.1 |
| SCLC | 1952.5 | 51.2 | 68.3 | 27.2 |

$x_{\text{target}} = (x_k, \ldots, x_N)$. The loss function is:

$$\mathcal{L}(\theta) = -\sum_{i=k}^{N} \log q_\theta(x_i \mid x_{1:i-1})$$

where $q_\theta$ is the LLM parameterized by $\theta$.

The core training run was performed across 8 H100 GPUs, taking around 7 days, corresponding to approximately 1,344 GPU hours, with full fine-tuning. The per-device batch size was 1, resulting in an effective batch size of 8.

### C.2.2. REASONING EXTENSION METHOD

Using fine-tuned LLMs as a prediction platform enables a variety of downstream tasks, including further fine-tuning to enable reasoning. Understanding the reasons behind a certain prediction is a critical task in biomedicine. In our example, we focus on the largest single indication NSCLC. We chose to focus on forecasting the development of neutrophil counts in the peripheral blood owing to their high measurement frequency and variability. We demonstrate this capability by extracting 3000 random patients from the NSCLC train dataset. We also extract 300 validation and test set patients. For all patients we then generate the training data as outlined above. After dropping all patients without enough neutrophil observations, we obtained 2385 train, 235 validation and 244 test set patients.

For the reasoning, we use a setting, in which we first generate the prediction and then provide the potential reasoning. As in most cases the literature to generate a truly logical step-by-step approach is limiting, we consider this step a hypothesis-generating approach. Additionally, in this setting, it is possible to provide a specific prediction and generate multiple possible explanations, an important tool to provide different hypotheses for a patient's response in clinical practice.

We first generated a reasoning dataset by prompting the Qwen3 Next 80B-A3B model to complete a reasoning chain given a patient's history and the true targets. We then applied deterministic basic filtering (e.g. formatting).

We then trained GDT on this dataset for the initial predictions. Finally, to improve reasoning and prediction performance, we then applied GRPO (Shao et al., 2024b) to the fine-tuned reasoning model. The GRPO reward function used only the

mean absolute error, with a lower value leading to a higher reward, and capped for errors above 20 units. $\beta = 0$, the learning rate was set to $10^{-6}$, using 8 generations, 0.1 warm up ratio, 1 epoch.

### C.3. Baselines & Metrics

#### C.3.1. FORECASTING BASELINES

We compare GDT with standard time-series models on the blood biomarker forecasting task.

To ensure fair comparisons, we performed extensive post-processing of the data for the baseline models. For the missingness, we performed linear interpolation together with forward and backward filling for the input time series, and forward filling for the target variable. Since potentially other lab variables are relevant for the prediction, we include other historical time series of all lab variables overlapping across all cancer types. This results in 74 lab variables for the real-world pan-cancer dataset and 43 variables in the clinical trials dataset. We use these variables as inputs in the multivariate capable models, making sure there is no temporal data leakage. We additionally provide a static feature vector with additional variables, using their last observed value if needed, including age, gender, number of diagnoses, number of genetic events, therapy name, therapy number and key blood biomarkers. We apply a 30-week look-back window, and 13-week prediction window, ensuring that the model sees over half a year of patient data. More historic blood lab value information further than 30 weeks into the past is most likely not producing additional information relevant for blood biomarker prediction. As in GDT, we also apply 3-sigma filtering to the data to remove outliers.

Evaluating baselines on the full pan-cancer dataset was constrained by the heterogeneity of the data structure (variable lengths, missingness patterns) which required extensive preprocessing for non-LLM architectures. We therefore utilized a random subsample, 2000 training patients for the baselines, resulting in 45,023 training samples with multiple lines of therapies and variables per sample. Note, in the clinical trial tasks, all fine-tuned models used identical training datasets.

To ensure a fair comparison, we then used an identical number of training patients and set up on the clinical trial validation data sets as for the GDT model, see Appendix D.1 for details. However, GDT still has improved performance.

TiDE, Chronos and Chronos Bolt are all implemented in AutoGluon (Erickson et al., 2020). For TiDE, we explored both, including and excluding all helper columns and a hyperparameter search for the learning rate. TiDE also processes the static features. Chronos and Chronos Bolt use only the target variable's history as input, since the models cannot process multiple input time series and we observed a lower validation loss when excluding the static features vector. We believe this is due to the implementation in AutoGluon, which requires an external regressor. For Chronos fine-tuning, we used the default learning rate of $10^{-3}$ and performed early stopping.

For the clinical trial cold start task, where only baseline measurements were available, we constructed a synthetic history for the time-series baselines (TiDE, Chronos) to satisfy their input window requirements. We backfilled the baseline value across the 30-week look-back window. This provided the models with a valid input tensor representing a constant history. This setup specifically evaluates the models' ability to infer future trajectory dynamics based on static covariates and baseline magnitude, rather than historical volatility.

**Metrics** We evaluate forecasting performance primarily using the Mean Absolute Scaled Error (MASE), with the copy forward as the naive predictor. MASE is particularly suitable for multi-modal clinical datasets where biomarker scales differ by orders of magnitude (e.g., neutrophils vs. albumin). We compute the metric by first aggregating the error across all patient samples for a given variable and horizon, and then taking the ratio against the aggregated copy forward (naive) baseline error. A MASE $< 1$ indicates that the model provides superior forecasts to simply carrying forward the last known value. We formally define the aggregated MASE for a specific variable $v$ as:

$$\text{MASE}_v = \frac{\sum_{p \in \mathcal{P}} \sum_{j=1}^{J} |y_{p,j} - \hat{y}_{p,j}|}{\sum_{p \in \mathcal{P}} \sum_{j=1}^{J} |y_{p,j} - y_{p,\text{last}}|}$$

where $p$ indexes the patients, $J$ is the number of forecasting horizons (up to 13 weeks in this case), $y_{p,j}$ is the true value, $\hat{y}_{p,j}$ is the predicted value, and $y_{p,\text{last}}$ is the last observed value before the split time $t$. We report the median of these $\text{MASE}_v$ scores across all variables and indications to ensure robustness against outliers common in real-world EHR data. We apply 3-sigma capping to minimize the effect of outliers.

Note, we evaluate forecasting performance using a population-aggregated variant of the standard Mean Absolute Scaled Error (MASE). Standard MASE (Hyndman & Koehler, 2006) typically scales the error of each individual time series by its own in-sample historical volatility. However, in the context of sparse and irregular clinical data, patient histories are often short or exhibit periods of stability, leading to near-zero denominators and numerical instability when calculating per-patient ratios.

Since time-varying variables are the most relevant in clinical settings, we define the 30 most time varying variables, based on the mean absolute percentage error of the copy-forward baseline.

### C.3.2. EVENTS BASELINES

We use the Random Survival Forest as a strong standard machine learning survival baseline focusing on the last observed values. For Random Survival Forest, we take the last observed values of critical patient parameters, representing a standard survival model. Specifically, it includes the following parameters as input: Indication, Gender, Events Until Split, Age, Number of Previous Diagnoses, Number of Genetic Events, Values from ROPRO Constants (Becker et al., 2020), Therapy Name, Therapy Line Number, Weight, Height, Oxygen Saturation, Systolic Blood Pressure, ECOG, Hemoglobin, Urea Nitrogen, Platelets, Calcium, Glucose, Lymphocytes Percent, Alkaline Phosphatase, Total Protein, Alanine Aminotransferase, Albumin, Total Bilirubin, Chloride, Monocytes Number, Eosinophils Percent, and Lactate Dehydrogenase. We use the 'scikit-survival' library, encoding all categorical variables using a one hot encoder. Hyperparameter selection followed a bifurcated approach: for the CGDB task, we maintained default settings to ensure computational efficiency, while for the clinical trials task, we conducted a grid search. Notably, the optimal configurations identified via search converged closely with the default values and performance (0.652 vs 0.648), suggesting the robustness of the initial baseline.

To compare with state of the art EHR foundation models, we employed CLMBR-T. Using CLMBR-T required extensive preprocessing of the data, as the vocabulary of the model is fixed. We transformed all diagnoses, medications, LOINC and demographic data into the MEDS-OMOP (Medical Event Data Standard - Observational Medical Outcomes Partnership) format. We then used CLMBR-T through the femr package (Steinberg et al., 2024b) to generate the patient embeddings, and then trained a Cox's proportional hazard's model with elastic net penalty from scikit-survival. We applied a grid search with cross validation to find the optimal hyperparameters.

Following the forecasting baselines, we use a 2000 patient subset for training on the pan-cancer dataset, and the full training set for the clinical trials.

**Metrics** For the landmark event tasks, we assess the model's ability to correctly rank patients by risk using the Inverse Probability of Censoring Weighting (IPCW) Concordance Index (C-index)(Uno et al., 2011). The standard C-index is biased in the presence of right-censored data; IPCW corrects for this by weighting patient pairs by the inverse probability of remaining in the study, estimated via the Kaplan-Meier estimator of the censoring distribution. Formally, for a pair of patients $i$ and $j$, it measures the probability that the patient with the higher predicted risk score experiences the event first:

$$\text{C-index} = \frac{\sum_{i,j} \mathbb{I}(T_i < T_j) \cdot \mathbb{I}(\hat{R}_i > \hat{R}_j) \cdot W_{ij}}{\sum_{i,j} \mathbb{I}(T_i < T_j) \cdot W_{ij}}$$

where $T$ is the event time, $\hat{R}$ is the predicted risk score (probability of event occurrence), and $W_{ij}$ are the censoring weights. A C-index of 0.5 implies random ranking, while 1.0 implies perfect discrimination. To further evaluate the reliability of the model's confidence, we additionally compute the Brier Score, which measures the mean squared difference between the predicted probabilities and the actual outcomes (Graf et al., 1999). Lower Brier scores indicate better calibration. The Brier Score results on the adjusted monotonic versions are detailed in Appendix B.2.1.

### C.4. Evaluation Inference Details

For the forecasting inference on the pan-cancer test set, we set $L = 3$, ensuring that over 90% of variables are evaluated in every indication, whilst still keeping the computational costs low.

Note, we observed that GDT outputs consistently, achieving a 0% formatting and syntax errors across all indications in the real-world dataset.

## C.4.1. EVALUATED VARIABLES (LOGICAL OBSERVATION IDENTIFIERS NAMES AND CODES - LOINC CODES)

*Table 3.* Evaluated forecasting variables by LOINC code across indications.

| LOINC | AML | BC | CLL | CRC | DLBCL | EC | FL | GE | HCC | HNSCC | MEL | MM | NSCLC | OC | Panc | RCC | SCLC | Undet. | aUC | mPC |
|---|---|---|---|---|---|---|---|---|---|---|---|---|---|---|---|---|---|---|---|---|
| 10334-1 | | | | | | | | | | | | | | ✓ | | | | ✓ | | |
| 11051-0 | | | | | | | | | | | | ✓ | | | | | | | | |
| 13992-3 | | | | | | | | | | | | ✓ | | | | | | | | |
| 14804-9 | | | | | ✓ | | | ✓ | | | ✓ | ✓ | | ✓ | | ✓ | | ✓ | | |
| 14979-9 | ✓ | ✓ | | ✓ | ✓ | ✓ | ✓ | ✓ | | ✓ | ✓ | ✓ | ✓ | ✓ | ✓ | ✓ | | ✓ | ✓ | ✓ |
| 15152-2 | | | | | | ✓ | | | | | | ✓ | ✓ | | | | | ✓ | | |
| 1742-6 | ✓ | ✓ | ✓ | ✓ | ✓ | ✓ | ✓ | ✓ | ✓ | ✓ | ✓ | ✓ | ✓ | ✓ | ✓ | ✓ | ✓ | ✓ | ✓ | ✓ |
| 1751-7 | ✓ | ✓ | ✓ | ✓ | ✓ | ✓ | ✓ | ✓ | ✓ | ✓ | ✓ | ✓ | ✓ | ✓ | ✓ | ✓ | ✓ | ✓ | ✓ | ✓ |
| 17817-8 | | | | | | | | | | | | ✓ | | | | | | | | |
| 17842-6 | | ✓ | | | | | | | | | | | | | | | | ✓ | | |
| 17861-6 | ✓ | ✓ | ✓ | ✓ | ✓ | ✓ | ✓ | ✓ | ✓ | ✓ | ✓ | ✓ | ✓ | ✓ | ✓ | ✓ | ✓ | ✓ | ✓ | ✓ |
| 18262-6 | | ✓ | | ✓ | | ✓ | | | | | ✓ | | | | ✓ | ✓ | | | | |
| 1834-1 | | | | | | | | ✓ | | | | | | | | | | | | |
| 19023-1 | ✓ | ✓ | ✓ | ✓ | | ✓ | ✓ | ✓ | | | ✓ | ✓ | ✓ | ✓ | ✓ | ✓ | | ✓ | ✓ | ✓ |
| 19123-9 | ✓ | ✓ | ✓ | ✓ | ✓ | ✓ | ✓ | ✓ | ✓ | ✓ | ✓ | ✓ | ✓ | ✓ | ✓ | ✓ | | ✓ | ✓ | ✓ |
| 1920-8 | ✓ | ✓ | ✓ | ✓ | ✓ | ✓ | ✓ | ✓ | ✓ | ✓ | ✓ | ✓ | ✓ | ✓ | ✓ | ✓ | ✓ | ✓ | ✓ | ✓ |
| 1952-1 | | | | | | | | | | | | ✓ | | | | | | ✓ | | |
| 1963-8 | | | | ✓ | | ✓ | | | | | | | | ✓ | | | | ✓ | | |
| 1968-7 | | ✓ | | | ✓ | | | ✓ | | ✓ | ✓ | ✓ | | ✓ | ✓ | ✓ | ✓ | ✓ | | ✓ | ✓ |
| 1971-1 | | ✓ | | | ✓ | ✓ | | | | ✓ | ✓ | ✓ | ✓ | ✓ | | ✓ | | ✓ | | ✓ |
| 1975-2 | ✓ | ✓ | ✓ | ✓ | ✓ | ✓ | ✓ | ✓ | ✓ | ✓ | ✓ | ✓ | ✓ | ✓ | ✓ | ✓ | ✓ | ✓ | ✓ | ✓ |
| 1988-5 | | | | | ✓ | | | | | | ✓ | ✓ | | | ✓ | | | ✓ | | |
| 1994-3 | | | | | | | | | | | | | | | | | | ✓ | | |
| 2028-9 | ✓ | ✓ | ✓ | ✓ | ✓ | ✓ | ✓ | ✓ | ✓ | ✓ | ✓ | ✓ | ✓ | ✓ | ✓ | ✓ | ✓ | ✓ | ✓ | ✓ |
| 2039-6 | | ✓ | | ✓ | | | | ✓ | | | | | ✓ | ✓ | ✓ | | | ✓ | | ✓ |
| 20482-6 | ✓ | ✓ | | ✓ | | ✓ | ✓ | | | | ✓ | | ✓ | ✓ | ✓ | ✓ | | ✓ | | ✓ |
| 20570-8 | ✓ | ✓ | ✓ | ✓ | ✓ | ✓ | ✓ | ✓ | ✓ | ✓ | ✓ | ✓ | ✓ | ✓ | ✓ | ✓ | ✓ | ✓ | ✓ | ✓ |
| 2075-0 | ✓ | ✓ | ✓ | ✓ | ✓ | ✓ | ✓ | ✓ | ✓ | ✓ | ✓ | ✓ | ✓ | ✓ | ✓ | ✓ | ✓ | ✓ | ✓ | ✓ |
| 2085-9 | | ✓ | | | | ✓ | | | | | | | ✓ | ✓ | | | | ✓ | ✓ | ✓ |
| 2093-3 | | ✓ | | ✓ | | | | | | | | | ✓ | ✓ | ✓ | | | ✓ | ✓ | ✓ |
| 2141-0 | | | | | | | | | | | ✓ | | | | | | | | | |
| 2143-6 | | | | | | | | | | | ✓ | | | | | | | | | |
| 2160-0 | ✓ | ✓ | ✓ | ✓ | ✓ | ✓ | ✓ | ✓ | ✓ | ✓ | ✓ | ✓ | ✓ | ✓ | ✓ | ✓ | ✓ | ✓ | ✓ | ✓ |
| 2161-8 | | | | | | | ✓ | | | | | ✓ | | | | ✓ | | | | ✓ |
| 2162-6 | | | | | | | | | | | | ✓ | | | | | | | | |
| 2276-4 | ✓ | ✓ | ✓ | ✓ | ✓ | ✓ | | ✓ | | ✓ | ✓ | ✓ | ✓ | ✓ | ✓ | ✓ | | ✓ | ✓ | ✓ |
| 2324-2 | | ✓ | | | | | | ✓ | | | | ✓ | ✓ | ✓ | | | | | | |
| 2345-7 | ✓ | ✓ | ✓ | ✓ | ✓ | ✓ | ✓ | ✓ | ✓ | ✓ | ✓ | ✓ | ✓ | ✓ | ✓ | ✓ | ✓ | ✓ | ✓ | ✓ |
| 24108-3 | | ✓ | | ✓ | | ✓ | | | ✓ | | | | ✓ | | ✓ | | | ✓ | | |
| 2458-8 | ✓ | | ✓ | | ✓ | | | | | | | ✓ | | | | | | ✓ | | |
| 2465-3 | | ✓ | | ✓ | ✓ | | | | | | | ✓ | | | | | | | | |
| 2472-9 | | | ✓ | | | | | | | | | ✓ | | | | | | ✓ | | |
| 2532-0 | ✓ | ✓ | | ✓ | ✓ | ✓ | ✓ | ✓ | | ✓ | ✓ | ✓ | ✓ | ✓ | ✓ | ✓ | | ✓ | ✓ | ✓ |
| 2571-8 | | ✓ | | ✓ | ✓ | | | | | | | | | ✓ | | | | ✓ | ✓ | ✓ |
| 26444-0 | ✓ | ✓ | ✓ | ✓ | ✓ | ✓ | ✓ | ✓ | | ✓ | ✓ | ✓ | ✓ | ✓ | ✓ | ✓ | | ✓ | ✓ | ✓ |
| 26449-9 | ✓ | ✓ | ✓ | ✓ | ✓ | ✓ | ✓ | ✓ | ✓ | ✓ | ✓ | ✓ | ✓ | ✓ | ✓ | ✓ | ✓ | ✓ | ✓ | ✓ |
| 26450-7 | ✓ | ✓ | ✓ | ✓ | ✓ | ✓ | ✓ | ✓ | ✓ | ✓ | ✓ | ✓ | ✓ | ✓ | ✓ | ✓ | ✓ | ✓ | ✓ | ✓ |
| 26453-1 | ✓ | ✓ | ✓ | ✓ | ✓ | ✓ | ✓ | ✓ | ✓ | ✓ | ✓ | ✓ | ✓ | ✓ | ✓ | ✓ | ✓ | ✓ | ✓ | ✓ |
| 26464-8 | ✓ | ✓ | ✓ | ✓ | ✓ | ✓ | ✓ | ✓ | ✓ | ✓ | ✓ | ✓ | ✓ | ✓ | ✓ | ✓ | ✓ | ✓ | ✓ | ✓ |
| 26474-7 | ✓ | ✓ | ✓ | ✓ | ✓ | ✓ | ✓ | ✓ | ✓ | ✓ | ✓ | ✓ | ✓ | ✓ | ✓ | ✓ | ✓ | ✓ | ✓ | ✓ |
| 26478-8 | ✓ | ✓ | ✓ | ✓ | ✓ | ✓ | ✓ | ✓ | ✓ | ✓ | ✓ | ✓ | ✓ | ✓ | ✓ | ✓ | ✓ | ✓ | ✓ | ✓ |
| 26484-6 | ✓ | ✓ | ✓ | ✓ | ✓ | ✓ | ✓ | ✓ | ✓ | ✓ | ✓ | ✓ | ✓ | ✓ | ✓ | ✓ | ✓ | ✓ | ✓ | ✓ |
| 26485-3 | ✓ | ✓ | ✓ | ✓ | ✓ | ✓ | ✓ | ✓ | ✓ | ✓ | ✓ | ✓ | ✓ | ✓ | ✓ | ✓ | ✓ | ✓ | ✓ | ✓ |
| 26499-4 | ✓ | ✓ | ✓ | ✓ | ✓ | ✓ | ✓ | ✓ | ✓ | ✓ | ✓ | ✓ | ✓ | ✓ | ✓ | ✓ | | ✓ | ✓ | ✓ |
| 26505-8 | ✓ | ✓ | ✓ | ✓ | ✓ | ✓ | ✓ | ✓ | | | | ✓ | ✓ | ✓ | | | | ✓ | | ✓ |

**Table 3 – continued from previous page**

| Variable | AML | BC | CLL | CRC | DLBCL | EC | FL | GE | HCC | HNSCC | MEL | MM | NSCLC | OC | Panc | RCC | SCLC | Undet. | aUC | mPC |
|---|---|---|---|---|---|---|---|---|---|---|---|---|---|---|---|---|---|---|---|---|
| 26507-4 | ✓ | ✓ | ✓ | ✓ | ✓ | ✓ |  | ✓ |  |  |  |  | ✓ |  | ✓ | ✓ |  | ✓ |  |  |
| 26508-2 | ✓ |  | ✓ |  |  |  |  |  |  |  |  |  |  |  |  |  |  |  |  |  |
| 26511-6 | ✓ | ✓ | ✓ | ✓ | ✓ | ✓ | ✓ | ✓ | ✓ | ✓ | ✓ | ✓ | ✓ | ✓ | ✓ | ✓ | ✓ | ✓ | ✓ | ✓ |
| 26515-7 | ✓ | ✓ | ✓ | ✓ | ✓ | ✓ | ✓ | ✓ | ✓ | ✓ | ✓ | ✓ | ✓ | ✓ | ✓ | ✓ | ✓ | ✓ | ✓ | ✓ |
| 2823-3 | ✓ | ✓ | ✓ | ✓ | ✓ | ✓ | ✓ | ✓ | ✓ | ✓ | ✓ | ✓ | ✓ | ✓ | ✓ | ✓ | ✓ | ✓ | ✓ | ✓ |
| 2857-1 |  |  |  |  |  |  |  |  |  |  |  |  |  |  |  |  |  | ✓ |  | ✓ |
| 2862-1 |  |  |  |  |  |  |  |  |  |  | ✓ | ✓ |  | ✓ |  |  |  |  |  |  |
| 2885-2 | ✓ | ✓ | ✓ | ✓ | ✓ | ✓ | ✓ | ✓ | ✓ | ✓ | ✓ | ✓ |  | ✓ | ✓ | ✓ | ✓ | ✓ | ✓ | ✓ |
| 2888-6 |  |  |  | ✓ |  | ✓ |  |  |  |  |  | ✓ |  | ✓ |  | ✓ |  |  |  | ✓ |
| 2889-4 |  |  |  |  |  |  |  |  |  |  |  | ✓ |  |  |  |  |  |  |  |  |
| 2947-0 |  | ✓ |  | ✓ |  | ✓ |  | ✓ |  |  | ✓ | ✓ | ✓ | ✓ | ✓ | ✓ |  |  | ✓ | ✓ |
| 2951-2 | ✓ | ✓ | ✓ | ✓ | ✓ | ✓ | ✓ | ✓ | ✓ | ✓ | ✓ | ✓ | ✓ | ✓ | ✓ | ✓ |  | ✓ | ✓ | ✓ |
| 2986-8 |  |  |  |  |  |  |  |  |  |  |  |  |  |  |  |  |  |  |  | ✓ |
| 3016-3 |  | ✓ |  |  |  | ✓ |  | ✓ |  | ✓ | ✓ |  | ✓ | ✓ | ✓ | ✓ | ✓ | ✓ |  |  |
| 30180-4 | ✓ | ✓ | ✓ | ✓ | ✓ | ✓ | ✓ | ✓ | ✓ | ✓ | ✓ | ✓ | ✓ | ✓ | ✓ | ✓ | ✓ | ✓ | ✓ | ✓ |
| 3024-7 |  |  |  | ✓ |  | ✓ |  |  | ✓ | ✓ | ✓ |  | ✓ | ✓ |  | ✓ |  |  |  | ✓ |
| 3026-2 |  |  |  | ✓ |  |  |  |  |  |  | ✓ |  | ✓ |  |  | ✓ | ✓ | ✓ |  |  |
| 30376-8 | ✓ |  |  |  |  |  |  |  |  |  |  |  |  |  |  |  |  |  |  |  |
| 30394-1 | ✓ | ✓ | ✓ | ✓ | ✓ | ✓ | ✓ | ✓ | ✓ | ✓ | ✓ | ✓ | ✓ | ✓ | ✓ |  |  | ✓ | ✓ | ✓ |
| 30395-8 | ✓ | ✓ | ✓ | ✓ | ✓ | ✓ | ✓ | ✓ | ✓ |  | ✓ |  | ✓ | ✓ | ✓ | ✓ | ✓ | ✓ | ✓ | ✓ |
| 30451-9 | ✓ |  | ✓ | ✓ |  | ✓ |  | ✓ | ✓ |  | ✓ | ✓ | ✓ | ✓ | ✓ |  |  | ✓ |  |  |
| 30522-7 |  | ✓ |  |  |  |  |  |  |  |  |  |  |  |  |  |  |  |  |  | ✓ |
| 3084-1 | ✓ |  | ✓ |  | ✓ | ✓ |  | ✓ |  | ✓ | ✓ | ✓ |  | ✓ | ✓ | ✓ |  | ✓ |  | ✓ |
| 3094-0 | ✓ | ✓ | ✓ | ✓ | ✓ | ✓ | ✓ | ✓ | ✓ | ✓ | ✓ | ✓ | ✓ | ✓ | ✓ | ✓ | ✓ | ✓ | ✓ | ✓ |
| 33358-3 |  |  |  |  |  |  |  |  |  |  |  | ✓ |  |  |  |  |  |  |  |  |
| 33558-8 |  | ✓ |  | ✓ |  | ✓ |  |  |  |  |  | ✓ |  |  |  |  |  |  |  |  |
| 33944-0 |  |  |  |  |  |  |  |  |  |  |  | ✓ |  |  |  |  |  | ✓ |  |  |
| 35591-7 |  | ✓ |  | ✓ |  | ✓ |  |  |  |  |  | ✓ | ✓ | ✓ |  | ✓ |  | ✓ | ✓ | ✓ |
| 36916-5 |  |  |  |  |  |  |  |  |  |  |  | ✓ |  |  |  |  |  |  |  |  |
| 38176-4 |  |  |  |  |  |  |  |  |  |  |  | ✓ |  |  |  |  |  |  |  |  |
| 38483-4 |  |  |  |  | ✓ |  |  |  |  |  | ✓ |  |  | ✓ |  |  |  |  |  | ✓ |
| 38875-1 |  | ✓ |  |  |  |  |  |  |  |  |  |  |  | ✓ |  |  |  | ✓ |  |  |
| 42483-8 |  |  |  |  |  |  |  |  |  |  |  | ✓ |  |  |  |  |  |  |  |  |
| 4537-7 |  | ✓ |  |  |  |  |  |  |  |  |  |  |  |  |  |  |  |  |  |  |
| 4544-3 | ✓ | ✓ | ✓ | ✓ | ✓ | ✓ | ✓ | ✓ | ✓ | ✓ | ✓ | ✓ | ✓ | ✓ | ✓ | ✓ | ✓ | ✓ | ✓ | ✓ |
| 4548-4 |  | ✓ |  |  |  |  |  |  |  |  |  |  |  | ✓ |  |  |  |  |  |  |
| 48642-3 |  | ✓ |  | ✓ | ✓ | ✓ | ✓ | ✓ | ✓ | ✓ | ✓ | ✓ | ✓ | ✓ | ✓ | ✓ | ✓ | ✓ | ✓ | ✓ |
| 48643-1 | ✓ | ✓ | ✓ | ✓ | ✓ | ✓ | ✓ | ✓ | ✓ | ✓ | ✓ | ✓ | ✓ | ✓ | ✓ | ✓ | ✓ | ✓ | ✓ | ✓ |
| 53962-7 |  |  |  |  |  |  |  | ✓ |  |  |  |  |  |  |  |  |  |  |  |  |
| 5902-2 | ✓ | ✓ |  | ✓ | ✓ | ✓ | ✓ | ✓ |  | ✓ | ✓ | ✓ | ✓ | ✓ | ✓ | ✓ |  | ✓ | ✓ | ✓ |
| 5905-5 | ✓ | ✓ | ✓ | ✓ | ✓ | ✓ | ✓ | ✓ | ✓ |  | ✓ | ✓ | ✓ | ✓ | ✓ | ✓ |  | ✓ | ✓ | ✓ |
| 6298-4 |  | ✓ |  |  | ✓ |  |  |  |  |  |  | ✓ |  | ✓ |  |  |  | ✓ | ✓ | ✓ |
| 6690-2 | ✓ | ✓ | ✓ | ✓ | ✓ | ✓ | ✓ | ✓ | ✓ | ✓ | ✓ | ✓ | ✓ | ✓ | ✓ | ✓ | ✓ | ✓ | ✓ | ✓ |
| 6768-6 | ✓ | ✓ | ✓ | ✓ | ✓ | ✓ | ✓ | ✓ | ✓ | ✓ | ✓ | ✓ | ✓ | ✓ | ✓ | ✓ | ✓ | ✓ | ✓ | ✓ |
| 6875-9 |  | ✓ |  |  |  |  |  |  |  |  |  |  |  |  |  |  |  |  |  |  |
| 69405-9 |  | ✓ | ✓ | ✓ |  | ✓ | ✓ | ✓ | ✓ | ✓ | ✓ | ✓ | ✓ | ✓ | ✓ | ✓ | ✓ | ✓ | ✓ | ✓ |
| 704-7 | ✓ | ✓ |  | ✓ | ✓ | ✓ | ✓ | ✓ | ✓ | ✓ | ✓ | ✓ | ✓ | ✓ | ✓ | ✓ | ✓ | ✓ | ✓ | ✓ |
| 705-4 | ✓ | ✓ |  |  |  |  |  |  |  |  |  |  |  |  |  |  |  |  |  |  |
| 706-2 | ✓ | ✓ | ✓ | ✓ | ✓ | ✓ | ✓ | ✓ | ✓ |  | ✓ | ✓ | ✓ | ✓ | ✓ | ✓ |  | ✓ | ✓ | ✓ |
| 707-0 | ✓ | ✓ |  |  | ✓ |  |  |  |  |  |  | ✓ | ✓ | ✓ | ✓ |  |  | ✓ | ✓ |  |
| 712-0 | ✓ | ✓ |  | ✓ |  |  |  |  |  |  | ✓ | ✓ | ✓ | ✓ | ✓ | ✓ |  |  |  | ✓ |
| 714-6 | ✓ | ✓ |  | ✓ | ✓ | ✓ |  | ✓ |  | ✓ |  | ✓ | ✓ | ✓ | ✓ | ✓ |  | ✓ |  | ✓ |
| 718-7 | ✓ | ✓ | ✓ | ✓ | ✓ | ✓ | ✓ | ✓ | ✓ | ✓ | ✓ | ✓ | ✓ | ✓ | ✓ | ✓ |  | ✓ | ✓ | ✓ |
| 731-0 | ✓ | ✓ | ✓ | ✓ | ✓ | ✓ | ✓ | ✓ | ✓ | ✓ | ✓ | ✓ | ✓ | ✓ | ✓ | ✓ | ✓ | ✓ | ✓ | ✓ |
| 732-8 | ✓ | ✓ |  | ✓ |  | ✓ |  | ✓ |  |  |  | ✓ | ✓ |  | ✓ | ✓ |  | ✓ |  | ✓ |
| 736-9 | ✓ | ✓ | ✓ | ✓ | ✓ | ✓ | ✓ | ✓ | ✓ | ✓ | ✓ | ✓ | ✓ | ✓ | ✓ | ✓ | ✓ | ✓ | ✓ | ✓ |
| 737-7 | ✓ | ✓ |  | ✓ |  | ✓ |  | ✓ |  |  | ✓ | ✓ | ✓ | ✓ |  | ✓ | ✓ |  |  | ✓ |
| 742-7 | ✓ | ✓ |  | ✓ | ✓ | ✓ | ✓ | ✓ | ✓ | ✓ | ✓ | ✓ | ✓ | ✓ | ✓ | ✓ | ✓ | ✓ | ✓ | ✓ |

**Table 3 – continued from previous page**

| Variable | AML | BC | CLL | CRC | DLBCL | EC | FL | GE | HCC | HNSCC | MEL | MM | NSCLC | OC | Panc | RCC | SCLC | Undet. | aUC | mPC |
|---|---|---|---|---|---|---|---|---|---|---|---|---|---|---|---|---|---|---|---|---|
| 743-5 | ✓ | ✓ | ✓ | ✓ | | ✓ | | ✓ | | | | | | | | | | ✓ | | ✓ |
| 744-3 | ✓ | ✓ | ✓ | ✓ | | ✓ | | | ✓ | | ✓ | ✓ | ✓ | ✓ | | ✓ | | ✓ | ✓ | ✓ |
| 751-8 | ✓ | ✓ | | ✓ | ✓ | ✓ | ✓ | ✓ | ✓ | ✓ | ✓ | ✓ | ✓ | ✓ | ✓ | ✓ | ✓ | ✓ | ✓ | ✓ |
| 753-4 | ✓ | | | | | | | | | | | | | | | | | | | |
| 764-1 | | ✓ | | | | ✓ | | | | | | | ✓ | ✓ | | | | ✓ | | |
| 769-0 | ✓ | ✓ | | ✓ | | | | | | | | | | | | | | ✓ | | |
| 770-8 | | ✓ | | ✓ | | ✓ | | | | | ✓ | ✓ | ✓ | ✓ | ✓ | | | ✓ | ✓ | ✓ |
| 77146-9 | | | | | | | | | | | | | | | | | | ✓ | | |
| 777-3 | ✓ | ✓ | ✓ | ✓ | ✓ | ✓ | | ✓ | ✓ | ✓ | ✓ | ✓ | ✓ | ✓ | ✓ | ✓ | ✓ | ✓ | ✓ | ✓ |
| 789-8 | ✓ | ✓ | ✓ | ✓ | ✓ | ✓ | | ✓ | | ✓ | ✓ | ✓ | ✓ | ✓ | ✓ | ✓ | | ✓ | ✓ | ✓ |
| 98979-8 | | ✓ | | ✓ | | | | | | | ✓ | | | ✓ | ✓ | | | | | ✓ |

# D. Extended Results & Discussion

Overall, TwinWeaver extends the domain adaptability of LLM-based models, providing a unified framework for joint continuous trajectory modeling and event prediction, with zero-shot generalization on out-of-distribution clinical trial datasets."

### D.1. Sample Size Ablation (GDT vs Baselines on 2k Patients)

To ensure a fair comparison, we also trained GDT on the identical 2,000-patient subset used for the baselines. The reduced-data GDT significantly outperforms the strongest baseline (TiDE FT-Multi, $p < 0.001$, Wilcoxon signed-rank test), yet remains below the full-data GDT, as shown in Figure 7.

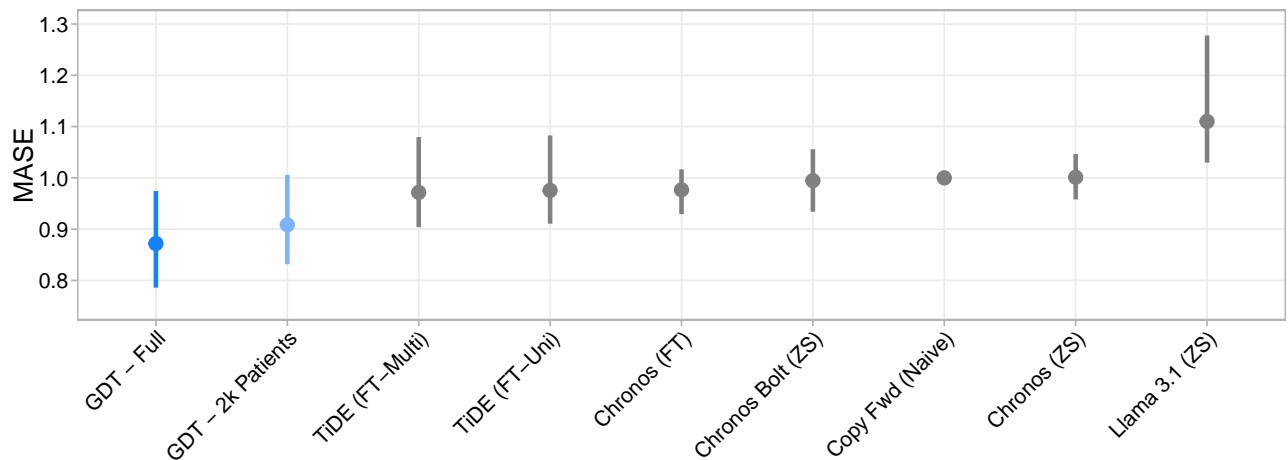

*Figure 7.* **GDT trained on the same 2,000-patient subset as the baselines still outperforms them, though it underperforms the full model.** Comparison of MASE for forecasting blood variables. The reduced-data GDT (Light Blue) surpasses TiDE and Chronos but does not match the performance of the GDT model trained on the full pan-cancer dataset (Teal), highlighting the benefits of scale. ZS denotes zero-shot, FT is fine tuned, Uni is univariate input, Multi is multivariate input.

## D.2. Forecasting Full Results (RWD)

We present the forecasting results across all variables in Figure 8.

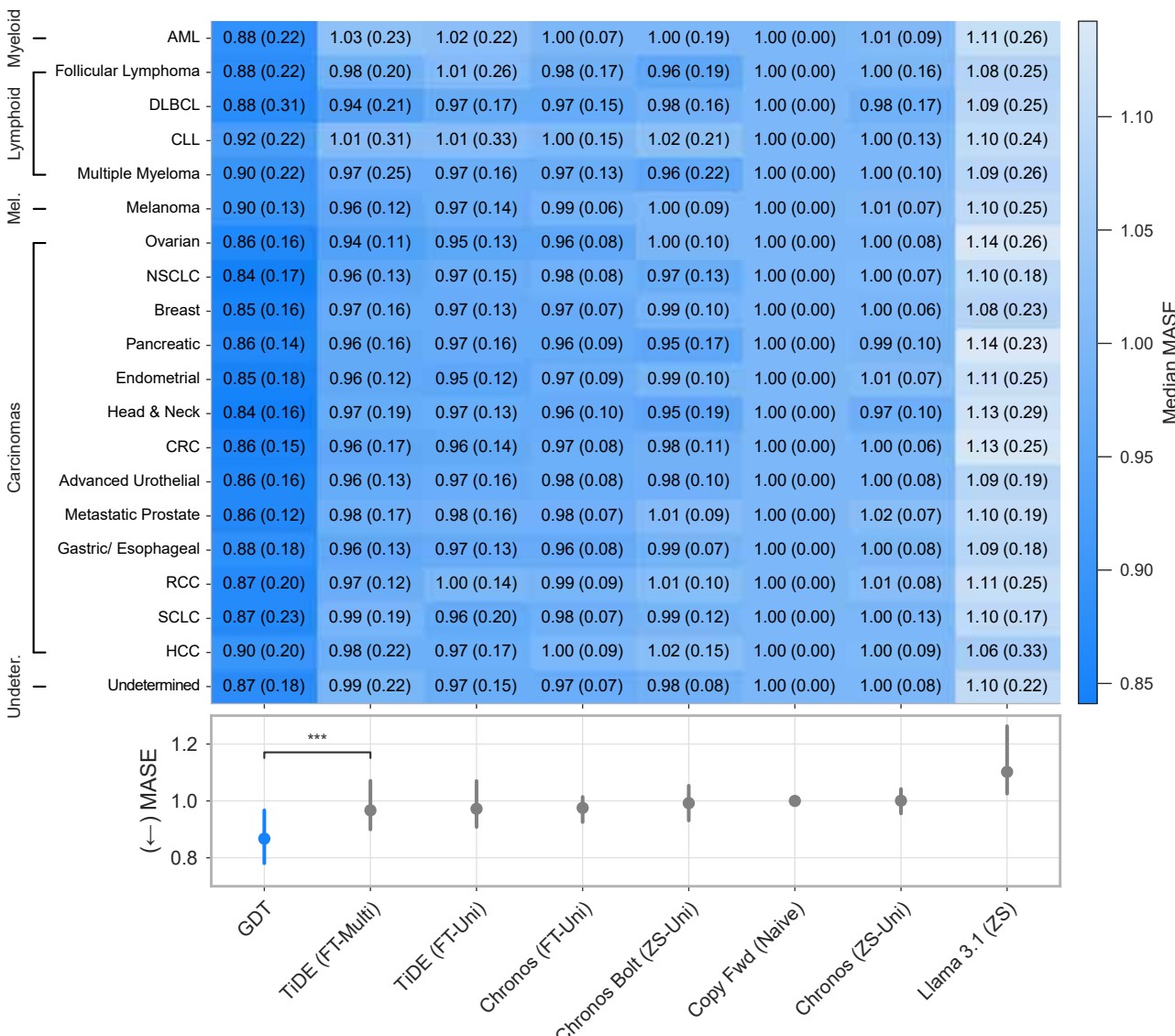

*Figure 8.* **Comprehensive blood biomarker forecasting results across all variables and indications.** The heatmap displays the Median MASE for GDT versus baselines across all 20 cancer types. GDT consistently achieves lower error rates across both myeloid/lymphoid and solid tumor indications compared to standard time-series models and zero-shot LLMs. ZS denotes zero-shot, FT is fine tuned, Uni is univariate input, Multi is multivariate input.

## D.3. Clinical Trial Task Setup

To construct the dataset, we prioritized the largest combination of clinical trials featuring overlapping treatment regimens. The two largest trials were designated as the training set for the experimental settings with fine-tuning: OAK ($n = 1126$; comparing atezolizumab vs. docetaxel) and IMpower131 ($n = 949$; comparing atezolizumab + carboplatin + paclitaxel vs. atezolizumab + carboplatin + nab-paclitaxel vs. carboplatin + nab-paclitaxel). For internal validation and hyperparameter tuning, we employed a random 20% holdout subset of these training trials. To rigorously evaluate the model's ability to generalize across different patient populations and protocols, we adopted a leave-trial-out testing strategy rather than a random pooled split. We utilized the two largest trials (OAK, IMpower131) for training to maximize data diversity during

learning, while reserving two distinct, independent trials (POPLAR, IMpower130) strictly for external testing: POPLAR ($n = 263$; atezolizumab vs. docetaxel) and IMpower130 ($n = 680$; atezolizumab + carboplatin + nab-paclitaxel vs. carboplatin + nab-paclitaxel). We used 17 blood biomarkers for forecasting, as well as survival and progression for event evaluation.

### D.4. Further Reasoning Results

#### D.4.1. REASONING DETAILS

A key design consideration of the reasoning extension is that ground-truth outcomes are used only by the teacher model when synthesizing the initial rationale corpus; during subsequent GRPO training and at inference, GDT generates its rationale and numerical forecast from the input patient history alone, with no access to true future values. Under this setup, the final reasoning model recovers from SFT-only performance (MASE 0.863) to 0.862 after GRPO alignment with a deterministic numerical reward $r = -\min\left(\frac{1}{N} \sum_{i=1}^{N} |\hat{y}_i - y_i|, 20\right)$, which directly ties policy updates to forecast accuracy. The marginal magnitude of this recovery suggests that hybrid token/numeric objectives are a viable but non-trivial direction, motivating dedicated future research on numerically grounded clinical LLM training.

To validate the clinical semantic alignment of the generated rationales, we employed a keyword-based concept analysis focusing on four distinct therapy regimens found in the NSCLC cohort: Chemotherapy, Immunotherapy (IO), Targeted Therapy, and Chemo-IO combinations. We manually curated a dictionary of inclusion and exclusion terms for specific clinical concepts (e.g., 'Marrow Suppression', 'Immune Activation', 'EGFR mutation'). Inclusion terms captured synonymous phrases and specific biological mechanisms, while exclusion terms were strictly applied to filter out negated contexts or unrelated mentions to ensure false positives were minimized.

For each concept within a therapy group, we computed two metrics:

- **Frequency:** The percentage of generated reasoning chains within a specific therapy group that contained at least one verified instance of the concept.

- **Relative Importance:** A normalized measure of concept specificity calculated row-wise. For a given concept, this is defined as its frequency in the specific therapy group divided by the maximum frequency of that concept observed across all four therapy groups. This scaling ensures that the therapy group with the strongest association for a given concept is assigned a value of 1.0, highlighting relative specificity regardless of absolute frequency.

#### D.4.2. REASONING EMBEDDINGS VISUALIZATION

To assess the reasoning chains, we extract them from the test set, and then embed them using Qwen3-8B. Finally, we perform UMAP (Uniform Manifold Approximation and Projection) and two-dimensional display of the embedded data.

#### D.4.3. EXAMPLE REASONING CHAIN

```
<prediction>
Task 1 is forecasting:
3 weeks later, the patient visited and experienced the following:
  neutrophils - 26499-4 is 3.4.

2 weeks later, the patient visited and experienced the following:
  neutrophils - 26499-4 is 4.6.

2 weeks later, the patient visited and experienced the following:
  neutrophils - 26499-4 is 4.8.

2 weeks later, the patient visited and experienced the following:
  neutrophils - 26499-4 is 4.6.

2 weeks later, the patient visited and experienced the following:
  neutrophils - 26499-4 is 3.9.
```

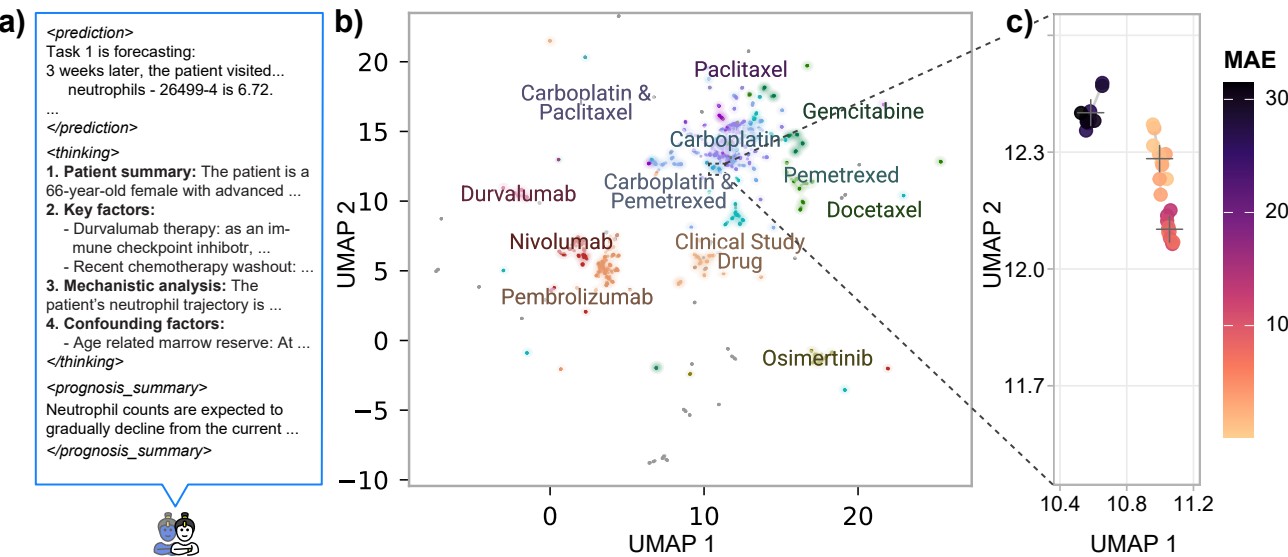

*Figure 9.* **GDT can be extended to provide reasoning, exemplified here in NSCLC neutrophil forecasting. a)** Example of a generated prediction and reasoning chain. **b)** UMAP visualization of reasoning chain embeddings, colored by the top 12 most common categories, shows structural alignment with clinical factors such as therapy types. **c)** Zoomed view of the patient with the highest MAE on the test set; the centroids (+) and clustering indicate that generated reasoning chains are consistent across multiple runs.

```
</prediction>
<thinking>
1. **Patient Summary:** The patient is a 52-year-old male with metastatic NSCLC,
   previously diagnosed with prostate cancer, currently on nivolumab (first-line
   immune checkpoint inhibitor) as his most recent therapy. He has progressive
   disease with bone and liver metastases, confirmed PD-L1 negativity on both
   tumor and immune cells, and a complex genomic profile including TP53, STK11,
   SMARCA4, and CDKN2A mutations, with MSI-stable status and no actionable EGFR or
   ALK alterations. Recent labs show stable renal function, mild anemia, and
   fluctuating but generally preserved neutrophil counts (last value: 4.4 x
   10^9/L). ECOG has improved from 2 to 0, indicating clinical stabilization. No
   active infection or acute inflammatory events are documented. The patient is
   not receiving G-CSF or other myeloid growth factors.

2. **Key Predictive Factors:**
   - **Nivolumab monotherapy:** Immune checkpoint inhibitors like nivolumab rarely
     cause severe neutropenia but can induce immune-mediated bone marrow
     suppression or transient cytopenias via T-cell activation against
     hematopoietic precursors. Neutrophil counts typically remain stable or show
     mild fluctuations.
   - **Recent partial response (PR) to therapy:** The documented PR suggests
     effective tumor control, which reduces tumor-derived cytokines (e.g., IL-6,
     G-CSF) that can stimulate neutrophil production. This may lead to a gradual
     normalization of neutrophil counts toward baseline.
   - **Persistent bone metastases:** Bone marrow infiltration by metastatic NSCLC
     may impair hematopoietic reserve, particularly affecting granulopoiesis.
     However, the absence of severe cytopenias (e.g., hemoglobin >13, platelets
     >300) suggests compensatory hematopoiesis is still functional.
   - **Genomic profile (TP53, STK11, SMARCA4 mutations):** These mutations are
     associated with genomic instability and may impair DNA repair in
     hematopoietic stem cells, potentially increasing susceptibility to
     subclinical marrow stress or delayed recovery after minor insults.
   - **Trend of neutrophil stability over prior visits:** Neutrophils have
     fluctuated between 3.5-5.2 over the last 5 visits (spanning 10 weeks), with
     no consistent downward trend. This suggests a homeostatic equilibrium under
```

```
        current conditions.

3. **Mechanistic Analysis:** The neutrophil trajectory is governed by a balance
    between immune-mediated stress and compensatory hematopoiesis. Nivolumab does
    not directly suppress neutrophil production but may induce low-grade immune
    activation that occasionally triggers transient neutrophil consumption or
    redistribution. The partial response reduces tumor-derived inflammatory signals
    that previously stimulated neutrophil release, leading to a gradual
    normalization of baseline neutrophil production. However, the presence of bone
    metastases limits the marrow's capacity for robust rebound, preventing
    overshoot. The TP53/STK11/SMARCA4 mutations may slightly delay recovery from
    minor stressors (e.g., subclinical inflammation or transient immune
    activation), explaining why neutrophil counts do not fully return to
    pre-treatment levels. The observed fluctuations (e.g., 4.4 -> 3.4 -> 4.6 -> 4.8
    -> 4.6 -> 3.9) reflect this dynamic equilibrium: transient immune activation
    (possibly from tumor cell death or inflammatory signaling from the tumor
    microenvironment) causes a slight dip, followed by compensatory granulopoiesis.
    The 2-week intervals between visits allow for partial recovery before the next
    immune-mediated perturbation, resulting in a cyclical pattern without sustained
    decline. The absence of G-CSF or infection means no external stimulus for rapid
    rise or crash.

4. **Confounding Factors:** Subclinical infection (e.g., from oral soft tissue
    disease or lymph node enlargement) could trigger neutrophilia, but no fever or
    elevated CRP/ESR is documented. Advanced age (52) is not a major risk factor
    for neutropenia in this context, but reduced marrow reserve due to prior cancer
    and metastases may blunt recovery if stressors accumulate.
</thinking>

<prognosis_summary>
Neutrophil counts are expected to remain within the low-normal to normal range with
    minor fluctuations, reflecting a dynamic equilibrium between immune-mediated
    stress and compensatory hematopoiesis, with no evidence of progressive
    myelosuppression. The primary clinical risk is transient neutropenia without
    febrile episodes, unlikely to require treatment interruption.
</prognosis_summary>
```

## D.5. Single Indication Training vs Pan-Cancer

To understand the contribution of the pan-cancer training, we performed an ablation study where we trained GDT on the three cancer types with the fewest patients using only cancer indication data. We focus on the fewest patient counts, as they are most likely to benefit from the pan-cancer capabilities and transfer learning.

In Figure 10, we see that the pan-cancer model consistently improves performance over the single indication models, albeit at a small level.

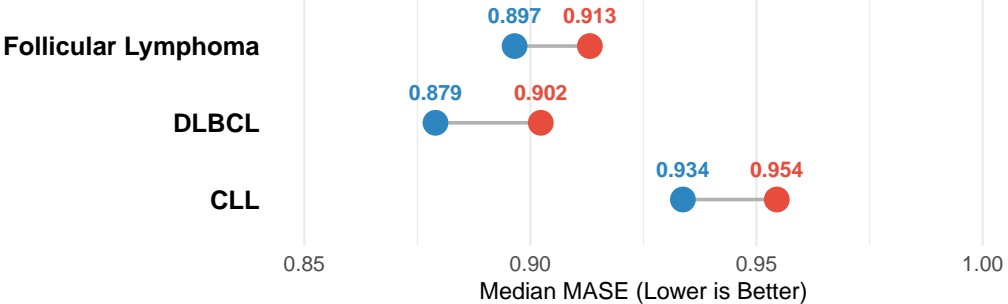

*Figure 10.* **Pan-cancer training consistently improves performance over single-indication models.** Comparison of Median MASE for the three cancer indications with the fewest patients ($< 500$). The Pan-cancer GDT model (Blue) achieves lower error rates than models trained solely on the specific indication data (Red), demonstrating the benefits of transfer learning in low-data regimes.

## D.6. Assessing Longitudinal Input Length on Performance

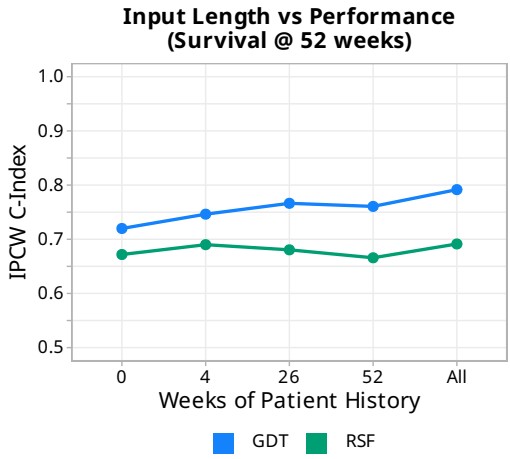

*Figure 11.* **Longer historic patient trajectories improve GDT, implying that clinical trials could benefit from additional information on patient history.** Here, we compare Random Survival Forest (RSF) with GDT on NSCLC for 52-week survival prediction on IPCW C-index. We select patients from the test set, patients with at least 52 weeks of input history, then split only at the start of therapy, and systematically cut the input history to the specific input length. 0-4 weeks can be compared to the data currently commonly gathered in clinical trials. We see that more input history improves model performance, implying that clinical trials could benefit from longer patient timelines. We additionally see that GDT outperforms RSF consistently, which focuses on last observed values only.

## D.7. Detailed Clinical Trial Forecasting Results

We provide the comprehensive breakdown of forecasting performance across all models for the clinical trial validation. On the trial POPLAR, the performance ranking by median MASE is as follows: GDT SFT (0.883, IQR 0.09), TiDE fine-tuned univariate (0.936, IQR 0.07), GDT zero-shot (0.947, IQR 0.121), Chronos fine-tuned (0.956, IQR 0.054), TiDE fine-tuned multivariate (0.993, IQR 0.131), Chronos zero-shot (0.998, IQR 0.001), Copy Forward (1.0), Chronos Bolt zero-shot (1.0), and Llama 3.1 (1.164, IQR 0.175).

For trial IMpower130, the ranking follows a similar pattern: GDT SFT (0.754, IQR 0.385), TiDE fine-tuned univariate (0.840, IQR 0.119), GDT zero-shot (0.874, IQR 0.163), Chronos fine-tuned (0.885, IQR 0.075), TiDE fine-tuned multivariate (0.853, IQR 0.208), Chronos zero-shot (0.998, IQR 0.001), Copy Forward (1.0), Chronos Bolt zero-shot (1.0), and Llama 3.1 (1.126, IQR 0.156).

We observe that the TiDE univariate model generally outperforms the multivariate equivalent in this setting. We hypothesize that the reason is the focus on baseline information, limiting the correlations the multivariate model can capture, thereby allowing the univariate model to converge more easily.

## D.8. Isotonic Regression Results

### D.8.1. PAN-CANCER REAL-WORLD DATA

In the pan-cancer real-world data results, we see an overall slight improvement when including isotonic regression (Figure 12). The Brier score shows an increased performance over the base model. However, further work is required for well calibrated probabilities, as seen in Figure 13.

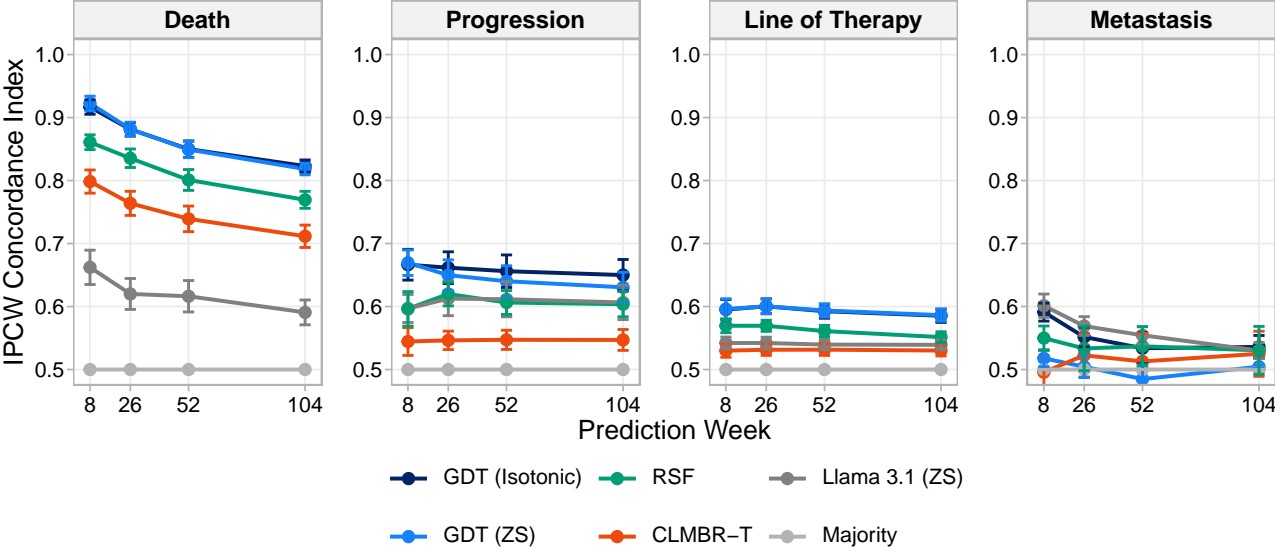

*Figure 12.* **Isotonic regression post-processing maintains robust ranking performance on pan-cancer real-world data.** The plots show the IPCW C-Index for Survival and Disease Progression. The performance of the isotonic version (Teal) is highly similar to the original GDT outputs (Light Blue), indicating that enforcing monotonicity does not degrade the model's discriminative ability.

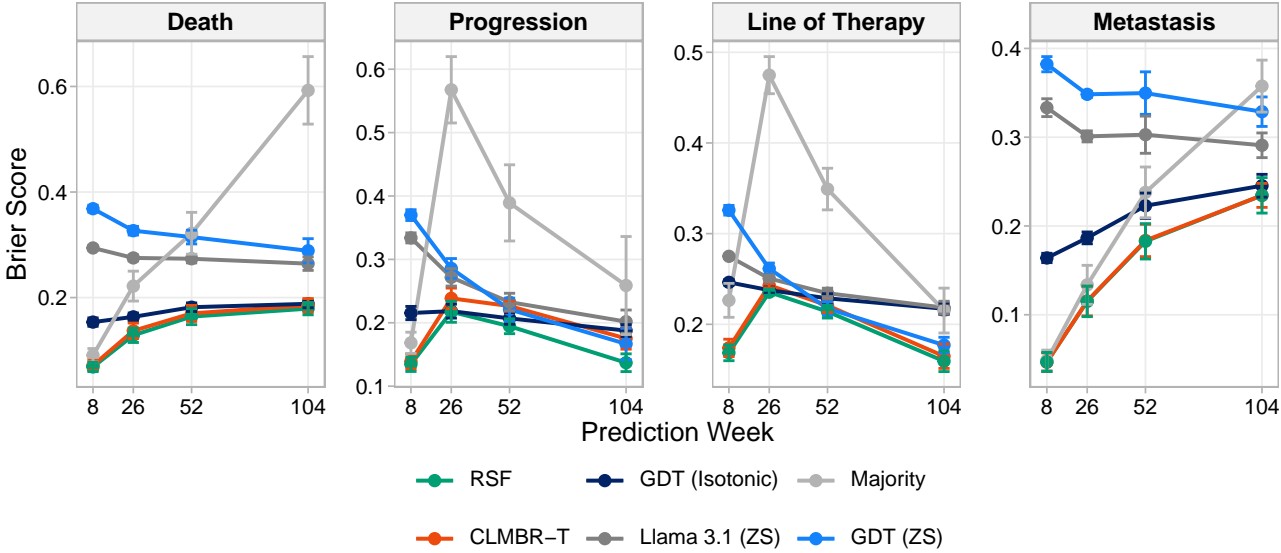

*Figure 13.* **Brier score analysis shows improvement over the base model with isotonic regression but indicates a need for further calibration.** Lower Brier scores indicate better calibration. While isotonic regression improves the score compared to the raw GDT output, particularly for Line of Therapy and Death, the results suggest that further work is needed to generate well-calibrated probabilities.

D.8.2. CLINICAL TRIAL RESULTS

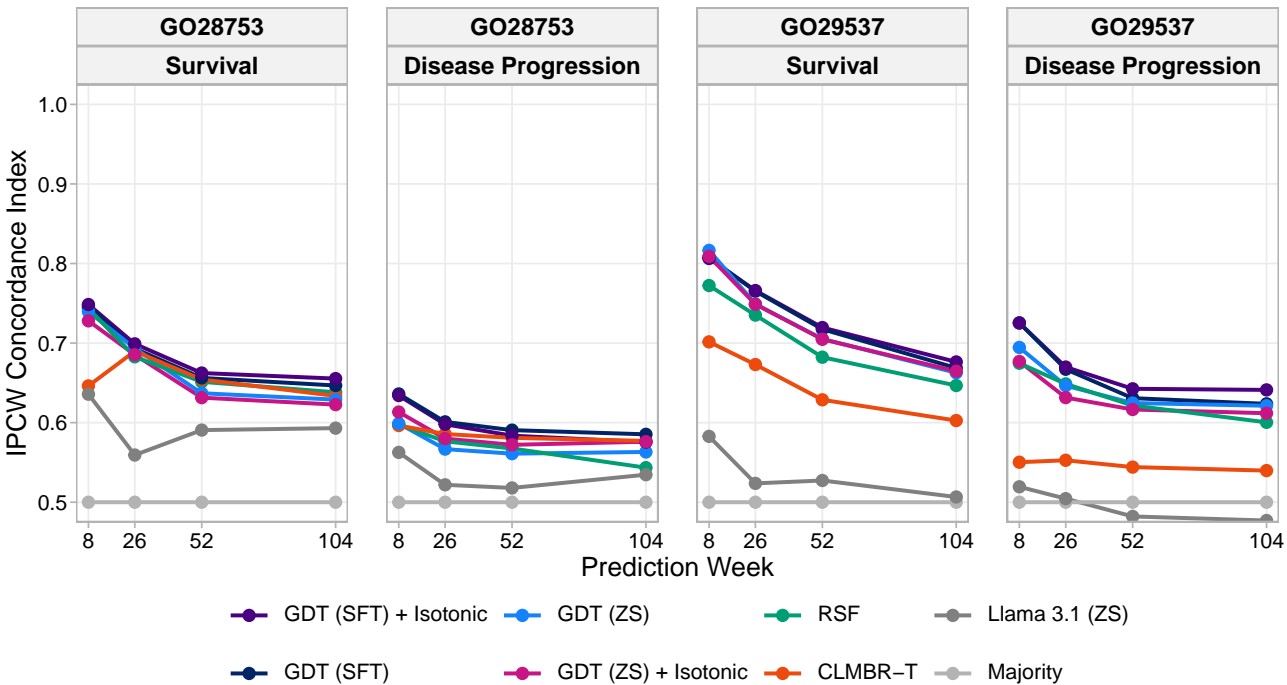

*Figure 14.* **Isotonic regression has minimal impact on clinical trial C-Index as GDT predictions are largely naturally monotonic.**
Results for trials POPLAR (GO28753) and IMpower130 (GO29537) show that the isotonic regression curves overlap significantly with the standard GDT curves, as only a small fraction (0.37%) of the zero-shot samples violated monotonicity.

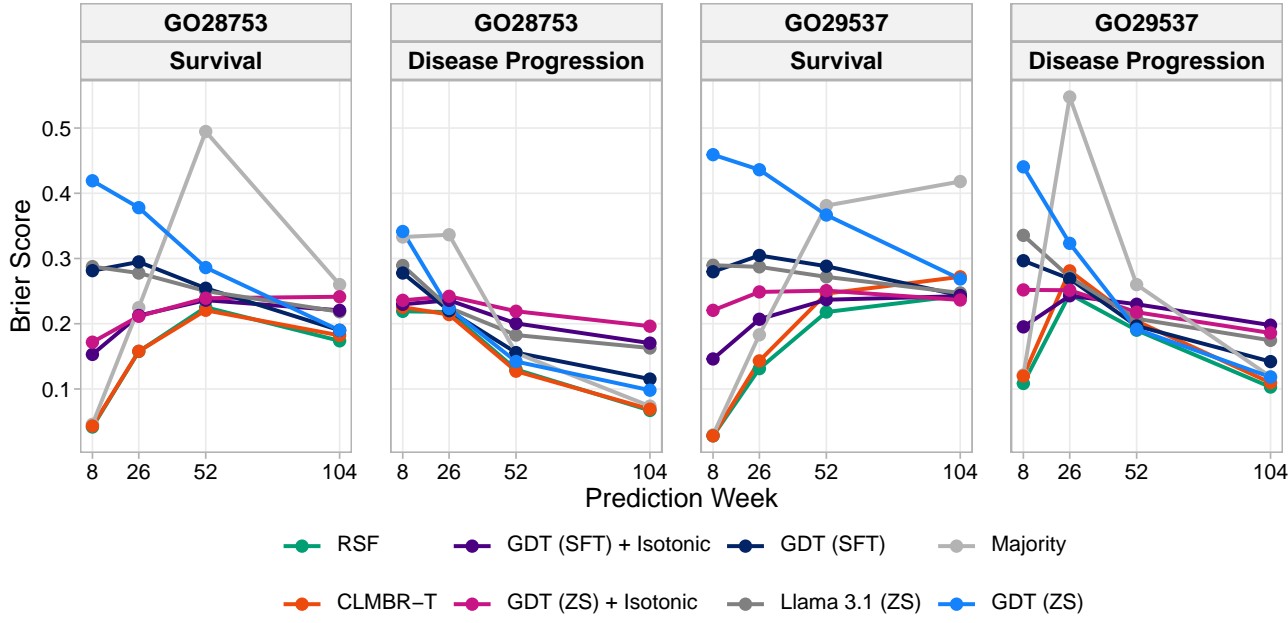

*Figure 15.* **Brier scores for clinical trials highlight the need for improved probability calibration despite strong ranking performance.**
Comparing the Brier scores across models on trials POPLAR (GO28753) and IMpower130 (GO29537) reveals that while GDT ranks patients effectively (high C-Index), the absolute probability estimates require refinement to minimize the Brier score.

## D.9. Example Forecast Predictions

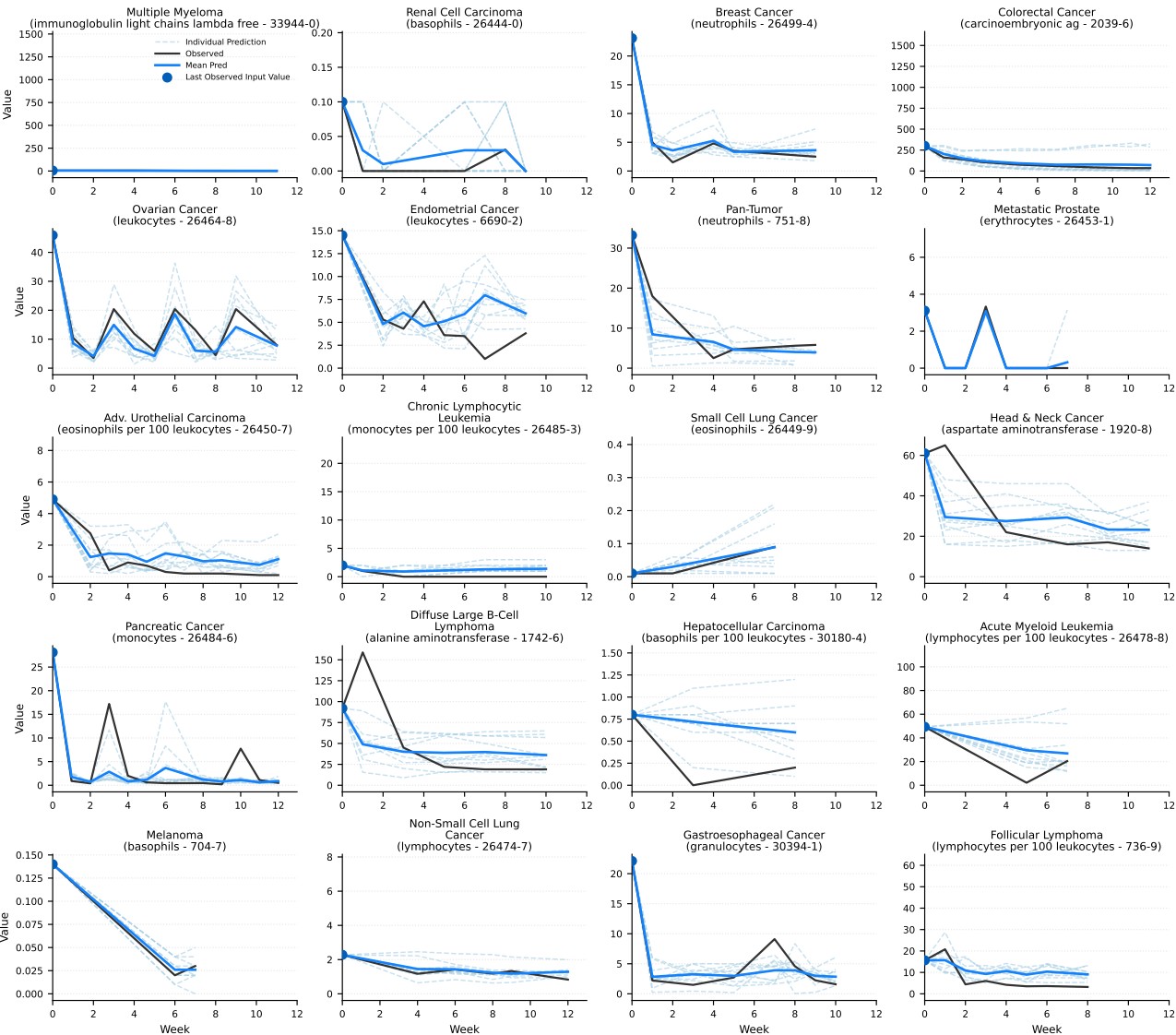

*Figure 16.* **Example blood biomarker forecasts by GDT.** The longitudinal trajectory of different blood-based markers are shown for a variety of tumor indications. Measured values are displayed in grey and predicted values in blue. Note that in some cases where the average prediction is incorrect, individual predictions sometimes are still able to capture the shape or values, such as the first spike in pancreatic cancer, or the dip in hepatocellular carcinoma. However, in other cases GDT is not able to capture large variability in the results, such as an initial strong increase in the alanine aminotransferase values in DLBCL.

### D.10. Synthetic Patient Example

Using a synthetic patient to ensure data privacy compliance.

Input:

```
The following is a patient, starting with the demographic data, following visit by
    visit everything that the patient experienced. All lab codes refer to LOINC
    codes.

Starting with demographic data:
  Patient gender is male,
  age of patient is 77 years,
    ...

On the first visit, the patient experienced the following:
  advanced cancer diagnosis is non small cell lung cancer,
  initial cancer diagnosis is non small cell lung cancer.
  metastasis Pleura is diagnosed.

2 weeks later, the patient visited and experienced the following:
  <genetic>
  ...
  Tissue: pleura Gene: CHEK2 short variant: frameshift parameters: likely truncation,
  ...
  </genetic>.

1 weeks later, the patient visited and experienced the following:
  ECOG is 2,
  alanine aminotransferase - 1742-6 is 9,
  albumin - 1751-7 is 39,
  calcium - 17861-6 is 8.9,
  aspartate aminotransferase - 1920-8 is 20,
  bilirubin - 1975-2 is 0.3,
  carbon dioxide - 2028-9 is 25,
  neutrophils per 100 leukocytes - 26511-6 is 72,
  platelets - 26515-7 is 419,
  ...
  potassium - 2823-3 is 4.7,
  protein - 2885-2 is 61,
  sodium - 2951-2 is 132,
  basophils per 100 leukocytes - 30180-4 is 0.3,
  ...

1 weeks later, the patient visited and experienced the following:
  Other fatigue is diagnosed,
  Encounter for antineoplastic immunotherapy - Z51.12 is diagnosed,
  drug pembrolizumab is 200,
  ECOG is 1,
  alanine aminotransferase - 1742-6 is 9,
  albumin - 1751-7 is 42,
  calcium - 17861-6 is 9.4,
  aspartate aminotransferase - 1920-8 is 21,
  neutrophils - 26499-4 is 5.9,
  neutrophils per 100 leukocytes - 26511-6 is 73.2,
  platelets - 26515-7 is 243,
  ...
  thyroxine - 3026-2 is 6.7,
  urea nitrogen - 3094-0 is 10,
  thyroxine free index - 32215-6 is 2,
```

```
  line number is 1,
  body height is 180.34,
  body weight is 63.45752,
  oxygen saturation in arterial blood by pulse oximetry is 98.

...

Here we repeat the last observed values of each genetic event in the input data:

0 weeks later, the patient visited and experienced the following:
  ...
  Tissue: pleura Gene: CHEK2 short variant: frameshift parameters: likely truncation,
  ...
  </genetic>.
  ...

The most recent line of therapy:
  Pembrolizumab

The last values of the variables in the input data are:
  hematocrit - 20570-8 was 33.6
  creatinine - 2160-0 was 0.6
  platelets - 26515-7 was 257

You will now have multiple tasks to complete. Please answer for each task in the
    same order as they are presented. Before every response state the task nr, e.g.
    'Task 2:'.

Task 1 is forecasting:
Your task is to predict the future values of the following variables for each
    cumulative week starting from the last visit:

  hematocrit - 20570-8 the future weeks 1, 4, 7, 10
  creatinine - 2160-0 the future weeks 1, 4, 7, 10
  platelets - 26515-7 the future weeks 1, 4, 7, 10

Task 2 is time to event prediction:
Your task is to predict whether the following event was censored 1 weeks from the
    last clinical visit and whether the event occurred or not: death.
Please provide your prediction in the following format: 'Here is the prediction:
    the event (<name of event>) was [not] censored and [did not occur]/[occurred].'
```

Output:

```
Task 1 is forecasting:
1 weeks later, the patient visited and experienced the following:
  hematocrit - 20570-8 is 36.1,
  creatinine - 2160-0 is 0.65,
  platelets - 26515-7 is 260.

3 weeks later, the patient visited and experienced the following:
  hematocrit - 20570-8 is 39,
  creatinine - 2160-0 is 0.6,
  platelets - 26515-7 is 257.

3 weeks later, the patient visited and experienced the following:
  hematocrit - 20570-8 is 36.6,
```

```
  creatinine - 2160-0 is 0.63,
  platelets - 26515-7 is 271.

3 weeks later, the patient visited and experienced the following:
  hematocrit - 20570-8 is 35.7,
  creatinine - 2160-0 is 0.61,
  platelets - 26515-7 is 257.

Task 2 is time to event prediction:
Here is the prediction: the event (death) was not censored and did not occur.
```

## D.11. System Prompt

```
As a specialist predictive model in personalized medicine, your task is to forecast
    the health trajectory of cancer patients by integrating genomic data, lifestyle
    factors, treatment history and anything else provided about the patient. Use
    the provided patient data, including genetic mutations, biomarker levels, and
    previous treatment responses, to predict all requested tasks. Deliver precise
    and clinically relevant predictions to enhance patient care and treatment
    planning.
```

## D.12. Qwen3 Next 80B-A3B Prompt

The following prompt was used to generate the initial reasoning chains using the Qwen3 Next 80B-A3B model. Note, that we then replace the prediction with the true prediction, and also shift it to the beginning, since this showed to provide more stable results in early experiments.

```
You are an expert hematologist-oncologist. You will receive a complete
patient history and a specific task to predict the patient's neutrophil
trajectory.

Your primary goal is to generate a step-by-step reasoning chain that
leads to your prediction. This rationale is more important than the
prediction itself.

Structure your entire response using the following tags. Do not include
any text outside of these tags.

<thinking>
Inside this tag, you must follow this four-step reasoning process:

1. **Patient Summary:** Briefly summarize the patient's current oncological
   and hematological status. Focus on the diagnosis, active treatments,
   and the most recent relevant lab values.
2. **Key Predictive Factors:** Identify the **top 5 most influential factors** from
    the patient's record that will drive the neutrophil trajectory.
   List each factor (e.g., specific drug, time since last treatment,
   comorbidity, recent lab trend) and provide a concise justification
   for its high importance.
3. **Mechanistic Analysis:** This is the most critical step. Synthesize
   the 5 factors you identified. Provide a detailed, step-by-step
   biological explanation of how these factors will interact to
   influence the neutrophil count *over time*.
   * Describe the specific biological pathways involved (e.g.,
```

```
        myelosuppression from a specific drug class, hematopoietic
        recovery kinetics, effects of G-CSF on bone marrow
        precursors, inflammatory cytokine release).
    * Explain the expected *timing* of these effects (e.g., "The
        patient is X days post-[Chemo], so we expect the nadir
        around day Y," or "The recent G-CSF administration will
        likely cause a transient leukocytosis followed by...").
4. **Confounding Factors:** Briefly mention 1-2 other factors (e.g.,
    potential infection, patient age, nutritional status) that could
    complicate or alter your primary predicted trajectory.

</thinking>

<prognosis_summary>
Based on your thinking and rationale, provide a 1-2 sentence summary of the expected
neutrophil trend (e.g., "Expect sharp decline into severe neutropenia,"
or "Anticipate slow but steady recovery") and the primary clinical risk
(e.g., "High risk of febrile neutropenia," or "Risk of treatment delay").
</prognosis_summary>

<prediction>
[Place the final, formatted prediction here as specified in the task
description.]
</prediction>
```

## D.13. Expanded Baseline Comparisons on Clinical Trials

To further validate the performance of Genie Digital Twin (GDT) against recent state-of-the-art specialized Electronic Health Record (EHR) models, we expanded our baseline comparisons on the clinical trial datasets. We included MOTOR and a version of CLMBR-T trained entirely from scratch on the specific clinical trial vocabulary. As shown in Table 4, GDT with Supervised Fine-Tuning (SFT) outperforms all evaluated specialized time-to-event architectures across the clinical trial event tasks.

*Table 4.* **Expanded event prediction baselines evaluated on clinical trial datasets.** GDT demonstrates superior risk stratification (IPCW C-Index) compared to standard and specialized EHR foundation models.

| Model | Training Setup | Overall Avg C-Index |
|---|---|---|
| **GDT (Ours)** | **SFT** | **0.6726** |
| RSF | Grid Search | 0.6524 |
| MOTOR | Pretrained + CoxPH Head | 0.6117 |
| CLMBR-T | Pretrained + CoxPH Head | 0.6098 |
| CLMBR-T | Trained from Scratch | 0.5966 |

## D.14. Influence of LLM Prior and Base Model Initialization

To isolate the performance drivers of our framework and explore sensitivity to the language model, we evaluated both Llama 3.1 and II-Medical-8B-1706 models, fine-tuned (FT) from scratch on the trial data.

As shown in Tables 5 and 6, the TwinWeaver framework successfully improves model performance across different foundational LLMs for both event prediction and biomarker forecasting. The pre-trained priors of these models provide a strong initialization point, with GDT (Llama 3.1 SFT) achieving the highest overall performance.

## D.15. Event Target Formulation Sensitivity

We evaluated the sensitivity of the model to the exact text descriptions used for the 3-class target formulation in the landmark event tasks. We performed an ablation for the trial SFT setting by substituting the standard target vocabulary with semantic synonyms (e.g., swapping "occurred" for "happened").

*Table 5.* Event prediction performance (Overall Avg C-Index) across different base models and initialization strategies.

| Foundation Model | Evaluation Setting | Overall Avg C-Index |
|---|---|---|
| **GDT (Llama 3.1)** | **SFT (Ours)** | **0.6726** |
| Llama | FT from Scratch | 0.6686 |
| II-Medical | FT from Scratch | 0.6603 |
| GDT (Llama 3.1) | Zero-Shot | 0.6568 |
| Llama | Zero-Shot | 0.5399 |
| II-Medical | Zero-Shot | 0.5388 |

*Table 6.* Forecasting performance (Median MASE) across different base models on POPLAR and IMpower130 clinical trials.

| Foundation Model | Evaluation Setting | POPLAR (IQR) | IMpower130 (IQR) |
|---|---|---|---|
| **GDT (Llama 3.1)** | **SFT (Ours)** | **0.8834 (0.0907)** | **0.7549 (0.3859)** |
| II-Medical | FT from scratch | 0.8984 (0.1078) | 0.8209 (0.2870) |
| Llama | FT from scratch | 0.9243 (0.1214) | 0.7782 (0.3676) |
| GDT (Llama 3.1) | Zero-Shot | 0.9471 (0.1213) | 0.8748 (0.1631) |
| Llama | Zero-Shot | 1.1641 (0.1759) | 1.1269 (0.1564) |
| II-Medical | Zero-Shot | 1.5320 (0.3373) | 1.2872 (0.7051) |

As reported in Table 7, while there is an expected slight degradation because the model was heavily fine-tuned on the original phrasing, the ablated model maintains highly competitive clinical ranking performance, suggesting that the architecture is robust to minor prompt variations.

*Table 7.* Ablation study on target phrase sensitivity for event prediction tasks.

| GDT-SFT Prompt Target Formulation | Overall Avg C-Index |
|---|---|
| **GDT+SFT:** "occurred" / "did not occur" / "censored." | **0.6726** |
| **GDT+Ablation SFT:** "happened" / "did not happen" / "was censored." | 0.6509 |

### D.16. Inter-Variable Variance and Model Stability

In our forecasting evaluations, we report the median Mean Absolute Scaled Error (MASE) along with the Interquartile Range (IQR) for each indication. It is important to note that the high variance (large IQR values) observed in these aggregate results is primarily an artifact of cross, variable aggregation. Combining different biological markers, each with distinct underlying kinetics, measurement scales, and natural fluctuations—inevitably inflates the overall variance of the error distribution.

To isolate the model's predictive stability from inherent biological inter-variable differences, we conducted a further stability analysis. We analysed the MASE variance for the top 30 most time-varying variables on an individual, per-variable basis across all indications.

As illustrated in Figure 17, when controlling for the specific variable, the model demonstrates high stability:

- The median IQR across indications drops to $0.085$.

- The largest IQR bucket is between $0$ and $0.05$.

- Only 2 variables exhibited an IQR exceeding $0.4$.

These findings confirm that while aggregating diverse clinical variables yields a wider error distribution, the model's underlying predictions for specific, individual biomarkers remain relatively stable and consistent across different patient cohorts.

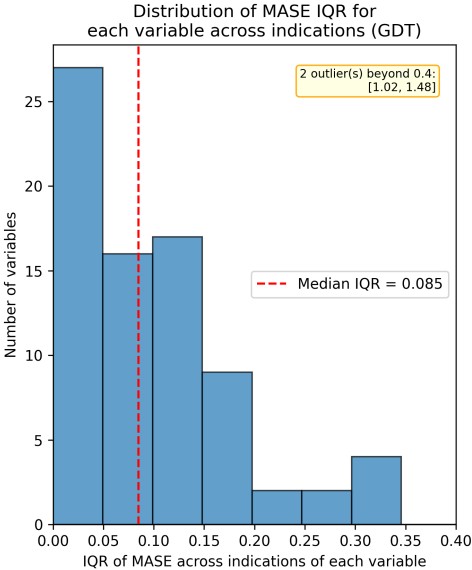

*Figure 17.* **Stability analysis of MASE variance for the top 30 most time-varying clinical variables.** When evaluated individually across indications, the vast majority of variables exhibit highly stable predictions (IQR < 0.05), demonstrating that the broader reported variance is largely an artifact of cross-variable aggregation.

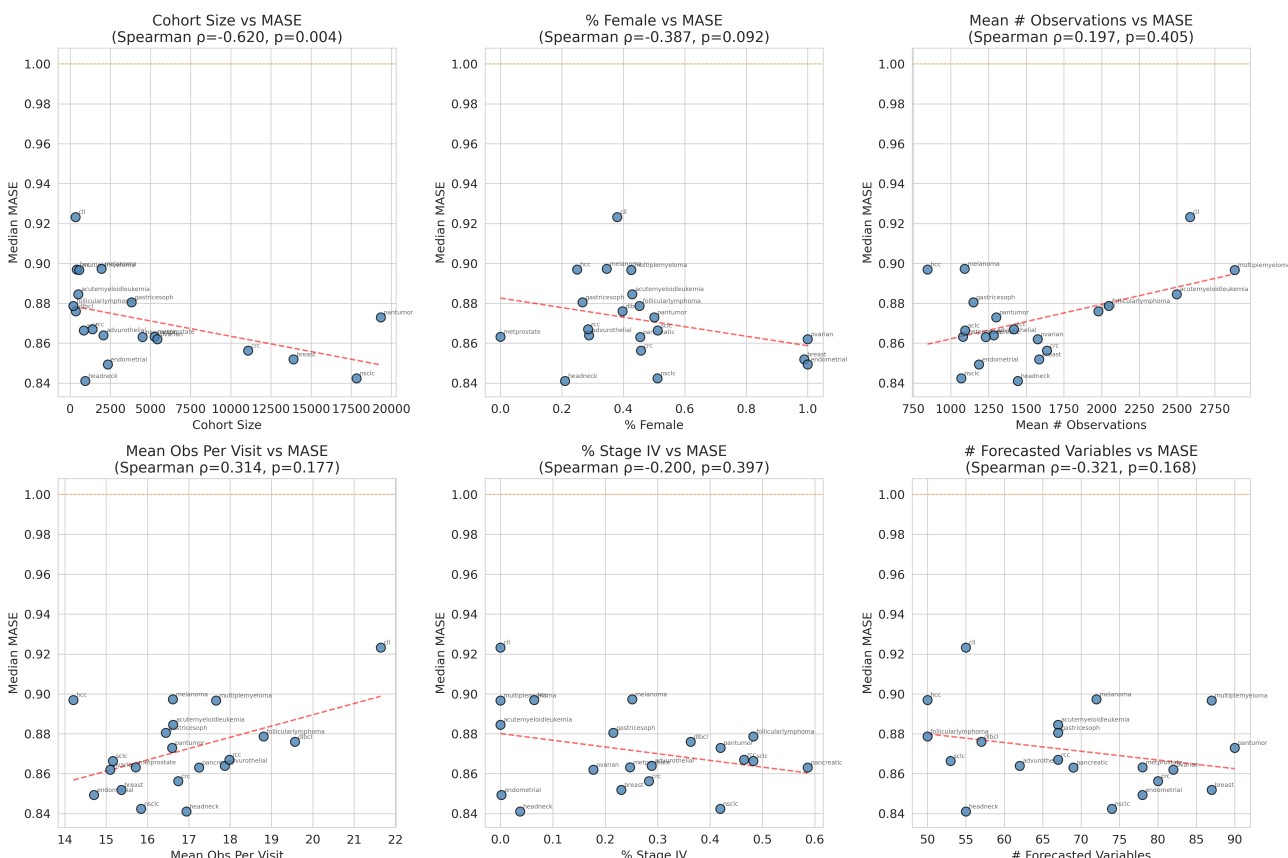

*Figure 18.* **Correlation of clinical and data features with forecasting error, measured by median MASE.**

### D.17. Analysis of Failure Modes

To provide a granular understanding of GDT's performance boundaries, we investigated the relationship between clinical characteristics, data density, and forecasting error (MASE).

**Correlation of Clinical and Data Features:**

We analyzed the Spearman correlation between median MASE and several key cohort features across all indications (Figure 18). We found a statistically significant negative correlation between cohort size and error ($\rho = -0.620$, $p = 0.004$), suggesting that smaller cohort sizes often result in worse performance and supporting the necessity of pan-cancer transfer learning. We also observed that a higher mean number of observations per visit correlates with higher MASE, suggesting that highly complex, data-dense patient records present modeling challenges. Conversely, a higher percentage of Stage IV cancer patients within a cohort correlated with a lower MASE, though this trend was not statistically significant. We also observed a negative correlation between the total number of forecasted variables per indication and median MASE (Spearman $\rho = -0.321$), consistent with multi-task learning benefits and suggesting that broader joint forecasting may further reduce error as context-length constraints relax.

**Extreme Variable-Specific Failures:** We identified specific clinical variables that systematically resulted in extreme forecasting failures (defined as MASE > 2). Notably, the most severe failure modes clustered around thyroid function markers:

- **Thyroxine free (LOINC 3024-7):** Failed extremely in 8 out of 8 indications where it appeared, reaching a maximum MASE of 8.2.

- **Thyrotropin/TSH (LOINC 3016-3):** Failed in 4 out of 11 indications, reaching a maximum MASE of 13.0.

- **Thyroxine/T4 (LOINC 3026-2):** Exhibited the highest overall mean MASE of 5.95 across the dataset.

This clustering suggests these variables are subject to biological complexities or measurement inconsistencies in routine EHR data that are not fully captured by our framework.

### D.18. Discussion: Improvement of LLMs over Specialized Time-Series Models

The improved performance of LLMs over specialized time-series models is a counterintuitive finding that warrants explicit discussion. We believe that we can attribute this performance gap to three primary factors:

- LLM Prior Knowledge: Pre-trained models provide generalized priors, whereas specialized architectures require retraining from scratch,

- Effective Scale: Our text-based formulation enables transfer learning across diverse pan-cancer cohorts (Appendix D.5),

- Sparsity: Natural language allows for the integration of full-patient histories, without any explicit imputation required for sparse EHR data, allowing the model to easily capture non-linear patterns between events, especially those which are rare.

### D.19. Compute and Wall-Time Comparison

LLM-based approaches like GDT are inherently more compute-intensive than specialized baselines. We report empirical wall-times on the clinical trial datasets to characterize this overhead. Despite higher absolute costs, per-patient inference remains under one minute, making the framework deployable in clinical and trial settings (locally on a single B200 GPU).

As LLM inference acceleration advances, this framework will benefit directly since it runs on standard LLM infrastructure.

*Table 8.* Training and inference wall-times (minutes) on clinical trial data.

| Task | Model | Training | Inference |
|------|-------|----------|-----------|
| Events | RSF | 1.8 | <1 |
| Events | CLMBR-T (rep+head) | <1 | 2.1 |
| Events | CLMBR-T (scratch) | 7.8 | 2.1 |
| Events | MOTOR | 25.8 | 13.0 |
| Events | Llama (FT scratch) | – | 68.0 |
| Events | II-Medical (FT scratch) | – | 70.9 |
| Events | GDT (Ours) | 154.4 | 68.5 |
| Forecast | TiDE (uni) | 23.2 | 2.1 |
| Forecast | TiDE (all) | 24.0 | 2.2 |
| Forecast | Chronos Bolt | – | 2.2 |
| Forecast | Chronos | 109.2 | 13.9 |
| Forecast | Llama | – | 127.5 |
| Forecast | II-Medical | – | 371.8 |
| Forecast | GDT (Ours) | 331.0 | 153.9 |

## D.20. Cross-Dataset Generalization via the MEDS Standard

To demonstrate that the TwinWeaver pipeline is not tied to the oncology cohort or to clinical trial data, we provide an end-to-end tutorial applying the framework to the public MIMIC-IV MEDS demo dataset, an intensive-care EHR resource with a markedly different patient population, vocabulary, and event cadence than the FH-FMI CGDB. The tutorial reuses the same serialization, prompt-construction, and sampling components described in Section 3 without any oncology-specific code paths, and ingests data directly in the emerging Medical Event Data Standard (MEDS) format. Because all clinical events are rendered as natural language rather than mapped to a fixed code vocabulary, adapting to MIMIC-IV requires no vocabulary alignment, embedding-table surgery, or new prediction heads, in contrast to fixed-vocabulary EHR foundation models such as CLMBR-T and MOTOR. The tutorial, together with the MEDS adapter, is released alongside the framework code and is intended as a reference for applying TwinWeaver to further EHR sources such as EHRShot and eICU.

## D.21. Framework Limitations

TwinWeaver's text-based representation expands sequence lengths substantially compared to fixed-vocabulary EHR models such as CLMBR-T and MOTOR, since each clinical event is encoded as multiple natural-language tokens rather than a single code, which incurs higher compute costs for both training and inference (Appendix D.19). To fit within context limits, we truncate intermediate visits while preserving the first visit and dynamically sample a subset of output variables per training instance. This restricts the breadth of variables forecast jointly at inference, although the vocabulary itself remains unconstrained and the model can still generate any prompted target. Additional methodological choices carry their own trade-offs: treating treatment switching as censoring follows standard real-world data practice but may discard informative signal in oncology, weekly aggregation by mean and mode can blur short-lived acute events, and 3-sigma filtering, while necessary to suppress EHR artifacts, risks removing clinically meaningful extremes such as severe lab abnormalities. Each of these choices was validated empirically in our experiments but warrants further systematic study.

## D.22. Differentiation from GenHPF

GenHPF (Hur et al., 2023) is a close prior work to TwinWeaver in spirit, however, the two frameworks differ in three substantive respects. First, in architecture: GenHPF trains discriminative classification heads on top of EHR embeddings, whereas TwinWeaver retains the LLM's generative next-token objective end-to-end, enabling direct natural-language outputs over an unconstrained vocabulary. Second, in task scope: GenHPF targets predefined binary classification outcomes, while TwinWeaver jointly handles continuous biomarker forecasting and discrete time-to-event prediction within a single unified model, supporting arbitrary on-demand horizons. Third, in interpretability: the generative design of TwinWeaver allows the same model to be extended with reinforcement-learning-aligned rationales (Section C.2.2), a capability not supported by classification-head architectures. Together, these differences broaden the scope from outcome classification to unified longitudinal modeling with hypothesis-generating reasoning.

