# OpenReview forum: "TwinWeaver: An LLM-Based Foundation Model Framework for Pan-Cancer Digital Twins"
_ICML.cc/2026/Conference — ICML 2026 regular_

### Official Review · Reviewer_a5j1 · 2026-03-12

**Soundness:** 3
**Presentation:** 3
**Significance:** 3
**Originality:** 3
**Overall Recommendation:** 4
**Confidence:** 4

**Summary:**

This study introduces TwinWeaver, an open-source framework that serializes longitudinal patient histories into text, enabling unified event prediction as well as forecasting with large language models, and uses it to build Genie Digital Twin (GDT) on 93,054 patients across 20 cancer types. The validation results showed that the proposed model achieved the best performance compared with existing models.

**Compliance With Llm Reviewing Policy:**

Affirmed.

**Key Questions For Authors:**

Are there failure models on some patients (more errors than other patients)?
Are there more data to evaluate the metastasis prediction?

**Limitations:**

no obvious limitation.

**Strengths And Weaknesses:**

Strengths:
New model application on pan-cancer cohort. It achieved high prediction accuracy, and can reason the prediction (possible explainable)

Weaknesses:
It is unclear if there are failure models on some patients (more errors than other patients).
Are there more data to evaluate the metastasis prediction?

---

> ### Author Rebuttal · Authors · 2026-03-30
>
> We greatly appreciate the reviewer's positive feedback and recognition of the model's accuracy, pan-cancer applicability, and explainability features.
>
> **Analysis of Failure Modes**
> To provide a more granular understanding of TwinWeaver's performance boundaries, we have performed an analysis of failure modes. We investigated the relationship between clinical characteristics and forecasting error (MASE), as well as specific biochemical markers that pose significant modeling challenges.
>
> **1. Correlation of Clinical and Data Features with Performance**
> We analyzed the Spearman correlation between median MASE and several key features across all indications. The following trends were observed (https://i.ibb.co/vCScq7ZG/scatter-features-vs-mase.png):
>
> * **Cohort Size:** There is a statistically significant correlation ($\rho = -0.620, p = 0.004$) suggesting that smaller cohort sizes often result in worse performance, supporting the hypothesis for needing pan-cancer learning.
> * **Data Density vs. Error:** We found that a higher mean number of observations and a higher mean number of observations per visit both correlate with higher MASE.
> * **Clinical Severity:** We observed that a higher percentage of Stage IV cancer patients within a cohort correlates with lower MASE. However, this trend is not statistically significant and should be interpreted cautiously.
> * **Multi-task Scaling:** An increase in the total number of forecasting variables led to a slight decrease in MASE, albeit with high variance.
>
> **2. Analysis of Extreme Variable-Specific Failures**
> We identified specific variables that systematically resulted in extreme failure (MASE > 2):
>
> * **Thyroxine free (LOINC 3024-7):** Failed extremely (MASE > 2) in 8 out of 8 indications where it appeared, reaching a maximum MASE of 8.2.
> * **Thyrotropin/TSH (LOINC 3016-3):** Failed in 4 out of 11 indications, with a maximum MASE of 13.0.
> * **Thyroxine/T4 (LOINC 3026-2):** Exhibited the highest overall mean MASE of 5.95.
>
> **Conclusion on Failure Modes:**
> Overall, these findings suggest that model performance degrades primarily in data-sparse or clinically complex settings, rather than uniformly across patients. Interestingly, all of the most extreme failure modes are related to thyroid function. This clustering suggests that these variables may be subject to biological complexities or specific measurement inconsistencies in real-world EHR data that are not fully captured by our current temporal window or aggregation strategy. We will add a discussion of these to the manuscript.
>
>
> **Metastasis Prediction Data**
> Unfortunately, we did not have access to further datasets with metastasis. Metastasis occurrence is not consistently coded or measured at regular intervals in routine real-world clinical practice, compared to standard endpoints such as overall survival or treatment switching. We intentionally retained this task in the manuscript to transparently showcase the low-data boundary and failure modes of the TwinWeaver architecture. We will update the manuscript to explicitly state that applying generative models to highly sparse, complex biological events such as metastasis remains an open challenge, potentially requiring richer multimodal data (e.g., imaging) beyond standard EHR text serialization.

---

> > ### Author Rebuttal · Reviewer_a5j1 · 2026-04-02
> >
> > The additional analysis is appropriate for answering the comments.

---

### Official Review · Reviewer_f4U1 · 2026-03-12

**Soundness:** 3
**Presentation:** 3
**Significance:** 3
**Originality:** 3
**Overall Recommendation:** 4
**Confidence:** 4

**Summary:**

This work proposes a new strategy for applying large language models to clinical modeling. Unlike previous approaches that directly train specialized foundation models on structured raw data, the proposed method first converts multimodal clinical data into textual corpora and then leverages language models for training, enabling a simpler and more general way to utilize existing data for pretraining. Furthermore, because the model is built on a large language model, it naturally supports interactive feedback and explanation through natural language.

**Compliance With Llm Reviewing Policy:**

Affirmed.

**Key Questions For Authors:**

1. In your experiments, GDT is compared only with Time-Series Forecasting Foundation Models from 2023–2024, but not with Clinical & EHR Foundation Models. Could you clarify the reason for not including this comparison? Providing such a comparison could potentially affect the assessment of GDT’s advantages in practical applications.
2. In the Metastasis prediction task, GDT is among the lowest-performing methods (Fig. 3d), and its MASE variance is relatively large when predicting the top 30 most time-varying variables. Could you provide additional analysis on model stability or variance? Such results could influence the evaluation of the model’s overall technical reliability.
3. During training, the paper formulates all prediction tasks as a next-token prediction problem, which may not be an ideal approach for precise continuous-value forecasting. Have you considered adding a loss function specifically designed for accurate numerical prediction to improve prediction accuracy? If this approach could enhance the predictive precision of GDT, it would further support the technical reliability of the method.

**Limitations:**

Yes.

**Strengths And Weaknesses:**

This paper presents a novel approach that converts electronic health record (EHR) data into textual representations to train a large language model (LLM) for longitudinal clinical time-series modeling, offering an innovative extension of LLMs beyond structured data. The paper is generally well-written and well-structured.

However, several weaknesses temper the soundness of the evaluation. The proposed method does not consistently achieve state-of-the-art performance; notably, it ranks among the two worst-performing approaches in the Metastasis prediction task (Fig. 3d). While the model achieves the best Mean Absolute Scaled Error (MASE) in another task, its relatively large variance raises concerns about stability. The authors are encouraged to conduct additional experiments to compare variances, as more stable results would more convincingly demonstrate the method's advantages. Furthermore, the baseline comparisons primarily include models from 2023 and 2024, omitting more recent state-of-the-art time-series and clinical foundation models. The authors should clarify why models from the Clinical & EHR Foundation Models category, discussed in the related work, were not included in the experimental comparison. Additionally, formulating all prediction tasks as next-token prediction may not be optimal for precise numerical forecasting, as this objective might not capture the sensitivity required for continuous value changes. The counterintuitive finding that LLMs outperform specialized time-series models also warrants further discussion.

Despite these concerns, the work holds practical value and originality by providing a new perspective on applying LLMs to longitudinal clinical data. Addressing the experimental limitations and providing clearer justification for design choices would significantly strengthen the paper.

---

> ### Author Rebuttal · Authors · 2026-03-30
>
> We thank the reviewer for their assessment and for recognizing the practical value and originality of our framework.
>
> # Comparison to EHR Foundation Models (Q1)
>
> ## Further Event Baselines
> We appreciate the suggestion to include recent clinical & EHR foundation models. We will further clarify that we compared GDT against the EHR foundation model CLMBR-T in the event evaluations (Figures 3 and 4). Additionally, we have now also benchmarked on clinical trial data GDT against the MOTOR model and a version of CLMBR-T trained from scratch on the specific clinical trial vocabulary. Due to the time constraints of the rebuttal period, we were unable to integrate the most recently published 2025 models (pending access), nevertheless, we believe our inclusion of MOTOR and CLMBR-T serves as a comparable proxy for the current state-of-the-art (please also see response Q4 to reviewer SH8J for further comparison). GDT-SFT outperforms both specialized architectures, averaged across all event types and clinical trials:
>
> | Model | Training Setup | Overall Avg C-Index |
> | :--- | :--- | :--- |
> | **GDT (Ours)** | **SFT** | **0.6726** |
> | RSF | Grid Searched (New) | 0.6524 |
> | RSF | Default | 0.6480 |
> | MOTOR | Pretrained + CoxPH Head (New) | 0.6117 |
> | CLMBR-T | Pretrained + CoxPH Head | 0.6098 |
> | CLMBR-T | Trained from Scratch (New) | 0.5966 |
>
>
> ## Forecasting
> For time-series forecasting, we did not compare to other common EHR foundation models, such as CLMBR-T or MOTOR, since they were primarily focused on next event prediction or time-to-event estimation. However, to the best of our knowledge, they do not provide a direct extension towards time-series forecasting, or a conditional generation approach, allowing us to forecast the numerical values at certain time points.
>
> **LLMs vs. Specialized Time-Series Models**
> We agree that the superior performance of LLMs over specialized time-series models is a counterintuitive finding that warrants explicit discussion. We will update the manuscript to attribute this performance gap to three primary factors:
> 1. **LLM Prior Knowledge:** Pre-trained models provide generalized priors, whereas specialized architectures require retraining from scratch,
> 2. **Effective Scale:** Our text-based formulation enables transfer learning across diverse pan-cancer cohorts (Appendix D.5),
> 3. **Sparsity:** Natural language allows for the integration of full-patient histories, without any explicit imputation required for sparse EHR data, allowing the model to easily capture non-linear patterns between events, especially those which are rare.
>
> # Variance, Stability, and Metastasis (Q2)
> We agree with the reviewer that further variance analysis is needed. We will clarify in the manuscript that the IQR values in the forecasting figure are aggregated across variables for a given indication, increasing variance due to biological differences between the variables.
>
> To address concerns about stability for the same variable, we analyzed the MASE variance for the top 30 most time-varying variables aggregated across all indications, for each variable individually. In the attached variable-level stability analysis (https://i.ibb.co/F4VYn3yG/mase-iqr-distribution-gdt-top-30-same-vars-across-indications.png), the median IQR across indications is 0.085, most variables fall in the 0-0.05 bucket, and only 2 variables exceed 0.4.
>
> Regarding the Metastasis task: we included this to transparently showcase a failure mode of the model. Metastasis is inconsistently measured in Real-World Data and presents high biological complexity. When combined with a low data volume for this specific endpoint, the model struggles.
>
> # Next-Token Prediction for Continuous Values (Q3)
> We thank the reviewer for the numerical idea. We did not add a separate numerical loss in the main GDT model, since the design goal was a single next-token objective shared by forecasting and event prediction. We agree that hybrid token and numeric objectives are a promising direction and we will add this to the future work section. As a first step toward numerically grounded training, our reasoning extension uses a negative-MAE reward.

---

> > ### Author Rebuttal · Reviewer_f4U1 · 2026-04-04
> >
> > After reviewing the authors' response, I appreciate their thorough and constructive replies. They have adequately addressed my concerns regarding baseline comparisons (Q1) by adding experiments with MOTOR and CLMBR-T, and provided a reasonable stability analysis for variance concerns (Q2). However, my concern about using next-token prediction for continuous numerical forecasting (Q3) remains only partially addressed, as the proposed numerical loss function was not incorporated into the current model. While I acknowledge the authors' plan to explore this direction in future work, the current evaluation of numerical prediction accuracy could be strengthened.

---

> > > ### Author Response · Authors · 2026-04-05
> > >
> > > We sincerely thank the reviewer for the continued constructive dialogue and for acknowledging our added baselines and stability analysis.
> > >
> > > # Further Evaluation
> > > We agree that the counter-intuitive success of next-token prediction for continuous values requires robust, multifaceted evaluation. To strengthen our current evaluation of numerical accuracy, we will add a dedicated section to the manuscript that combines several new granular analyses:
> > >
> > > * Overall Improvement: As noted, GDT exhibits statistically significant improvements over the strongest specialized forecasting baselines (including TiDE, which uses a numeric loss) in both Real-World Data (20 indications, $p < 0.001$) and unseen clinical trials ($p < 0.05$).
> > >
> > > * Failure Mode Analysis: We conducted a new failure mode analysis correlating clinical characteristics with forecasting error. We found that larger cohort sizes (Spearman $\rho = -0.620$) and a higher number of total forecasted variables (Spearman $\rho = -0.321$) both correlate with lower Mean Absolute Scaled Error (https://i.ibb.co/vCScq7ZG/scatter-features-vs-mase.png).
> > >
> > > * Variable-Level Stability: To explore the stability of the accuracy, our new variable-level stability analysis confirms that across the top 30 most time-varying variables, the median IQR of the error is concentrated (0.085), suggesting the model does not suffer from extreme variance on a per-variable basis.
> > >
> > > * Continuous Smoothing via Ensembling: For continuous-value sensitivity, we will clarify that GDT does not rely on a single discrete token path. At inference, GDT generates M independent textual completions that are decoded into numerical trajectories and averaged (Section 3.2.1). This ensemble decoding acts as a smoothing mechanism, reducing the variance inherent in autoregressive discrete sampling and yielding more stable numerical outputs.
> > >
> > > By adding this combined stability and failure-mode analysis, we provide a more rigorous evaluation of where the next-token numerical predictions succeed and where they reach their limits.
> > >
> > > # Proof-of-concept of Numerical Loss via GRPO
> > > We agree that relying solely on cross-entropy may not capture the fine-grained sensitivity required for continuous values. As a first step toward numerically grounded training, we explored incorporating a direct numerical loss function during our reasoning experiments (NSCLC neutrophils forecasting; Sec. 5.4).
> > >
> > > Specifically, we found that relying solely on supervised fine-tuning (SFT) to distill text-based reasoning from the teacher model caused a slight degradation in forecasting precision (MASE 0.863 vs 0.828 base). We then applied Group Relative Policy Optimization (GRPO) using a deterministic reward function acting as a surrogate numerical loss, tied directly to the negative Mean Absolute Error (MAE), and capped to reduce extreme prediction effects, with $r$ being the reward function, $\hat{y}$ is the prediction decoded from textual prediction, $y$ the target time series, and $N=|\hat{y}|$:
> > >
> > > $$r=-\min(\frac{1}{N}\sum_{i=1}^N |\hat{y}_i-y_i|, 20)$$
> > >
> > > While incorporating this continuous reward function stabilized the degradation (recovering the MASE slightly from 0.863 to 0.862), we agree with the reviewer's assessment: this marginal difference highlights how non-trivial it is to design optimized numerical loss/reward functions for complex EHR data. We will add these exact results to the manuscript, and additionally explicitly frame it as an experiment that suggests the viability of numerical objectives, motivating dedicated future research.
> > >
> > > # Future Work
> > > We strongly agree with the reviewer that explicitly integrating numerical loss functions is a highly promising direction, particularly to reduce stochasticity and improve precision in continuous clinical forecasting alongside reasoning. While these initial reasoning experiments highlight the viability of numerical objectives, scaling a hybrid token/numeric loss architecture across the entire pan-cancer foundation model requires extensive evaluation and compute time beyond what is available in the rebuttal period. We will explicitly highlight this as a limitation and prioritize it in our expanded Future Work section.

---

### Official Review · Reviewer_Dev4 · 2026-03-12

**Soundness:** 2
**Presentation:** 2
**Significance:** 3
**Originality:** 2
**Overall Recommendation:** 3
**Confidence:** 4

**Summary:**

This paper proposes to transform patient longitudinal health history into text and then fine-tune LLMs for tasks such as biomarker forecasting and event prediction (survival, progression, therapy switch and metastasis prediction). Genie digital twin model is used and trained on cohort of ~93K patents across multiple indicators. Empirical results show GDT performs better than chosen baselines. The model seems to perform reasonably well on out of distribution clinical trial data as well as zero shot setting.

It is a reasonable idea. However, authors make many strong claims which need to be toned down. Training samples are within 90 days of therapy as a result model is more likely;y learn from treatment transition rather entire disease history. Metastasis prediction is available in three indications and doesn't perform well.

**Compliance With Llm Reviewing Policy:**

Affirmed.

**Final Justification:**

I appreciate the additional experiments!

However, my central concerns remain. As noted by authors raw three class setup is not a full survival model and that the resulting probabilities still need cautious interpretation even after isotonic post processing,  Moreover, the analyses do not convincingly show that pan cancer pretraining provides any gain since Llama fine tuned from scratch is already quite close to GDT on event prediction, and the reasoning extension still worsens the primary metric. I keep my original score.

**Key Questions For Authors:**

How sensitive are the event results to the exact description of the three answer used in the softmax over completions procedure?


It is unclear what components contribute or drive predictions? Whether the gains come from the language-model prior, multimodal prompt design, larger effective training set, or choice of Llama-based fine-tuning?


Look at weaknesses.

**Limitations:**

Nothing novel about the method it is just an interesting end application to pan cancer.

Weak baselines, more validation is needed. Many over claims should be toned down.

**Strengths And Weaknesses:**

Strengths

A single unified approach that does both time series forecasting and event prediction which makes it a goof fit for clinical setting.

Training on real-world oncology cohort and inclusion of out-of -distribution clinical trail data is a good real world setting.


Weaknesses

My main concern is the formulation of event modeling is more of a hack and not a survival model. Prediction is a 3-class problem (occurred, not-occurred, and censored) and the score of occurred is used for ranking. This is not a time to event model and output is pretty much dependent on how text was tokenized and summary was generated. There is no guarantee that cumulative incidence curve is monotonic. So unclear to me whether predicted probabilities are clinically relevant scores.


In appendix author use ranking scores to estimate cumulative incidence curve by re-normalizing over non-censored outcomes and then running calibration. Here assumption is treatment switching is treated as a censoring or a competing event which is not really correct for oncology data where treatment changes are informative not random.


It is unclear whether performance gain is from architecture or scale. As noted by authors training GDT on a subset of 2000 patients outperformed baseline by gains are only noticeable when trained on entire dataset.


Baselines are weak. Authors use RSF with default hyper parameters and CLMBR-T is just embeddings used with elastic net Cox model. Paper does cite MOTOR and SCOPE but does not compare against them.

Unclear what is the rationale behind weekly aggregation by mean or mode as it can blur the contribution of meaningful acute events. Besides, three sigma filtering will remove extreme lab values that are more likely to be a factor for cancer.



It is unclear what events dominate the text description that is prompt length. Since authors explicitly truncate intermediate visit when context is long. But this means model is not using the whole record and relying on hand crafted summarization policy which could put more emphasis on recent events.

Reasoning results are overstated. Supervision signal in reasoning is explicitly conditioned on the correct answer.

Training samples are drawn from visits within 90 days of a new therapy so much is relying around treatment transition window not the entire history.  Besides, metastasis is only available for there indications and doesn't perform well.


Code is not included so open-source claim is a stretch.

Many acronyms are used earlier and introduced later.

---

> ### Author Rebuttal · Authors · 2026-03-30
>
> We thank the reviewer for their constructive feedback.
> # Strengthening Baselines
> We agree that robust baselines are critical. Lacking CGDB access due to licensing, new evaluations use clinical trial datasets. We added MOTOR, CLMBR-T (trained from scratch with vocabulary), and grid-searched RSF baselines (SCOPE was omitted due to data incompatibility & time restrictions):
>
> | Model | Training Setup | Overall Avg C-Index |
> | :--- | :--- | :--- |
> | **GDT (Ours)** | **SFT** | **0.6726** |
> | RSF | Grid Searched (New) | 0.6524 |
> | RSF | Default | 0.6480 |
> | MOTOR | Pretrained + CoxPH Head (New) | 0.6117 |
> | CLMBR-T | Pretrained + CoxPH Head | 0.6098 |
> | CLMBR-T | Trained from Scratch (New) | 0.5966 |
>
> # Source of Performance (Q2)
>
> ## 1. LLM Prior
> Isolating drivers, we evaluated Llama and II-Medical-8B-1706 LLMs fine-tuned (FT) from scratch, i.e. only on trial data. TwinWeaver improves performance for both.
>
> **Events:**
>
> | Foundation Model | Evaluation Setting | Overall Avg C-Index |
> | :--- | :--- | :--- |
> | **GDT (Llama 3.1)** | **SFT (Ours)** | **0.6726** |
> | Llama | FT from Scratch (New) | 0.6686 |
> | II-Medical | FT from Scratch (New) | 0.6603 |
> | GDT (Llama 3.1) | Zero-Shot | 0.6568 |
> | Llama | Zero-Shot | 0.5399 |
> | II-Medical | Zero-Shot (New) | 0.5388 |
>
> **Forecasting:**
>
> | Foundation Model | Evaluation Setting | POPLAR Median MASE (IQR) | IMpower130 Median MASE (IQR) |
> | :--- | :--- | :--- | :--- |
> | **GDT (Llama 3.1)** | **SFT (Ours)** | **0.8834 (0.0907)** | **0.7549 (0.3859)** |
> | II-Medical | Fine-tuned from scratch (New)| 0.8984 (0.1078) | 0.8209 (0.2870) |
> | Llama | Fine-tuned from scratch (New) | 0.9243 (0.1214) | 0.7782 (0.3676) |
> | GDT (Llama 3.1) | Zero-Shot | 0.9471 (0.1213) | 0.8748 (0.1631) |
> | Llama | Zero-Shot | 1.1641 (0.1759) | 1.1269 (0.1564) |
> | II-Medical | Zero-Shot (New) | 1.5320 (0.3373) | 1.2872 (0.7051) |
>
> ## 2. Effective Training Set
> Pan-cancer pretraining improves performance (Appendix D.5). Additionally, our MASE analysis confirms larger cohort sizes yield better performance (https://i.ibb.co/vCScq7ZG/scatter-features-vs-mase.png).
>
> ## 3. Prompt Design and Length
> Appendix D.6 shows prompt length and truncation effects. Including extended history vs. recent events improves 1-year survival prediction by 9.9%, supporting our truncation strategy. However, further studies are needed, and we will add this to the limitations.
>
> # Event Formulation Sensitivity and Monotonicity (Q1)
> The reviewer raised a valid question regarding prompt sensitivity. We performed an ablation for trial SFT by replacing the target vocabulary with different synonyms:
>
> | GDT-SFT Prompt Target Formulation | Overall Avg C-Index |
> | :--- | :--- |
> | **GDT+SFT:** "occurred" / "did not occur" / "censored." | **0.6726** |
> | **GDT+Ablation SFT:** "happened" / "did not happen" / "was censored." | 0.6509 |
>
> While there is an expected slight drop, since the model was before fine-tuned heavily on the original phrases, the ablated model maintains highly competitive clinical ranking.
>
> We agree the raw 3-class setup isn't a full survival model. Its uncalibrated score should be interpreted as a ranking tool that achieves improved IPCW C-indexes with unified next-token training, rather than a calibrated cumulative-incidence estimate. We will sharpen this distinction in the paper. Post-hoc isotonic regression enforces monotonicity and aids calibration, yet these probabilities still require cautious use (Appendix B.2.1, D.8).
>
> # Clarifications
> * **Weekly Aggregation:** We chose weekly aggregation to match the cadence of oncology therapies/labs and to avoid over-representing acute hospital noise. We agree it can blur short-lived acute events and will add that limitation (Jeon et al., 2021, 10.2196/25035).
> * **Treatment Switching:** We agree. While censoring switchers is standard RWD methodology, it overlooks informative changes (Collet et al., 2025, 10.1002/cpt.70013). We will explicitly add this to our limitations.
> * **3-Sigma Filtering:** While filtering was necessary to remove EHR artifacts, we agree it risks discarding true biological extremes relevant to cancer and we will add this to limitations.
> * **2k Ablation:** GDT trained on 2k patients is statistically significantly better than the best baseline (TiDE FT-Multi) with p < 0.001 (Wilcoxon signed-rank, new).
> * **90 day treatment focus:** We focus on the 90 days since this was deemed the most important by cancer specialists. TwinWeaver allows arbitrary split times, depending on the task.
> * **Reasoning:** Only the teacher model is conditioned on the true answer to generate targets. During GRPO and final evaluation, GDT generates reasoning based only on the input history.
> * **Open Source:** Anonymous code is in the Supplementary Material zip file; a public link will be added if accepted.
> * **Acronyms:** We thank the reviewer for highlighting the acronym issue and will add all acronym definitions to their first use.
>
> All points above will be addressed with manuscript revisions.

---

> > ### Author Rebuttal · Reviewer_Dev4 · 2026-04-03
> >
> > I thank the authors for the additional experiments and detailed response. I still see number of issues:
> >
> > a) GDT vs grid search RSF shoes a delta of 0.02 C-index. This marginal gain over simple baseline shows limited value.
> >
> > b) Llama FT from scratch nearly matches GDT SFT which in my understanding suggests that pan cancer pretraining provides no added value to event prediction.
> >
> > c) Event formulation still seems a hack to me. For example, i a synonym swap causes a 3% relative degradation, the model is learning tokenization artifacts, not risk structure.
> >
> > d) Reasoning degrades the primary task.  MASE changes from 0.828 to 0.862. A feature that makes predictions worse while generating post-hoc rationalizations distilled from a teacher conditioned on ground truth doesn't seem like a compelling contribution.
> >
> > The rebuttal adds new results but doesn't resolve the core weaknesses.

---

> > > ### Author Response · Authors · 2026-04-04
> > >
> > > We thank the reviewer for their thoughtful engagement. We acknowledge that our earlier explanation may not have fully conveyed the context of these results within the broader evaluation. We will revise the manuscript to more clearly describe the evaluation setting, assumptions, and limitations to ensure the results are appropriately contextualised.
> > >
> > > # Comprehensive Performance vs. RSF (Concern A)
> > > The 0.02 C-index delta was observed only in the clinical trial evaluation, not in the other 3 evaluation settings. More importantly, GDT still outperforms all baselines in all evaluated settings, except for metastasis prediction on RWD. Clinical trials represent a cold-start setting with limited longitudinal history, which is where static models like RSF excel.
> > >
> > > While RSF is a highly effective baseline for single tasks in cold-start clinical trials, focusing solely on this specific setting may obscure the broader advantages of the TwinWeaver framework: deep multi-task, longitudinal modeling. In Real-World Data (RWD), where longitudinal history is available, GDT demonstrates a statistically significant improvement over RSF ($p_{adj} < 0.05$) across Survival, Progression, and Therapy Switching. Furthermore, unlike RSF, GDT jointly enables continuous forecasting, significantly outperforming the strongest time-series baseline (TiDE) in both RWD ($p < 0.001$) and clinical trials ($p < 0.05$). GDT performs all these tasks simultaneously better than the evaluated baselines.
> > >
> > > # The Value of Pan-Cancer Pretraining (Concern B)
> > > While base LLMs are indeed strong at event prediction when fine-tuned with TwinWeaver, the pan-cancer pretraining model has the highest performance. Crucially, it enables zero-shot event prediction, matching trained baselines without updates, and substantially improves joint forecasting (e.g. POPLAR MASE 0.9243 vs 0.8834).
> > >
> > > Furthermore, pan-cancer pretraining is what unlocks cross-cohort transfer learning. As shown in Appendix D.5, it consistently improves forecasting for indications with low patient counts (e.g., Follicular Lymphoma, DLBCL, and CLL) compared to models trained solely on those indications.
> > >
> > > # Event Formulation and Tokenization (Concern C)
> > > Our text-based survival formulation is a deliberate architectural choice to unify clinical forecasting without altering the LLM's core causal language modeling loss.
> > >
> > > To contextualize the 3% relative performance drop (C-index 0.6509) observed when introducing synonyms, we emphasize that this ablated model remains highly competitive; it matches the performance of grid-searched RSF (0.6524) while maintaining a margin over both default RSF (0.6480) and CLMBR-T (0.6098). To explain the performance drop, the model was pretrained on 2.49 million samples using specific target phrases. Swapping these phrases with a short SFT on clinical trials naturally causes a slight performance drop due to token distribution shift.
> > >
> > > Crucially, our experiments suggest the model is learning clinical risk structure, not tokenization artifacts. As shown in our "from scratch" experiments, entirely different base models with fundamentally different tokenizer libraries (Llama vs. II-Medical/Qwen) still achieve highly competitive event prediction C-indices (0.6686 and 0.6603, respectively) when trained on the clinical trial data. If the performance relied solely on arbitrary token patterns, we would not see such cross-architecture generalization.
> > >
> > > Furthermore, our isotonic regression post-processing (Appendix B.2,D.8) guarantees monotonicity and improves calibration, helping bridge the gap between raw LLM outputs and valid cumulative incidence functions.
> > >
> > > # Reasoning and Task Trade-offs (Concern D)
> > > The knowledge distillation from a teacher model conditioned on ground truth was a relatively short, initial SFT step to establish formatting, following standard approaches (Shao et al., 2024, arXiv 2402.03300). The majority of the training was performed in the Reinforcement Learning phase (GRPO), tying the negative mean absolute error to the reward function.
> > >
> > > Crucially, Figure 6 demonstrates these are not disconnected post-hoc rationalizations. Our analysis suggests that the generated rationales mechanistically align with established biology and the model's own numerical outputs. For example, when the rationale identifies "Marrow Suppression," it correlates with lowered neutrophil trajectories, and correctly associates high TMB with immunotherapy.
> > >
> > > While this interpretable extension incurs a slight MASE degradation (0.828 to 0.862), the predictions remain highly competitive. In medicine, black-box forecasting is a primary barrier to adoption. Trading a fraction of raw accuracy for transparent, biological hypothesis generation is a step toward building clinician trust.

---

### Official Review · Reviewer_SH8J · 2026-03-17

**Soundness:** 3
**Presentation:** 4
**Significance:** 3
**Originality:** 2
**Overall Recommendation:** 5
**Confidence:** 4

**Summary:**

This paper focuses on the problem of forecasting with clinical time-series. These models can be used for risk stratification of patients which influences timely care. The authors propose a pipeline for converting EHR data to text that we can finetune LLMs on for forecasting. This is interesting because (a) perhaps LLMs learn useful forecast priors that we can take advantage of (b) this can scale to multi-institutional/dataset models (c) there maybe more information in a text representation of EHR history as opposed to discretizing events. However, this is challenging as EHR sequences are quite long and text tokenization magnifies this issue. Past work in fact has proposed text representations of EHR data [1] and showed ways to efficiently do so via hierarchical models, however this work did not train generative zero-shot models, I believe it trained on specific binary classification tasks in a multi-institutional setting. This work does analysis of the zero-shot generative setting and looks into out-of-distribution performance on clinical trials data, showing zero-shot capabilities on par with fine-tuned baselines, and superior finetuned performance (although there is very high variance in the Blood Biomarker Forecasting in Figure 4 (d). There also is a breadth of literature in the Zero-shot EHR generation domain, and this work needs to do defend it's novelty compared to this literature [2,3,4,5,6,7,8,9]. This work has some limitations in computational scalability that it needs to discuss, the authors have to restrict the types of observations they can generate for example due to sequence length limitation. Other Zero-shot approaches limit the vocabulary size too but there must be some discussion of the different tradeoffs when it is treated as text as opposed to previous vocabulary definition approaches.


[1] Hur, K., Oh, J., Kim, J., Kim, J., Lee, M. J., Cho, E., ... & Choi, E. (2023). Genhpf: General healthcare predictive framework for multi-task multi-source learning. IEEE Journal of Biomedical and Health Informatics, 28(1), 502-513.
[2] Waxler, S., Blazek, P., White, D., Sneider, D., Chung, K., Nagarathnam, M., ... & Shah, R. (2025). Generative medical event models improve with scale. arXiv preprint arXiv:2508.12104.
[3] Renc, P., Jia, Y., Samir, A.E. et al. Zero shot health trajectory prediction using transformer. npj Digit. Med. 7, 256 (2024
[4] Yang, Z., Mitra, A., Liu, W. et al. TransformEHR: transformer-based encoder-decoder generative model to enhance prediction of disease outcomes using electronic health records. Nat Commun 14, 7857 (2023).
[5] Foresight—a generative pretrained transformer for modelling of patient timelines using electronic health records: a retrospective modelling study. Kraljevic, Zeljko et al. The Lancet Digital Health, Volume 6, Issue 4, e281 - e290
[6] Theodorou, B., Xiao, C. & Sun, J. Synthesize high-dimensional longitudinal electronic health records via hierarchical autoregressive language model. Nat Commun 14, 5305 (2023).
[7] Redekop E, Wang Z, Kulkarni R, Pleasure M, Chin A, Hassanzadeh HR, Hill BL, Emami M, Speier WF, Arnold CW. Zero-shot medical event prediction using a generative pretrained transformer on electronic health records. J Am Med Inform Assoc. 2025 Dec 1;32(12):1833-1842. doi: 10.1093/jamia/ocaf160. PMID: 41060255; PMCID: PMC12646381.
[8] McDermott, M. MEDS "Everything-is-code" Autoregressive Model [Computer software].
[9] Pawel Renc, Michal K Grzeszczyk, Nassim Oufattole, Deirdre Goode, Yugang Jia, Szymon Bieganski, Matthew B A McDermott, Jaroslaw Was, Anthony E Samir, Jonathan W Cunningham, David W Bates, Arkadiusz Sitek, Foundation model of electronic medical records for adaptive risk estimation, GigaScience, Volume 14, 2025, giaf107

**Compliance With Llm Reviewing Policy:**

Affirmed.

**Final Justification:**

The authors clarified the novelty of their work compared to GenHFP which also ingests time-sereis patient data as text. GenHFP only performs classification, while this work supports text generation that can be decoded into zero-shot forecasts or tte predictions. This pipeline is significantly less efficient than other proposed methods, but computational efficiency is orthogonal to advancing predictive performance and generality of models. Because this model operates on text they pool together 20 datasets to train on, demonstrating how easily the model can extend to various medical code names, and they even apply it at inference on out of distribution datasets achieving high performance. This paper is highly performant against a wide selection of baselines, I raised my score to an accept.

Additionally this work propses a quite different approach to zero-shot forecasting compared to autoregressive approaches which predict the next observation. This work can do specific forecasts and tte predictions, which is novel compared to that line of work I provided citations for.

**Key Questions For Authors:**

(1) Defend the ease of use of the pipeline

(2) Discuss compute tradeoffs of text vs non-text tokenization for zero-shot generation.

(3) Clearly describe how generated __text__ trajectories are being averaged in section 3.2.1.

(4) Defend the novelty of this work agains the plethora of other zero-shot generative EHR approaches, as well as the existing models that operate on textified EHR time series data and have multi-instutional/dataset generalization. Consider adding them as baselines so you can argue you have a performance benefit over them.

**Limitations:**

__Compute Challenges__
Some patients can have hundreds of thousands of time series observations in the EHR, tokenizing this as raw text can significantly increase the number of tokens required to represent this already large sequence. The authors need to acknowledge this limitation or provide justification that their approach is computationally efficient despite this.


__Ease of Use__
How much preprocessing is expected/required by users of TwinWeaver? List these requirements as limitations.

**Strengths And Weaknesses:**

# Strengths

The paper is very well written, clear, and easy to follow. Notation is consistent. The forecasting evaluations are quite comprehensive as well. They achieve consistent discriminative performance improvements over the selected baselines. The authors even demonstrate out-of-distribution zero-shot performance on clinical trial datasets.

The code shared assumes that data in an event-stream format which is pretty common in the space. This is good, but it is not demonstrated on another dataset casting doubt on ease of use on different datasets.


# Weaknesses

__Novelty__
Other Zero-Shot methods that convert EHR data into tokens and evaluate zero-shot forecasting performance exist [2,3,4,5,6,7,8,9] and also provide open-source code that is easy to use, and provide pipelines for tokenizing EHR data and performing zero-shot generation. The authors need to defend the novelty of their work to these existing pipelines. There also exists this paper [1] which explictly converts EHR data into a sequence of text just as the authors claim to. [1] evaluates this on 3 datasets and demonstrates the actual motivation for using free-form text as the level of tokenization, to be able to train models on data from multiple-instutitions/datasets. They even show zero-shot performance (on binary classification) for a model trained on dataset A when used for inference on an unseen dataset B. They do not do zero-shot generative evaluations though, which would be a piece of novelty this paper could do. But as this paper currently stands it only evaluates on a single dataset and does not perform a multi-instution/dataset evaluation, a piece of analysis that would make this paper shine.

[1] Hur, K., Oh, J., Kim, J., Kim, J., Lee, M. J., Cho, E., ... & Choi, E. (2023). Genhpf: General healthcare predictive framework for multi-task multi-source learning. IEEE Journal of Biomedical and Health Informatics, 28(1), 502-513.
[2] Waxler, S., Blazek, P., White, D., Sneider, D., Chung, K., Nagarathnam, M., ... & Shah, R. (2025). Generative medical event models improve with scale. arXiv preprint arXiv:2508.12104.
[3] Renc, P., Jia, Y., Samir, A.E. et al. Zero shot health trajectory prediction using transformer. npj Digit. Med. 7, 256 (2024
[4] Yang, Z., Mitra, A., Liu, W. et al. TransformEHR: transformer-based encoder-decoder generative model to enhance prediction of disease outcomes using electronic health records. Nat Commun 14, 7857 (2023).
[5] Foresight—a generative pretrained transformer for modelling of patient timelines using electronic health records: a retrospective modelling study. Kraljevic, Zeljko et al. The Lancet Digital Health, Volume 6, Issue 4, e281 - e290
[6] Theodorou, B., Xiao, C. & Sun, J. Synthesize high-dimensional longitudinal electronic health records via hierarchical autoregressive language model. Nat Commun 14, 5305 (2023).
[7] Redekop E, Wang Z, Kulkarni R, Pleasure M, Chin A, Hassanzadeh HR, Hill BL, Emami M, Speier WF, Arnold CW. Zero-shot medical event prediction using a generative pretrained transformer on electronic health records. J Am Med Inform Assoc. 2025 Dec 1;32(12):1833-1842. doi: 10.1093/jamia/ocaf160. PMID: 41060255; PMCID: PMC12646381.
[8] McDermott, M. MEDS "Everything-is-code" Autoregressive Model [Computer software].
[9] Pawel Renc, Michal K Grzeszczyk, Nassim Oufattole, Deirdre Goode, Yugang Jia, Szymon Bieganski, Matthew B A McDermott, Jaroslaw Was, Anthony E Samir, Jonathan W Cunningham, David W Bates, Arkadiusz Sitek, Foundation model of electronic medical records for adaptive risk estimation, GigaScience, Volume 14, 2025, giaf107


__Ease of Use?__
The authors propose a pipeline for tokenizing EHR data into text. There is not a demonstration or justification of ease of use that this would provide. Even simply demonstrating results on a few more datasets would make it more convincing that this pipeline is easy to use or generalizes to other datasets.

__Compute__
Many papers in the EHR zero-shot space implicitly filter the data to a subset of codes/observables, as they are limited by sequence length. Your approach will strictly increase the sequence length as each observation instead of being represented as a single token in typical zero-shot pipelines, is now treated as a sequence of text tokens, this performance impact should be characterized. This paper actually has to reduce the set of output variables they can sample during generation to combat this. What is the impact on generative performance? Restricting the generated vocabulary during forecasting but not during training. Is the vocabulary subset for the output $V'$ also used for patient history $X_p(t)$? the restriction is clearly used for $Y_p$. The author's claim that this text approach has a more flexible vocabulary than CLMBR-T for example, while simultaneously restricting the model's vocabulary is a bit contradictory.


__Generation -> Classification Is Unclear__
The model averages the trajectories somehow to get a stable mean prediction. Section 3.2.1 Does not make sense. $\hat{Y}_p^j$ is a generated text trajectory. How are we able to compute an average of these? In the next part (3.2.2), it is $L_{y,p}$ supposed to just be computed over a single forecasted sequence that averages all of those text trajectories somehow?

__LLM Initialization sensitivity__
How sensitive are the results to different LLMs you initialize with? What about MEDGemma for example? The authors only use LLama 3.1 which is an LLM that is not trained on medical data as far as I am aware.

__Clinical Trial Experiment__
Why not compare CLIMBR-T in a supervised training setting on this clinical trial data, since the claim is that (right column line 356) is that the vocabular constraints make it underperform. You finetune your model, so why not train CLIMBR-T from scratch on this new clinical trials vocabulary, or even create a joint vocabulary between the oncology dataset and the clinical trial datasets so you can pretrain on the oncology dataset and finetune on the clinical trials datasets. That would be a more fair comparison.

---

> ### Author Rebuttal · Authors · 2026-03-30
>
> We sincerely thank the reviewer for the constructive feedback.
>
> # Novelty and Comparison to Existing EHR Models (Q4)
> We thank the reviewer for highlighting these baselines, particularly GenHPF [1]. We will clarify our Related Work to emphasize the novelty of GDT being the combination of flexibility to input types, enabling unified conditional forecasting and event prediction, as well as a reasoning extension. Furthermore, our clinical trial validation uses a distinct data source, demonstrating the requested zero-shot multiple dataset evaluation:
>
> > "... Further zero-shot EHR models have been proposed, including ETHOS [3] and [7] exploring zero-shot performance [3], Curiosity [2] and TransformEHR [4] demonstrating that such models scale to large datasets, HALO towards synthetic EHR generation [6], ARES [9] extending ETHOS to adaptive risk estimation, with [8] developing a generalized EHR code framework. However, these models rely on fixed, code-based vocabularies that constrains out-of-distribution adaptability (Bedi et al., 2026) and inclusion of further complex modalities, such as genetic data. Foresight [5] adds an NLP code extraction module for flexibility, but has limited ability for numerical forecasting. GenHPF [1] converts EHRs into text for cross-institutional adaptation, yet remains focused on classifying predefined outcomes rather than generative patient forecasting. These constraints underscore the need for unified forecasting and prediction models for heterogeneous, long-horizon clinical trajectories, capable of generalizing beyond their training distribution.
>
> >... TwinWeaver extends the domain adaptability of LLM-based models, providing a unified framework for joint continuous trajectory modeling and event prediction, with zero-shot generalization on out-of-distribution clinical trial datasets."
>
> # EHR Foundation Model Baselines & CLMBR-T from Scratch
> We evaluated MOTOR and CLMBR-T (trained from scratch on the trial vocabulary). Lacking CGDB access due to data licensing, all new ablations use the clinical trial datasets. Training CLMBR-T from scratch degraded performance slightly.
>
> | Model | Training Setup | Overall Avg C-Index |
> | :--- | :--- | :--- |
> | **GDT (Ours)** | **SFT** | **0.6726** |
> | RSF | Grid Searched (New) | 0.6524 |
> | RSF | Default | 0.6480 |
> | MOTOR | Pretrained + CoxPH Head (New) | 0.6117 |
> | CLMBR-T | Pretrained + CoxPH Head | 0.6098 |
> | CLMBR-T | Trained from Scratch (New) | 0.5966 |
>
> # LLM Initialization Sensitivity
> To explore base LLM sensitivity, we fine-tuned (FT) II-Medical-8B-1706 on the trial data (due to improved performance over MedGemma on public benchmarks). The TwinWeaver framework improves model performance across LLMs.
>
> **Event prediction:**
>
> | Foundation Model | Evaluation Setting | Overall Avg C-Index |
> | :--- | :--- | :--- |
> | **GDT (Llama 3.1)** | **SFT (Ours)** | **0.6726** |
> | Llama | FT from Scratch (New) | 0.6686 |
> | II-Medical | FT from Scratch (New) | 0.6603 |
> | GDT (Llama 3.1) | Zero-Shot | 0.6568 |
> | Llama | Zero-Shot | 0.5399 |
> | II-Medical | Zero-Shot (New) | 0.5388 |
>
> **Forecasting:**
>
> | Foundation Model | Evaluation Setting | POPLAR Median MASE (IQR) | IMpower130 Median MASE (IQR) |
> | :--- | :--- | :--- | :--- |
> | **GDT (Llama 3.1)** | **SFT (Ours)** | **0.8834 (0.0907)** | **0.7549 (0.3859)** |
> | II-Medical | Fine-tuned from scratch (New) | 0.8984 (0.1078) | 0.8209 (0.2870) |
> | Llama | Fine-tuned from scratch (New) | 0.9243 (0.1214) | 0.7782 (0.3676) |
> | GDT (Llama 3.1) | Zero-Shot | 0.9471 (0.1213) | 0.8748 (0.1631) |
> | Llama | Zero-Shot | 1.1641 (0.1759) | 1.1269 (0.1564) |
> | II-Medical | Zero-Shot (New) | 1.5320 (0.3373) | 1.2872 (0.7051) |
>
> # Ease of Use (Q1)
> We agree that this requires more emphasis. TwinWeaver features an integration module and tutorial with the emerging MEDS standard, making it deployable to many existing datasets. Additionally, we applied it to separate RWD and clinical trials.
>
>
>
>
> # Compute/Vocabulary Tradeoffs (Q2)
> The reviewer correctly notes the increased length of text tokenization. We mitigate this via middle-truncation, maintaining high performance (Appendix B.1.4 & D.6), but will expand on this in limitations.
>
> Regarding the vocabulary contradiction: GDT does not restrict the input vocabulary, since it uses natural language, allowing ingestion of any medical/genetic event. Restricting output variables is a computational sampling strategy to fit token limits, not a static architectural bottleneck like CLMBR-T's vocabulary. The model can generate any prompted target. We will explicitly add this overhead to our limitations.
>
> # Averaging Clarification (Q3)
> We agree that that the averaging section is mathematically ambiguous. We will revise the manuscript to clarify that the text outputs from the completions are first parsed into numerical trajectories using deterministic decoding rules. We aggregate trajectories by calculating the mean for each variable and time point separately.

---

> > ### Author Rebuttal · Reviewer_SH8J · 2026-04-02
> >
> > 1. The novelty compared to GenHPF is not adequately addressed, that model also operates on ehr tokenized as text and demonstrates mult-institution generalization. It's unclear what this does that GenHPF does not. Architecturally and data input wise they are very similar. Is it mainly training on a handfull of preselected forecasting and landmark prediction tasks? The authors should make it clear in the paper what is the core novelty relative to GenHPF.
> >
> > 2. The explanation of the sampling strategy which restricts output variables is still insufficiently clarified. How is this performed and how does it affect performance
> >
> > 3. Lack of datasets. This work only trains on one dataset, although clinical trials are provided for out of domain experiments. Do results extrapolate to new datasets. What if you train and evaluate on a new actual EHR dataset? Does performance remain high. The lack of presenting results on another EHR datasets indicates this is computationally challenging or the pipeline does not easily extend to other datasets like EHRSHOT, eICU, etc.
> >
> > 4. Compute. There should be an empirical comparison for compute in terms of inference and training wall times between this method and the other zero-shot methods. It is unclear if inference is scalable in this approach and only qualitative explanations are provided.

---

> > > ### Author Response · Authors · 2026-04-03
> > >
> > > We sincerely thank the reviewer for the continued engagement.
> > >
> > > # Core Novelty Compared to GenHPF (Q1)
> > >
> > > While GenHPF is a strong multi-source model, there are fundamental differences that define TwinWeaver's novelty that we will add in the manuscript:
> > > 1. *Architecture:* GenHPF trains classification heads on EHR embeddings. TwinWeaver generates direct natural language, enabling true zero-shot generation on unseen variables without retraining.
> > > 2. *Unified Tasks:* GenHPF focuses on classification of 12 tasks. TwinWeaver jointly forecasts numerical time-series (dozens of variables/indication) and discrete time-to-events.
> > > 3. *Reasoning:* Our text-based design can output biomedical hypotheses.
> > >
> > > Thus, TwinWeaver encompasses a substantially broader scope than a handful of preselected tasks.
> > >
> > > # Output Variable Sampling Strategy (Q2)
> > > We agree that the sampling strategy was insufficiently clarified.
> > >
> > > Unlike models requiring separate prediction heads for fixed medical codes (e.g., CLMBR-T), TwinWeaver’s unbounded natural language vocabulary enables zero-shot and multi-task predictions. To manage LLM context windows, we dynamically sample 1-3 target variables per instance. Sampling probability is weighted by observation frequency and volatility to prioritize clinically relevant, highly fluctuating variables (details Appendix B.1).
> > >
> > > *Effect on Performance:* While we forecast small subsets at inference to manage context limits, results suggest that the model effectively leverages multi-task learning. Correlating the total number of forecasted variables per indication against forecasting error (MASE) shows a lower error with more variables (Spearman ρ = -0.321; https://i.ibb.co/vCScq7ZG/scatter-features-vs-mase.png). This analysis will be added and we will add to future work to explore the effect of predicting more variables as LLMs with longer context windows are developed.
> > >
> > > # Extrapolation, Datasets, and Ease of Use (Q3)
> > >
> > > We agree that demonstrating easy, multi-dataset generalization is essential.
> > >
> > > * Extrapolation: To demonstrate the framework's adaptability, we go beyond zero-shot inference by training on out-of-distribution clinical trial data. Specifically, we fine-tuned our model on two distinct, large-scale clinical trials, each with custom processing pipelines: OAK ($n=1126$; atezolizumab vs. docetaxel) and IMpower131 ($n=949$; atezolizumab + carboplatin + paclitaxel variants), details Appendix D.3. We then evaluated its performance on two entirely unseen trials (POPLAR and IMpower130), demonstrating cross-trial extrapolation and generalization ("SFT" in the manuscript).
> > >
> > > * Datasets: It is important to emphasize that our primary real-world EHR dataset is not a single, monolithic dataset, but rather a collection of 20 distinct cancer indication datasets processed individually. To underscore this multi-dataset learning capability, Appendix D.5 directly compares models trained on isolated single-cancer datasets versus the unified pan-cancer model. The results show that training across 20 diverse datasets improves performance over single-indication models, supporting that the architecture is able to learn cross-cohort representations.
> > >
> > > * Ease of Use: To transparently demonstrate the pipeline's ease of use and its ability to ingest completely new, non-oncology EHR datasets, we have created a tutorial applying TwinWeaver to the public MIMIC-IV MEDS demo dataset (will be open-sourced). We summarize the key steps in the provided figure, illustrating how easily the framework adapts to the emerging MEDS data standard: https://i.ibb.co/chbjqHWT/meds-demo-tutorial-figure.png
> > >
> > > We have submitted access requests for additional public EHR datasets (e.g. EHRShot). Meanwhile, our clinical trial results serve as out-of-distribution tests, showing high performance on unseen data.
> > >
> > > # Compute, Scalability, and Empirical Comparisons (Q4)
> > >
> > > We acknowledge that TwinWeaver, being an LLM-based approach, is inherently, significantly slower than specialized baselines and we will further add this to the limitations. However, it is deployable in clinical and trial settings, running locally on a B200 GPU. Wall-time comparisons (clinical trials, in minutes) will be added to the Appendix:
> > >
> > > * Training (Mins, 2075 patients): Events: CLMBR-T(rep+head) <1, RSF 1.8, CLMBR-T(scratch) 7.8, MOTOR 25.8, GDT 154.4. Forecasts: TiDE(uni) 23.2, TiDE(all) 24.0, Chronos 109.2, GDT 331.0.
> > > * Inference (Mins, 943 patients): Events: RSF <1, CLMBR-T 2.1, MOTOR 13.0, LLama 68.0, GDT 68.5, II-Medical 70.9. Forecasts: TiDE(uni) 2.1, Chronos Bolt 2.2, Chronos 13.9, Llama 127.5, GDT 153.9, II-Medical 371.8.
> > >
> > > As LLM inference speeds grow (e.g. Zandieh et al, 2025, arXiv:2504.19874), this framework will directly benefit since it runs natively on the same infrastructure. Importantly, the per patient inference remains at under 1 minute, making it clinically deployable, a setting where high predictive performance is paramount.

---

### Decision · Program_Chairs · 2026-04-30

**Decision:**

Accept (regular)

**Comment:**

This paper studies a practically relevant problem: using language models for unified clinical forecasting and event prediction from longitudinal patient histories serialized as text. Reviewers generally agreed that the paper is well written, the empirical evaluation is broad, and the out-of-distribution clinical trial results are promising.

The main concerns were about novelty relative to prior text-based and generative EHR models, computational efficiency, and the interpretation of the event modeling formulation. I have read the reviews, rebuttal, and discussion carefully. The rebuttal clarified several of these points, added stronger comparisons and analyses, and improved the paper’s positioning. While some limitations remain, especially around efficiency and claim scope, I find the overall contribution technically solid and useful.

I therefore recommend accept. In the final version, the authors should sharpen the novelty claims and state the computational and modeling limitations more explicitly.